# CoHIT: a one-pot ultrasensitive ERA-CRISPR system for detecting multiple same-site indels

Yin Liu[1,2,3,9], Xinyi Liu[4,9], Dongyi Wei[5], Lu Dang[6], Xiaoran Xu[5], Shisheng Huang[7], Liwen Li[4], Sanyun Wu[1], Jinxian Wu[1], Xiaoyan Liu[1], Wenjun Sun[8], Wanyu Tao[8], Yongchang Wei[5], Xingxu Huang [8], Kui Li [4] ✉, Xinjie Wang [4] ✉ & Fuling Zhou [1] ✉

Genetic testing is crucial for precision cancer medicine. However, detecting multiple same-site insertions or deletions (indels) is challenging. Here, we introduce CoHIT (Cas12a-based One-for-all High-speed Isothermal Test), a one-pot CRISPR-based assay for indel detection. Leveraging an engineered AsCas12a protein variant with high mismatch tolerance and broad PAM scope, CoHIT can use a single crRNA to detect multiple *NPM1* gene c.863_864 4-bp insertions in acute myeloid leukemia (AML). After optimizing multiple parameters, CoHIT achieves a detection limit of 0.01% and rapid results within 30 minutes, without wild-type cross-reactivity. It successfully identifies *NPM1* mutations in 30 out of 108 AML patients and demonstrates potential in monitoring minimal residual disease (MRD) through continuous sample analysis from three patients. The CoHIT method is also competent for detecting indels of *KIT*, *BRAF*, and *EGFR* genes. Integration with lateral flow test strips and microfluidic chips highlights CoHIT's adaptability and multiplexing capability, promising significant advancements in clinical cancer diagnostics.

With the rapid development of molecular genetics and sequencing technologies, genetic testing has gained increasing importance in precision cancer medicine[1,2]. Many gene mutations have now been included in clinical guidelines to help clinicians diagnose disease, select therapies and monitor cancer recurrence[3–5]. For instance, the measurement of minimal residual disease (MRD) levels, which indicate the remaining quantity of malignant cells in the body, has important prognostic and therapeutic implications[6,7]. Moreover, the early detection of cancer-related mutations may provide opportunities for early intervention[8]. Common types of gene mutations include base

substitutions and insertions and deletions (indels). The frequency of indels in the cancer genome is only approximately 10% of base substitutions[9], but indels are more likely to cause frameshifts, resulting in abnormal protein expression and promoting cancer development. For instance, the c.863_864 4-bp insertions in the Nucleophosmin 1 (*NPM1*) gene drove approximately 30% of acute myeloid leukaemia (AML) cases by replacing the seven C-terminal amino acids with another eleven amino acids, leading to abnormal cytoplasmic localization of the NPM1 protein[10]. Other typical examples include exon 19 deletions and exon 20 insertions in the *EGFR* gene[11,12], exon 20

[1]Department of Hematology, Zhongnan Hospital of Wuhan University, Wuhan University, Wuhan, China. [2]Wuhan University Shenzhen Research Institute, Shenzhen, China. [3]College of Chemistry and Molecular Sciences, Wuhan University, Wuhan, China. [4]Shenzhen Branch, Guangdong Laboratory of Lingnan Modeatarn Agriculture, Genome Analysis Laboratory of the Ministry of Agriculture and Rural Affairs, Agricultural Genomics Institute at Shenzhen, Chinese Academy of Agricultural Sciences, Shenzhen, China. [5]Department of Radiation and Medical Oncology, Zhongnan Hospital of Wuhan University, Wuhan University, Wuhan, China. [6]Department of Reproductive Medicine, Third Affiliated Hospital of Guangzhou Medical University, Guangzhou, China. [7]Zhejiang Lab, Hangzhou, China. [8]School of Life Sciences and Technology, ShanghaiTech University, Shanghai, China. [9]These authors contributed equally: Yin Liu, Xinyi Liu. ✉e-mail: likui@caas.cn; wangxinjie@caas.cn; zhoufuling@whu.edu.cn

insertions in the *HER2* gene[13], β3-αC loop deletions in the *BRAF* gene[14], and exon 11 deletions in the *KIT* gene[15]. The increasing emergence of mutation markers and mutation-targeted therapies in cancer highlights the need for relevant genotyping methods[16].

Current clinical approaches for detecting indels mainly include first-generation sequencing (FGS, also known as Sanger sequencing), next-generation sequencing (NGS), real-time quantitative polymerase chain reaction (qPCR), and droplet digital PCR (ddPCR)[17–19]. However, these methods have their limitations in clinical indel detection. For instance, both FGS and NGS require expensive sequencers and complex operations, and these requirements limit their widespread adoption, particularly in resource-constrained settings. In addition, FGS has a limit of detection (LoD) of 10%[20], rendering it inadequate for the identification of mutations present in low abundance. NGS has a long turnaround time of approximately 10 days[21], which may be incompatible with the demands of timely clinical decision-making. Conversely, qPCR and ddPCR tests are fast (2 - 4 h) and widely used but require costly equipment and cannot easily cover multiple indels in one test. Notably, multiple indel types at a single gene site are common in cancer-related mutations[19,22]. For example, more than one hundred c.863_864 4-bp insertions in the *NPM1* gene have been reported in AML patients[10]. Therefore, there is still a need to develop a sensitive, rapid, low-cost, convenient method that can be applied to multiple variants for cancer-related indel detection.

Recently, clustered regularly interspaced short palindromic repeats (CRISPR) and CRISPR-associated (Cas) proteins have been widely used as molecular diagnostic tools[23–25]. Among them, the CRISPR/Cas12a system has both specific cis-cleavage activity and nonspecific trans-cleavage activity[23]. Guided by a CRISPR RNA (crRNA) that carries a target recognition sequence, the Cas12a protein can bind and cleave the target double-stranded DNA (dsDNA) with a short protospacer adjacent motif (PAM). At the same time, the nonspecific trans-cleavage activity of Cas12a is activated, which can result in the cleavage of nearby single-stranded DNA (ssDNA) labelled with fluorescence reporters. This "collateral cleavage" allows for sensitive and rapid detection of the target DNA. The most commonly used Cas12a proteins, *Acidaminococcus sp*. Cas12a (AsCas12a) and *Lachnospiraceae bacterium* Cas12a (LbCas12a), have been applied in the diagnosis of many infectious and noninfectious diseases[26–30]. Compared to LbCas12a, AsCas12a shows stronger mismatch tolerance and can therefore be used for the simultaneous detection of multiple similar gene variants[30].

Here, we combine CRISPR/Cas12a with enzymatic recombinase amplification (ERA) and develop a one-pot ERA-Cas12a assay, termed CoHIT (Cas12a-based One-for-all High-speed Isothermal Test), for the detection of multiple indels at same genetic locus using only a single target crRNA. In this assay, an unreported engineered enAsCas12a-Ultra-R (enAsU-R) protein is utilized because of its high mismatch tolerance and wide NNCV and TTTV PAM. We show the ability of CoHIT in detecting multiple prevalent *NPM1* gene c.863_864 4-bp indels, achieving a LoD as low as 0.01% within a 30-minute timeframe, and without a wild-type (WT)-induced cross signal. Furthermore, we explore the versatility of CoHIT through the detection of other cancer gene indels including the *KIT* gene p.W557_K558del, the *BRAF* gene p.V487_T491del, and the *EGFR* gene p.E746_A750del. In clinical applications, we use the CoHIT system to detect 108 AML patient samples and 17 MRD samples, and show its clinical practicability. Moreover, the combination of CoHIT with a lateral flow strip assay (LFA) and a microfluidic chip has established its potential as a streamlined, sensitive, and rapid diagnostic tool for cancer management.

## Results

### Workflow of the CoHIT system

Variable indels at the same gene site are a common occurrence in cancer-related mutations. For instance, more than one hundred 4-bp insertions at the *NPM1* gene c.863_864 site in AML patients have been reported[10]. To address the challenge of detecting multiple indel variants in a one-pot reaction, we developed the CoHIT assay, which includes an unreported engineered AsCas12a protein that can target multiple variants with only a single crRNA. By combining functional amino acid substitutions, we engineered AsCas12a to exhibit increased mismatch tolerance and expanded PAM performance, enabling the recognition of dsDNA with a few base mismatches to the crRNA sequence (Fig. 1a). Briefly, in the workflow of the CoHIT system, genomic DNA from a patient is directly added to a one-pot ERA-Cas12a reaction, after which real-time fluorescence is read within 30 min using a portable fluorescence detector at a constant temperature of 39 °C (Fig. 1b). In particular, the reaction system includes ERA reagents, a pair of primers, FAM-ssDNA-BHQ1 reporters, the engineered AsCas12a protein enAsU-R, and a single crRNA. Multiple indel variants (e.g. Variants 1 - 3 with 4-bp insertions, which mutually differ by only a few bases) can be recognized by the Cas12a/crRNA complex, triggering collateral ssDNA cleavage activity and the emission of a green fluorescence signal. However, the WT DNA fails to induce a fluorescence signal due to its distinct dissimilarity from the CRISPR RNA (crRNA) recognition sequence, thereby ensuring the specificity of the CoHIT assay (Fig. 1b). This one-pot integration of ERA and CRISPR/Cas12a increases detection efficiency and prevents uncapping-related cross-contamination.

### Engineering Cas12a variants to improve mismatch tolerance

To detect multiple same-site indels with one crRNA, we engineered and selected a Cas12a variant that could tolerate more base mismatches. Initially, we compared the mismatch tolerance of two commonly used WT Cas12a proteins, AsCas12a and LbCas12a, by an in vitro cleavage assay. Specifically, 2E10 copies of DNA fragments were used as substrates, including the original sequence (OS) that is completely complementary to the crRNA and three sequences with two or three base mismatches at different locations. The results demonstrated that AsCas12a has a higher mismatch tolerance than LbCas12a (Supplementary Fig. 1). To further increase the mismatch tolerance, we performed additional protein engineering on AsCas12a through amino acid substitution. We generated two previously unreported AsCas12a protein variants, enAsU-R and enAsU-VR, by combining reported functional amino acid substitutions including E174R, M537R, S542R, K548R/V, N552R, K607R and F870L (Fig. 2a, b). Among these substitutions, E174R, S542R, and K548R are from the enAsCas12a (enAs) variant which has an expanded target range[31], M537R and F870L are from the AsCas12a-Ultra (AsU) variant which has improved gene editing efficiency[32], and K548V, N552R, and K607R are from the AsCas12a-RVR variant and the AsCas12a-RR variant which recognize TATV and TYCV PAMs, respectively[33]. Then, we compared the mismatch tolerance of enAsU-R and enAsU-VR with four reported AsCas12a variants, including WT AsCas12a, enAs, AsU, and enAsCas12a-Ultra (enAsU)[32]. Specifically, DNA fragments of four sites (Sites 1 - 4) were used as substrates, each including the OS and 10 different mismatch sequences (M1 - M10) (Fig. 2c). Among them, Site 1 and Site 2 had two-base mismatches at different locations, and Site 3 and Site 4 had three base mismatches. After a 30 min of in vitro cleavage reaction at 37 °C, the fluorescence intensity results showed that all AsCas12a variants had different degrees of mismatch tolerance for the four sites. Notably, although the locations with the highest tolerance differed among the four sites, the high-tolerance locations of the different AsCas12a variants were basically consistent for each same site. This indicates that these amino acid substitutions of AsCas12a have little influence on the mismatch tolerance location. To compare the mismatch tolerance among AsCas12a variants, we calculated the mismatch sample cases with a normalized fluorescence (NF) > 0.5. The results showed that enAsU-R have the most NF > 0.5 cases for all four sites, with 3, 4, 5 and 2 cases, respectively, indicating the highest mismatch tolerance (Fig. 2c).

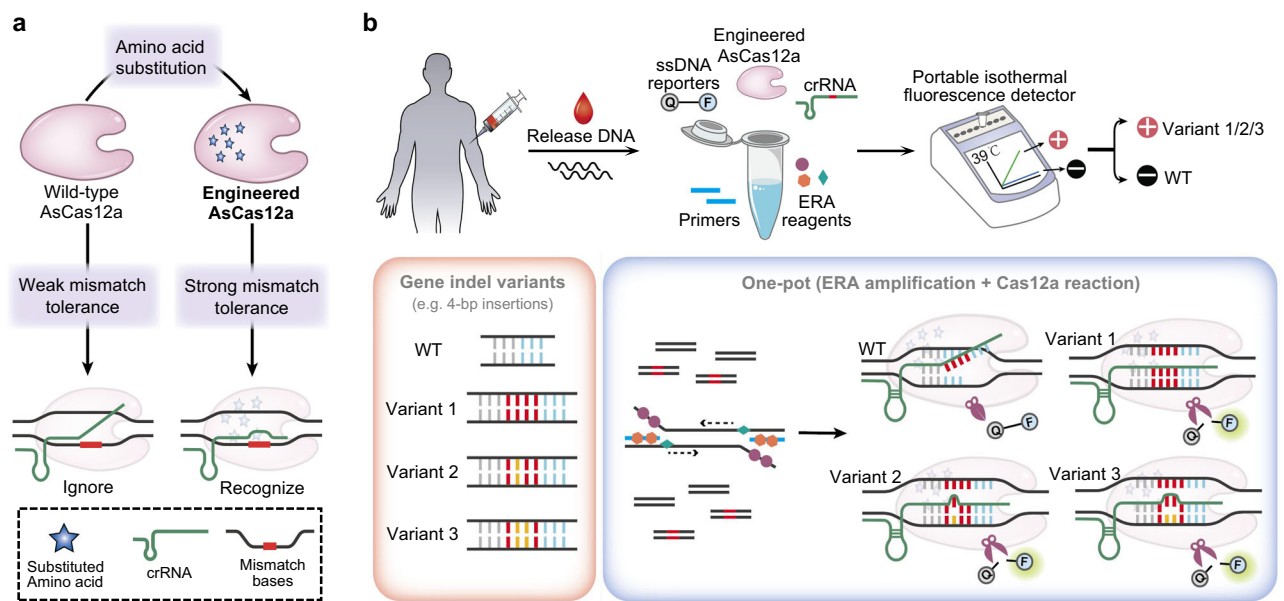

**Fig. 1 | Workflow of the CoHIT system. a** By the introduction of amino acid substitutions, the engineered AsCas12a protein was endowed with stronger mismatch tolerance than the WT AsCas12a protein and was able to recognize dsDNA with a few mismatched bases relative to the crRNA. **b** Genomic DNA of a patient is extracted from the cell sample and detected using the one-pot ERA-Cas12a reaction. In addition to the WT sequence, the target DNA site can contain indels with several possible variants (e.g. Variants 1 – 3 with 4-bp insertions). The reaction mixture contains ERA reagents, a pair of primers, FAM-ssDNA-BHQ1 reporters, the engineered AsCas12a protein enAsU-R, and a single crRNA. During the isothermal

amplification reaction at 39 °C, both the WT and Variants 1 – 3 are rapidly amplified. At the same time, the Cas12a/crRNA complex captures the Variant 1 amplicons through sequence complementation, as well as the Variant 2 and 3 amplicons due to the mismatch tolerance of the engineered AsCas12a protein, and then, green fluorescence signals are emitted. However, WT amplicons are ignored because of the quite difference from the crRNA recognition sequence, and no cross signal is observed. The detection can be completed within 30 min using only a portable isothermal fluorescence detector. Source data are provided as a Source Data file.

Overall, our screen identified an unreported AsCas12a variant, enAsU-R, for use to establish a mismatch-tolerant CoHIT system for detecting multiple similar DNA sequences with a single crRNA.

## Detection of multiple *NPM1* mutations via an enAsU-R Cas12a-induced in vitro cleavage assay

To determine the crRNA design rules of the enAsU-R Cas12a, we conducted a PAM identification assay on enAsU-R, as well as on enAs and enAsU for comparison. The results showed that enAsCas12a has a PAM preference for TTYN, VTTV and TRTV (Supplementary Fig. 2a), which is highly consistent with previous reports[31]. enAsU Cas12a displayed a PAM feature similar to that of enAs but with higher cleavage of many PAM types (Supplementary Fig. 2a)[32]. Remarkably, the enAsU-R Cas12a, displayed a wide and distinctive PAM preference for NNCV and TTTV (Fig. 3a). As shown in Fig. 3b, there was no significant difference between enAsU and enAsU-R on TTTV PAM. However, for NNCV, especially for NVCA, RVCC, and NNCG PAM, enAsU-R showed more than twice the cleavage efficiency of enAs and enAsU (32.55% vs. 10.97% vs. 14.68% for NCVA, 41.98% vs. 10.94% vs. 17.72% for RVCC, 32.66% vs. 9.95% vs. 13.92% for NNCG) (Fig. 3b). Furthermore, on the VMCV PAM, enAsU-R displayed approximately three times the cleavage efficiency of enAs and enAsU (41.80% vs. 10.02% vs. 15.80%, respectively) (Supplementary Fig. 2b). This broad PAM recognition capability significantly simplifies crRNA design for most mutation sites within the human genome.

To confirm the ability of enAsU-R Cas12a to detect multiple insertions with a single crRNA, we selected the *NPM1* gene c.863_864 4-bp insertions. Among the 256 ($4^4$) insertion possibilities, 11 types account for more than 97% of the *NPM1*-mutated AML cases[10]. Specifically, c.863_864insTCTG (1793/2348, 76.4%), insCATG (215/2348, 9.2%), and insCCTG (168/2348, 7.2%) represent the most common types[10] (Fig. 3c, Supplementary Data 1). Considering the minimal nucleotide divergence of insCCTG from the six most frequent

insertion types, we designed crRNAs (*NPM1*-crRNA1 – 6) complementary to the insCCTG sequence rather than the most common insTCTG sequence, to obtain stronger fluorescence signals for multiple frequent insertions (Fig. 3d and Supplementary Fig. 3). By comparing six crRNAs across six AsCas12a variants, we measured the fluorescence ratio of insTCTG to WT DNA to identify the most specific crRNA. enAsU-R/crRNA1 exhibited good specificity and the highest insTCTG signal (Fig. 3e). To evaluate whether *NPM1* insertions other than insTCTG could be detected by enAsU-R/crRNA1, DNA fragments harbouring c.863_864insNNNN (N = A/G/C/T) sequences were synthesized and TA cloned. Then 32 random TA clones (X1 – 32) were PCR-amplified and purified for Cas12a/crRNA1-induced in vitro cleavage assay (Fig. 3f). More clones with higher fluorescence signals were detected by enAsU-R than by the other five Cas12a variants (Fig. 3g and Supplementary Fig. 4a). Subsequent Sanger sequencing showed that most of the 32 clones had three or four mismatches to *NPM1*-crRNA1, further confirming the heightened mismatch tolerance of the enAsU-R protein (Supplementary Data 2). Finally, by employing enAsU-R/crRNA1, we detected the top eleven *NPM1* gene c.863_864 4-bp insertions, including insTCTG, CATG, CCTG, CCAG, CCGG, CTTG, TATG, TCGG, TAAG, CAGG, and CAGA, covering more than 97% of the *NPM1*-mutated AML cases in clinical practice. enAsU-R consistently produced robust fluorescence signals for all the tested insertions, outperforming the other five Cas12a variants, and no significant WT cross-reactivity was observed (Fig. 3h and Supplementary Fig. 4b, c). Overall, we identified the NNCV and TTTV PAM preference of enAsU-R Cas12a and confirmed its ability to detect multiple insertions with a single crRNA.

## Establishment and optimization of the CoHIT system

For rapid and convenient detection, we combined Cas12a/crRNA detection with ERA isothermal amplification to establish a one-pot ERA-Cas12a assay, CoHIT. Taking *NPM1* insertion detection as an example, we used 1E4 copies of the plasmid template, carrying the

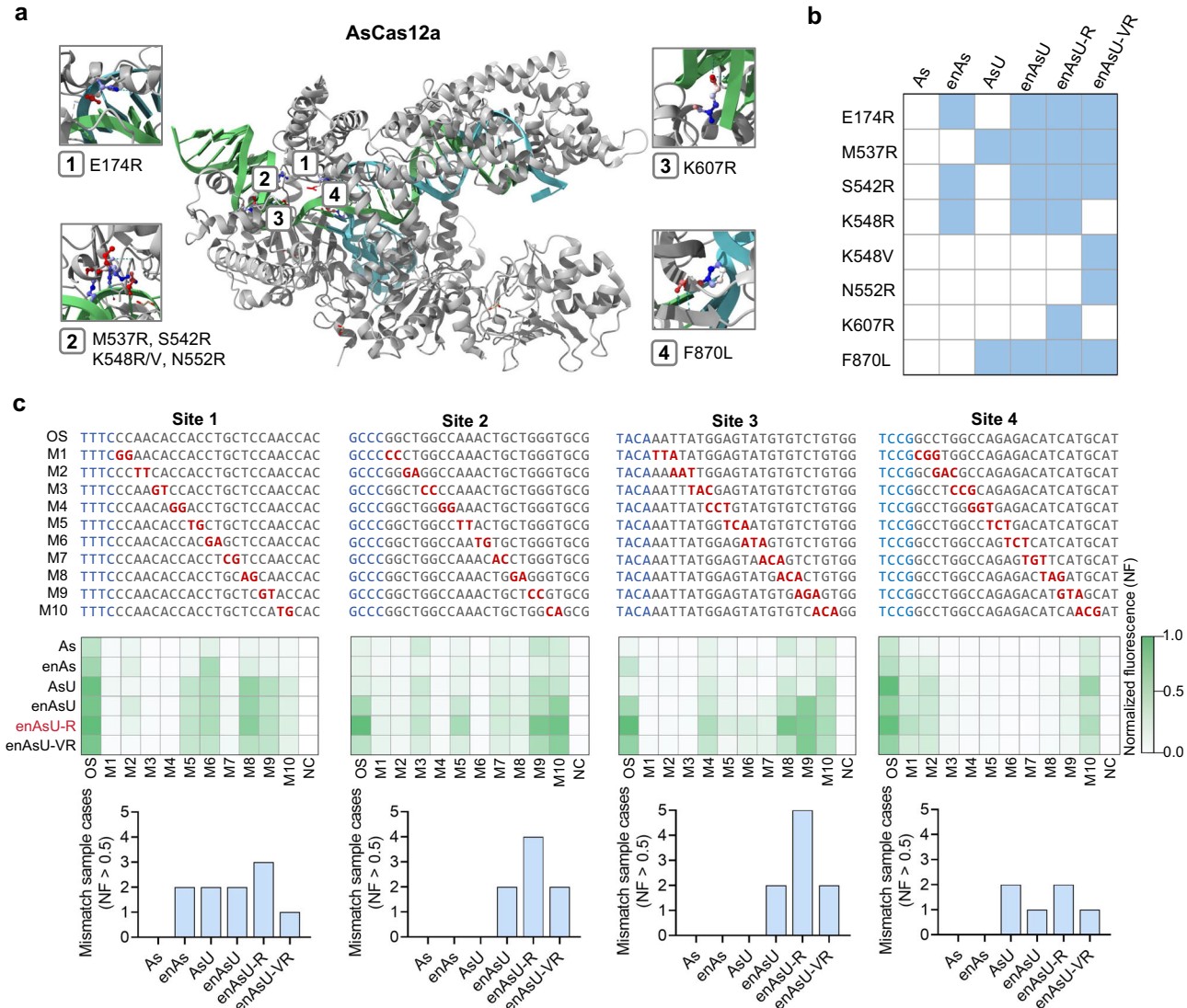

**Fig. 2 | Engineering of Cas12a variants to improve mismatch tolerance. a** 3D structure of AsCas12a protein (PDB ID: 5B43) and locations of amino acid substitutions introduced during engineering. **b** Amino acid substitutions included in each AsCas12a variant are marked in blue. **c** Mismatch tolerance assay of six AsCas12a variants. The top panel shows the original sequence (OS) and mismatch sequences (M1 - M10) of each investigated site (Sites 1 - 4), with PAMs marked in blue and mismatches marked in red. The checkerboard diagrams in the middle panel show the relative fluorescence intensities of OS and M1 - M10 detected by different Cas12a variants, with 2E10 copies of DNA substrates and reaction for 30 min at 37 °C. The bottom histograms show the number of mismatch sample cases (M1 - M10) with an NF > 0.5 for each site. NF, normalized fluorescence. Source data are provided as a Source Data file.

533-bp c.863_864insTCTG fragment to screen the best ERA primer pair (F5R4, Fig. 4a, b) from five forward primers (F1 - 5) and four reverse primers (R1 - 4) through a traditional two-step assay. Then, we attempted to establish a one-pot system by directly combining the ERA system with the CRISPR/Cas12a in vitro cleavage system. This initial system was subjected to multiparametric optimization, including the buffer compositions, reaction temperatures, and concentrations of reaction components (Fig. 4c). First, we found that the presence of NEBuffer3.1 significantly inhibited the fluorescence signal of the one-pot assay, alternative NEBuffers (1.1, 2.1, and CutSmart) showed less inhibition, and omitting NEBuffer resulted in the strongest fluorescence signal (Fig. 4d and Supplementary Fig. 5a). Second, seven ERA reaction buffers (original RB and RB1 - 6) were screened by a 30 min one-pot assay at 37 °C. The results showed that RB1 was the best for the one-pot system, yielding approximately twice the fluorescence intensity of the original RB (Fig. 4e and Supplementary Fig. 5b). Then, we explored different reaction temperatures, including 37 °C, 39 °C, and 42 °C, and found that 39 °C resulted in the highest fluorescence signal,

which was further confirmed for the other top five insertion templates (Fig. 4f and Supplementary Fig. 6). Additionally, the concentrations of the FAM-ssDNA-BHQ1 reporter, enAsU-R Cas12a protein, and crRNA were sequentially screened to further optimize the CoHIT system. The results showed that the 1 pmol/μL FAM-ssDNA-BHQ1 reporter had the highest signal-to-noise ratio, and the 8 ng/μL enAsU-R Cas12a protein and 20 fmol/μL crRNA induced the strongest fluorescence signal (Fig. 4g–i and Supplementary Fig. 7). After optimization of the above parameters, the final CoHIT system exhibited a 40 × increase in fluorescence intensity for the detection of the *NPM1* gene c.863_864insTCTG (Fig. 4j).

To further determine the LoD of the optimized CoHIT system, we first used it to detect 1E6 copies of plasmid templates with gradient *NPM1* mutation ratios ranging from 100% to 0.01% by mixing mutated and WT plasmids. The results indicated that the LoD of the CoHIT system was 0.01% for the most common insTCTG mutation, and 0.01%, 0.1%, and 0.01% for the other three common mutations, insCCAG, insCATG, and insCTTG, respectively (Fig. 4k, l and Supplementary Fig. 8).

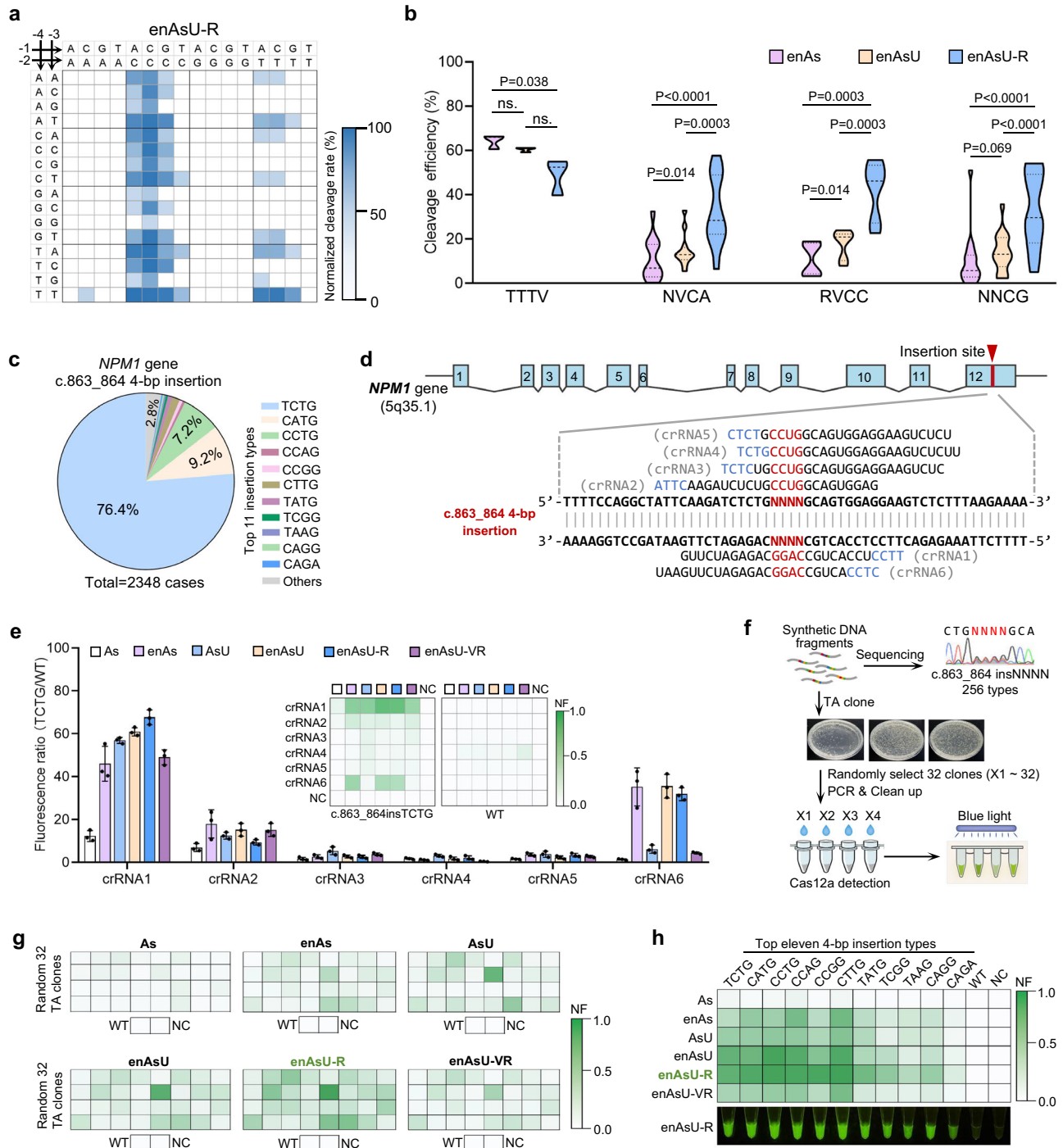

**Fig. 3 | Detection of multiple NPM1 mutations via an enAsU-R Cas12a-induced in vitro cleavage assay. a** PAM identification assay of enAsU-R Cas12a. Normalized cleavage rates of all possible 4-base PAMs are shown at different blue depths. **b** Cleavage efficiency comparison of enAs, enAsU, and enAsU-R Cas12a on PAM TTTV (n = 3 targets), NVCA (n = 12 targets), RVCC (n = 6 targets), and NNCG (n = 16 targets), respectively. *P* values are determined by two-tailed Student's t-tests, ns., no significance. **c** Proportions of different *NPM1* gene c.863_864 4-bp insertion types in 2348 *NPM1*-mutated AML patients. The most common 11 types are marked in different colours. **d** The genomic location of the c.863_864 4-bp insertion is shown above and marked by a red triangle. The blue blocks represent exons of the *NPM1* gene. The mutant DNA sequence and six designed crRNAs are shown below, with the inserted bases marked in red and PAMs marked in blue. The PAM is not included in the synthesized crRNA. N = A/C/T/G. **e** Comparison of 36 Cas12a/crRNA

pairs by detecting 2E10 copies of insTCTG and WTDNA substrates, reacting for 30 min at 37 °C. Values and error bars reflect the means and standard deviation (s.d.) of three biological replicates. The chessboard diagrams display the final relative fluorescence intensity, and the histogram shows the fluorescence intensity ratio (TCTG/WT). **f** Flowchart of the TA clone-based mismatch tolerance assay. **g** The six chessboard diagrams display the normalized fluorescence intensity of six Cas12a/crRNA1-induced in vitro detection of the PCR purified products of 32 random TA clones, WT, and NC. **h** The normalized fluorescence intensity of six Cas12a/crRNA1-induced in vitro detection of the top 11 *NPM1* gene c.863_864 4-bp insertions using 5E10 copies of DNA as substrates, and reacting for 20 min at 37 °C. The fluorescence image shows the naked-eye result of enAsU-R under a blue light. Source data are provided as a Source Data file.

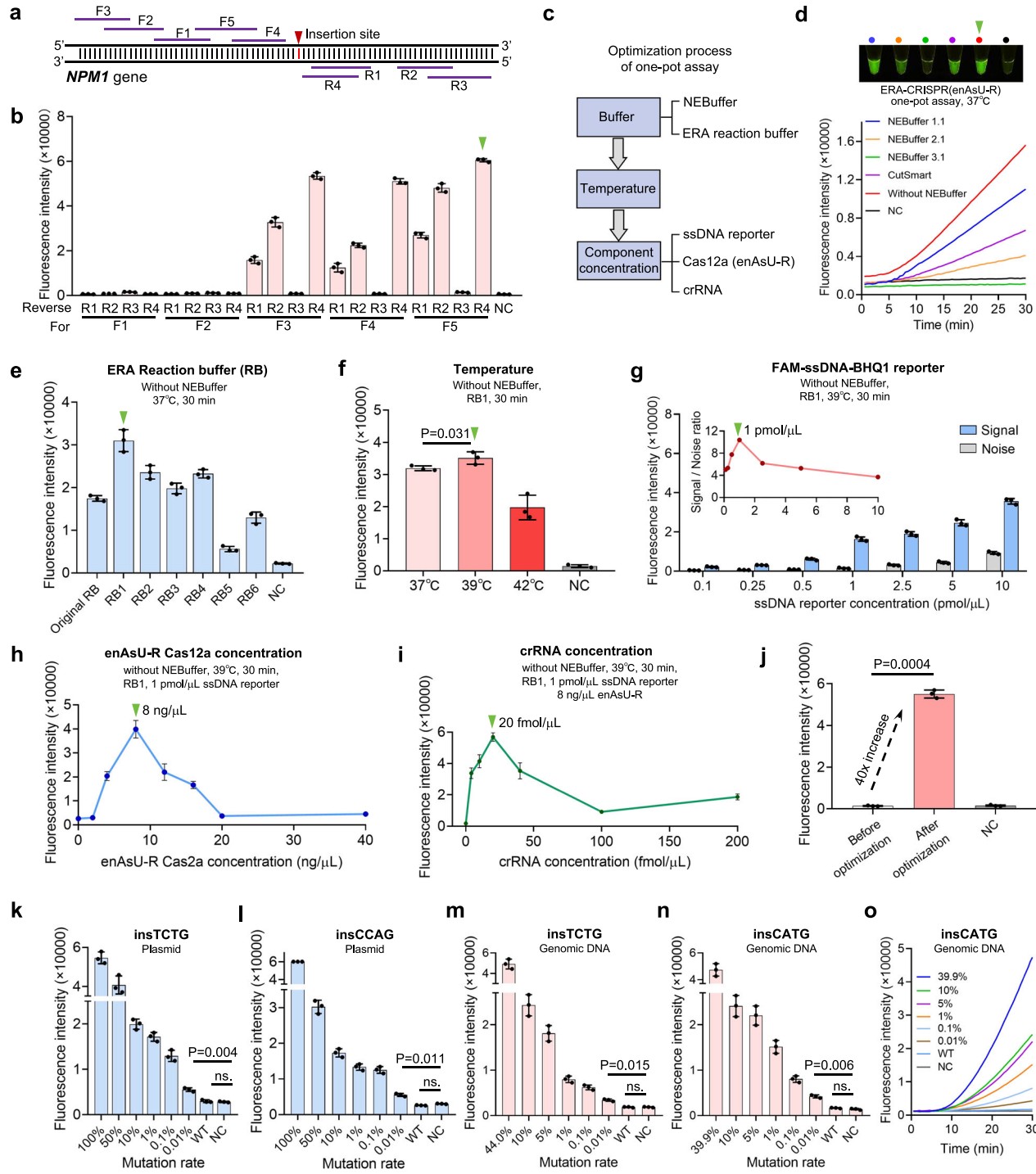

**Fig. 4 | Establishment and optimization of the CoHIT system. a** Locations of the ERA primers designed for the *NPM1* target. **b** Amplification efficiency of 20 ERA primer pairs. 1E4 copies of plasmids carrying the *NPM1* gene c.863_864insTCTG fragment were amplified for 20 min at 37 °C and then detected by an enAsU-R/ crRNA1 in vitro cleavage assay for 20 min at 37 °C. Values and error bars reflect the means and s.d. of three biological replicates. The best pair F5R4 is indicated by a green triangle. **c** Optimization process of the ERA-CRISPR one-pot assay. **d**–**f** Comparison of different NEBuffers, ERA reaction buffers, and reaction temperatures for the one-pot system using 1E6 copies of the insTCTG plasmid as a template, reacting for 30 min. The best option for each optimization experiment is indicated by a green triangle. The updated reaction conditions are shown below the titles. Values and error bars reflect the means and s.d. of three biological replicates. *P* values are determined by two-tailed Student's t-tests. **g** Concentration optimization of the FAM-ssDNA-BHQ1 reporter. For the Signal group, the insTCTG plasmid was used as the template, and for the Noise group, ddH2O was used as the

template. Values and error bars reflect the means and s.d. of three biological replicates. The included chart shows the final fluorescence intensity ratio of Signal/ Noise. **h**, **i** Concentration optimization of the enAsU-R Cas12a protein and crRNA. Values and error bars reflect the means and s.d. of three biological replicates. **j** Comparison between one-pot systems before and after optimizations for detecting insTCTG mutation. **k**, **l** Plasmid-based LoD assay of the CoHIT system for insTCTG and insCCAG mutations using 1E6 copies of plasmids as templates, reacting for 30 min at 39 °C. Values and error bars reflect the means and s.d. of three biological replicates. *P* values are determined by two-tailed Student's t-tests. **m**, **n** Genomic DNA-based LoD assay of the CoHIT system for insTCTG and insCATG mutations using 100 ng genomic DNA as a template, reacting for 30 min at 39 °C. Values and error bars reflect the means and s.d. of three biological replicates. *P* values are determined by two-tailed Student's t-tests. **o** Time–course results of the genomic DNA-based LoD assay for insCATG detection. Source data are provided as a Source Data file.

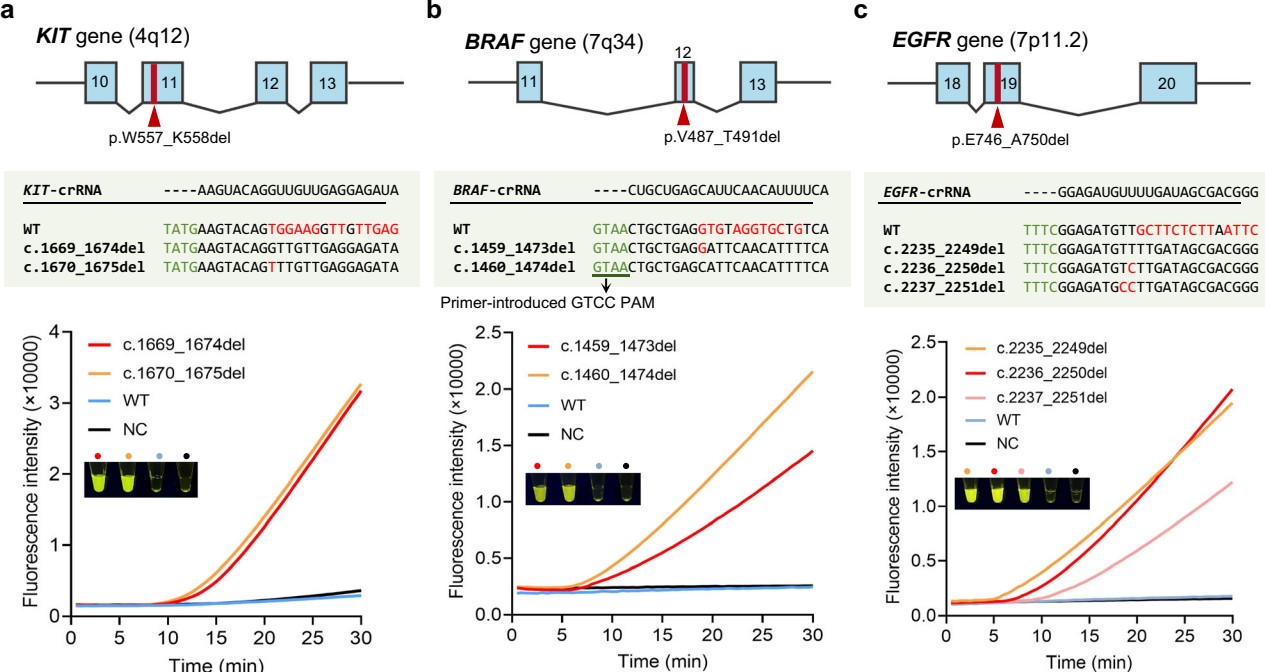

**Fig. 5 | Detection of other cancer-related gene indels using the CoHIT system.** **a** The upper panel shows the genomic location of the *KIT* gene p.W557_K558del mutation, and the blue blocks represent exons. The middle panel lists the sequences of the variants-targeted crRNA and the DNA of the WT and two deletion variants. PAM is marked in green, and DNA bases mismatched to the crRNA are marked in red. The chart below shows the time–course results of CoHIT detection of two variants, the WT, and NC. The included fluorescence image is the naked-eye result under blue light after 30 min of reaction. **b**, **c** Results of CoHIT detection of the *BRAF* p.V487_T491del variants and *EGFR* p.E746_A750del variants. Similar information to that in Fig. 5a is shown. Source data are provided as a Source Data file.

To simulate a real-world clinical testing scenario, we applied the CoHIT assay to genomic DNA samples from three AML patients. These samples contained 44.0% insTCTG, 39.9% insCATG, and 46.7% insCCTG mutations, respectively. Various mutant-to-WT ratios ranging from 10% to 0.01% of the mutated samples were generated by mixing their DNA with genomic DNA from healthy WT individuals. With 100 ng of genomic DNA input, the CoHIT system successfully achieved an LoD of 0.01% for all three insertion types (Fig. 4m–o and Supplementary Fig. 9). Notably, only a single *NPM1*-crRNA1 sequence complementary to the inCCTG sequence was used to target all the abovementioned insertion types, and there was no observable WT-induced cross signal. These findings demonstrated the one-for-all capability and high specificity of the CoHIT system. Furthermore, the use of the CoHIT system is rapid (within 30 min), convenient (a one-step and isothermal reaction), and inexpensive (no need for expensive reagents or highly trained operators or analysts).

### Detection of other cancer-related gene indels using the CoHIT system

To further demonstrate the versatility of the CoHIT system, we detected other cancer-related gene indels. We first tested the *KIT* gene p.W557_K558del mutation, which has been strongly linked to liver metastasis of gastrointestinal stromal tumours and has two common clinical variants, c.1669_1674del and c.1670_1675del (Fig. 5a)[15]. After screening to identify a crRNA that can detect both variants with high sensitivity and with no WT-induced cross signal, six pairs of ERA primers were screened for the one-pot assay (Supplementary Fig. 10). Then, 1E5 copies of plasmid templates carrying the c.1669_1674del, c.1670_1675del, and WTDNA fragments were detected at 39 °C. The time–course results showed that both deletion variants began to emit rapidly increasing fluorescence signals after approximately 10 min, and no significant cross signal was observed in the WT sample, indicating that the CoHIT system was sensitive and specific (Fig. 5a).

Notably, although the c.1670_1675del variant has a 1-base mismatch to *KIT*-crRNA, its fluorescence signal was as strong as that of the c.1669_1674del variant, confirming the ability of the CoHIT system to detect different deletions using a single crRNA.

In addition to *KIT* gene deletion, we employed the CoHIT system to detect indels in the *BRAF* and *EGFR* genes. The *BRAF* gene p.V487_T491del mutation is an oncogenic mutation in the β3-αC loop and has two important variants, c.1459_1473del and c.1459_1474del (Fig. 5b)[14]. The *EGFR* gene p.E746_A750del mutation is a well-known TKI-sensitive mutation in lung cancer and has three most common variants, c.2235_2249del, c.2236_2250del, and c.2237_2251del (Fig. 5c)[13]. Similar to *KIT* detection, the CoHIT system displayed good sensitivity and specificity for both the *BRAF* gene p.V487_T491del variants and *EGFR* gene p.E746_A750del variants (Fig. 5b, c and Supplementary Figs. 11, 12). Moreover, we tested the ability of the CoHIT system to detect 2-bp and 1-bp small indels using three simulated targets, and the results showed that three to four variants of each target could be sensitively detected using a single crRNA, with no significant WT-induced cross signal (Supplementary Fig. 13). These successful applications further demonstrate the versatility and efficacy of the CoHIT system in gene indel detection.

### Detection in clinical samples using the CoHIT system

To assess the clinical applicability of the CoHIT system, we detected the *NPM1* mutation status in 108 AML patients (P1 - 108). Before large-scale detection, the blood genomic DNA of three *NPM1*-mutated AML patients (P9 with 12.3% c.863_864insTCTG, P33 with 39.9% insCATG, and P111 with 44.0% insTCTG) was tested to determine the suitable input template amount for the CoHIT assay. As shown in Fig. 6a, inputs of 10, 25, 50, 100, 250, 500, and 1000 ng of genomic DNA were compared. As the input amount increased, the fluorescence signal first increased and then decreased, and a range of 50–250 ng was identified as suitable for the input template amount (Fig. 6a and Supplementary Fig. 14).

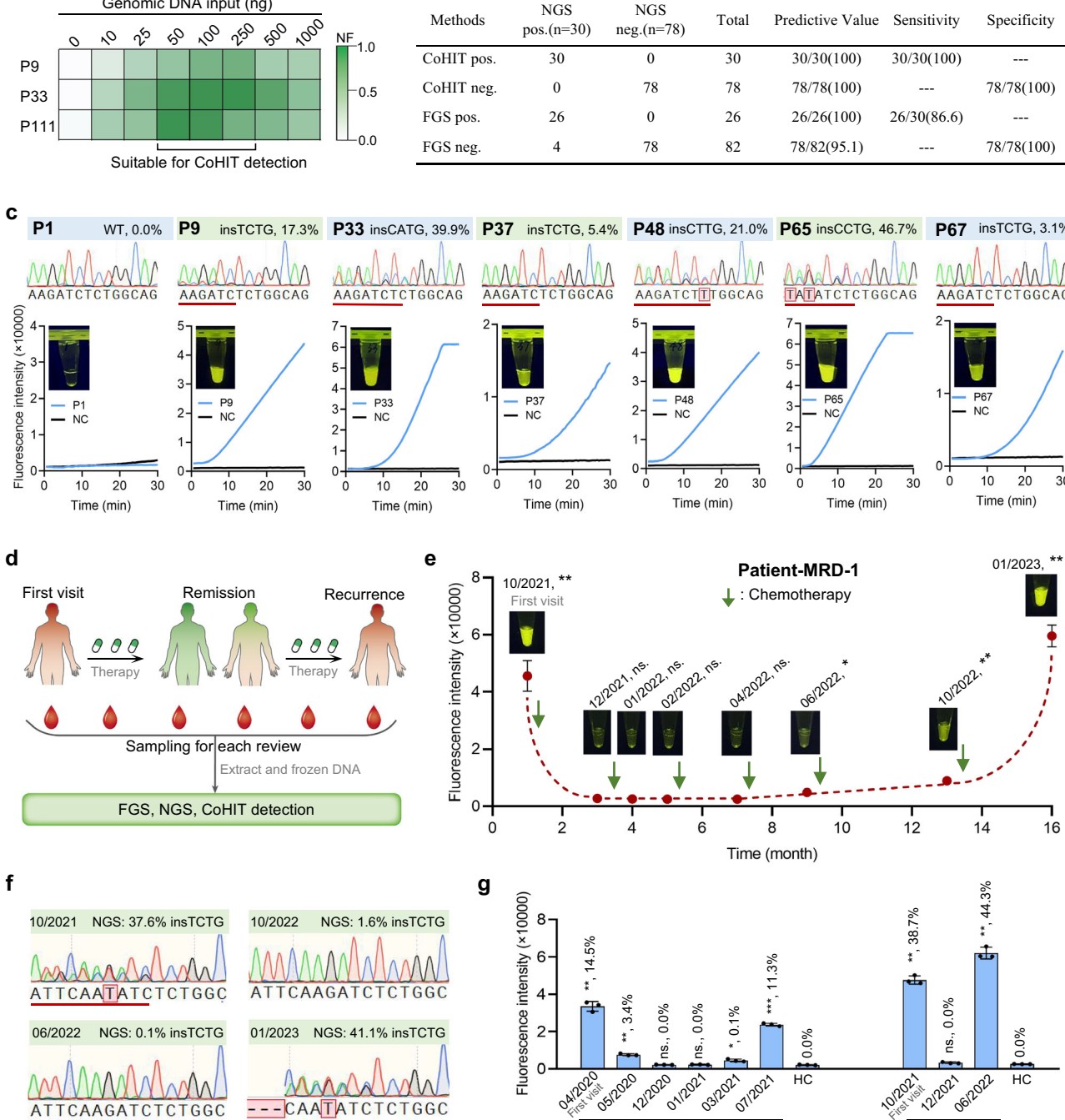

**Fig. 6 | Detection of clinical samples and MRD using the CoHIT system.**
**a** Template input assay of the CoHIT system using blood genomic DNA of three *NPM1*-mutated AML patients, including P9 with 12.3% c.863_864insTCTG, P33 with 39.9% insCATG, and P111 with 44.0% insTCTG. **b** Statistical table of the sensitivity and specificity of the CoHIT method in detecting 108 AML patients compared with FGS and using NGS as a standard reference. **c** Detection results of representative clinical samples (P1, P9, P33, P37, P48, P65, and P67) using the NGS, FGS, and CoHIT methods. The genotypes and mutation rates detected by NGS are shown at the top. FGS peak diagrams are shown in the middle, with the mutated bases underlined. The naked-eye results and time course of CoHIT detection are shown below. **d** Flowchart of the MRD experiment. **e** CoHIT result for Patient-MRD-1. The sampling dates and naked-eye results are shown for each sample. Values and error bars reflect the means and s.d. of three biological replicates. *P* values compared with

healthy WT control (HC) are determined by two-tailed Student's *t*-tests. The asterisks indicate the ranges (**P* < 0.05, ***P* < 0.005, ****P* < 0.0005, ns no significance.). The respective *P* values in left-to-right order are 0.005, 0.4734, 0.8067, 0.8209, 0.9333, 0.019, 0.0117, 0.0015. **f** NGS and FGS results of Patient-MRD-1 samples dated 10/2021, 06/2022, 10/2022, and 01/2023. **g** CoHIT and NGS results for Patient-MRD-2 and 3. Values and error bars reflect the means and s.d. of three biological replicates. *P* values compared with HC are determined by two-tailed Student's *t*-tests. The asterisks indicate the ranges (**P* < 0.05, ***P* < 0.005, ****P* < 0.0005, ns no significance.). The respective *P* values in left-to-right order are 0.0025, 0.0048, 0.6713, 0.2075, 0.0321, 0.0003, 0.0009, 0.3449, 0.0009. The percentages are the insTCTG mutation rates determined by NGS. Source data are provided as a Source Data file.

Then, we used 100 ng of DNA from the 108 AML patients to perform the CoHIT assay, as well as FGS and NGS for comparison. After 30 min of reaction at 39 °C, 30 out of 108 samples exhibited positive fluorescence signals in the CoHIT assay, which was completely consistent with the NGS results. However, FGS missed four positive samples with low mutation rates (P37, 5.4% insTCTG; P67, 3.1% insTCTG; P78, 5.6% insTCTG; P104, 7.5% insTCTG), indicating that CoHIT has higher sensitivity than FGS (100% vs. 86.6%) (Fig. 6b and Supplementary Fig. 15–21). Notably, the CoHIT system successfully detected four patients with other insertion types, including P33 (39.9% insCATG), P48 (21.0% insCTTG), P65 (46.7% insCCTG), and P70 (31.0% insCATG) (Fig. 6c and Supplementary Fig. 19). These results further confirmed the one-for-all capacity of the CoHIT system.

Considering that *NPM1* mutation is a clinically recommended MRD testing item[34], we tested the ability of the CoHIT system to detect residual leukaemia in *NPM1*-mutated AML patients. We utilized cryopreserved bone marrow DNA samples collected from three patients (Patient-MRD-1 - 3) at multiple time points, ranging from initial diagnosis through remission to recurrence during follow-up visits. These samples underwent simultaneous testing using FGS, NGS, and the CoHIT method for comparative analysis (Fig. 6d). The CoHIT result of Patient-MRD-1 showed a robust fluorescence signal in the sample dated 10/2021, indicating a positive *NPM1* mutation at the initial diagnosis, which was corroborated by NGS revealing a 37.6% insTCTG mutation (Fig. 6e, f, Supplementary Fig. 22). After several courses of chemotherapy, the fluorescence signal disappeared in samples dated 12/2021, 01/2022, 02/2022, and 04/2022, indicating complete remission without detectable MRD, all of which were confirmed by NGS as 0.0% mutation. Then, for the sample dated 06/2022, a weak but significant fluorescence signal was observed in the CoHIT reaction and a 0.1% insTCTG mutation was verified by NGS, indicating a detectable MRD and a trend toward recurrence. Several months later, the recurrence samples dated 10/2022 and 01/2023 showed gradually increasing fluorescence signals, which were confirmed by NGS to be 1.6% and 41.1% insTCTG mutations, respectively. Notably, FGS failed to detect the mutated samples dated 06/2022 and 10/2022, demonstrating that the CoHIT system can provide earlier warning of leukaemia recurrence than FGS (Fig. 6e, f and Supplementary Fig. 22). Similar to those of Patient-MRD-1, the CoHIT results of Patient-MRD-2 and Patient-MRD-3 were also completely consistent with the patients' disease progression and NGS results (Fig. 6g and Supplementary Fig. 23 and 24). Given the sensitivity, rapidity, convenience, and cost-effectiveness of the CoHIT method, it holds promise as a point-of-care tool for MRD detection and could prove invaluable in the regular follow-up and efficacy evaluation of cancer patients.

### CoHIT system can combine with lateral flow strips and has multiplexing potential

In addition to employing a fluorescence detector, CoHIT can also be combined with a lateral flow assay (LFA) for more convenient application. To establish the CoHIT-LFA system, the FAM-ssDNA-BHQ1 reporters used in the CoHIT system were replaced with biotin-ssDNA-FAM reporters. When cleaved reporters in the CoHIT reaction product flow through the strip, nanoparticles labelled with anti-FAM antibodies capture the FAMs and lead the cleaved reporters to an anti-anti-FAM antibody-labelled test line, while uncleaved reporters will be intercepted at the streptavidin-labelled control line (Fig. 7a). Given that the reporter concentration has a significant influence on the LFA assay, we first used strips to directly detect different concentrations of uncleaved biotin-ssDNA-FAM reporters dissolved in ddH$_2$O. The results showed that 2.5 nM resulted in incomplete nanoparticle binding, leading to false-positive signals, and concentrations exceeding 5 nM were determined to be appropriate (Fig. 7b). Therefore, in the CoHIT-LFA assay, a 25 μL one-pot CoHIT system (containing 20 nM biotin-ssDNA-FAM reporters) was first incubated at 39 °C for the

reaction, and then 75 μL of ddH$_2$O was added to dilute the reporters to 5 nM. Then, an LFA strip was vertically inserted into the tube, and the result was read after approximately 2 min (Fig. 7c). We explored the CoHIT reaction time for subsequent LFA assay by detecting *NPM1*-mutated patient samples and found that the test line signals increased rapidly at 10, 20, 30, and 60 min and slowly after 90 min, indicating that 90 min is a suitable time (Fig. 7d). Then, we set the CoHIT reaction time to 90 min to test two *NPM1*-WT patients (P1 and P2) and eight *NPM1*-mutated patients (P37, P44, P48, P55, P67, P70, P85, and P108). The results showed that the CoHIT-LFA method successfully distinguished between positive and negative patient samples (Fig. 7e).

CoHIT can also be integrated with microfluidic devices to enable multiplexed gene detection. As illustrated in Fig. 7f, we developed a microfluidic-based method to simultaneously detect seven targets with a single loading step. The microfluidic device has one loading chamber and nine reaction chambers for the NC, Target 1 - 7, and IC (inner control, GAPDH gene) groups, respectively. As a test, the reaction chambers were preinjected with CoHIT systems, and the primer pair and the crRNA were placed in the chamber. After a 30 min reaction at 39 °C, the fluorescence signal on the chip was observed under blue light (Fig. 7f). To show its practicality, we used this method to detect the *NPM1* gene c.863_864 4-bp insertion (Target 1) and six internal tandem duplication (ITD) sites of the *FLT3* gene (Targets 2 - 7), which is the most common mutated gene in AML and has many different ITD sites (Supplementary Fig. 25)[35]. By detecting 1E6 copies of mutated fragment-containing plasmids in 100 ng of WT genomic DNA, the CoHIT chip displayed good specificity for each target (Fig. 7g). Then, we tested its LoD by detecting Target 2 with 10%, 1%, and 0.1% mutation ratios, and the results showed that the fluorescence signal of the 0.1% sample was observable by the naked eye, indicating good sensitivity (Fig. 7h). Finally, six AML patients were detected using CoHIT chips, and five *NPM1*-mutated samples (P5, P9, P48, P52, P85) were successfully identified. Notably, the *FLT3*-ITD mutations of P30 and P85 were also simultaneously identified and verified by sequencing (Fig. 7i). Collectively, these results demonstrated the flexibility and multiplexing potential of the CoHIT assay.

## Discussion

As genomic profiling becomes more prevalent in diagnosing and guiding therapy for various cancers, precision oncology is swiftly transforming the landscape of cancer care[1]. Advances in the discovery of hotspot mutations in cancer-related genes have enabled more precise cancer classification, prognosis, and disease monitoring through genetic testing[36,37]. However, the existence of multiple variants at the same gene site poses a significant challenge to current genotyping methods. For example, there are at least 16 kinds of in-frame deletion variants at residue 747–750 in *EGFR* gene exon 19, accounting for approximately 45% of *EGFR*-mutated non-small cell lung cancer (NSCLC) cases[38]. These variants are similar in sequence, and all can lead to TKIs resistance, therefore, full-coverage detection is necessary to avoid false-negative results.

In this study, we aimed to develop a CRISPR-based method to address the challenge of conveniently detecting multiple same-site variants. The Cas12a orthologues AsCas12a and LbCas12a have been extensively used in CRISPR-based nucleic acid detection assays. Previous studies have reported that AsCas12a exhibits poorer single-base resolution or greater mismatch tolerance than LbCas12a because AsCas12a has less stringent base pairing requirements[30]. In addition, protein engineering may further improve mismatch tolerance, as various engineered Cas12a proteins have been developed to improve specificity and cleavage efficiency and broaden PAM recognition[31,32,39,40]. Here, we engineered a previously unreported AsCas12a protein variant, enAsU-R Cas12a, by combining several

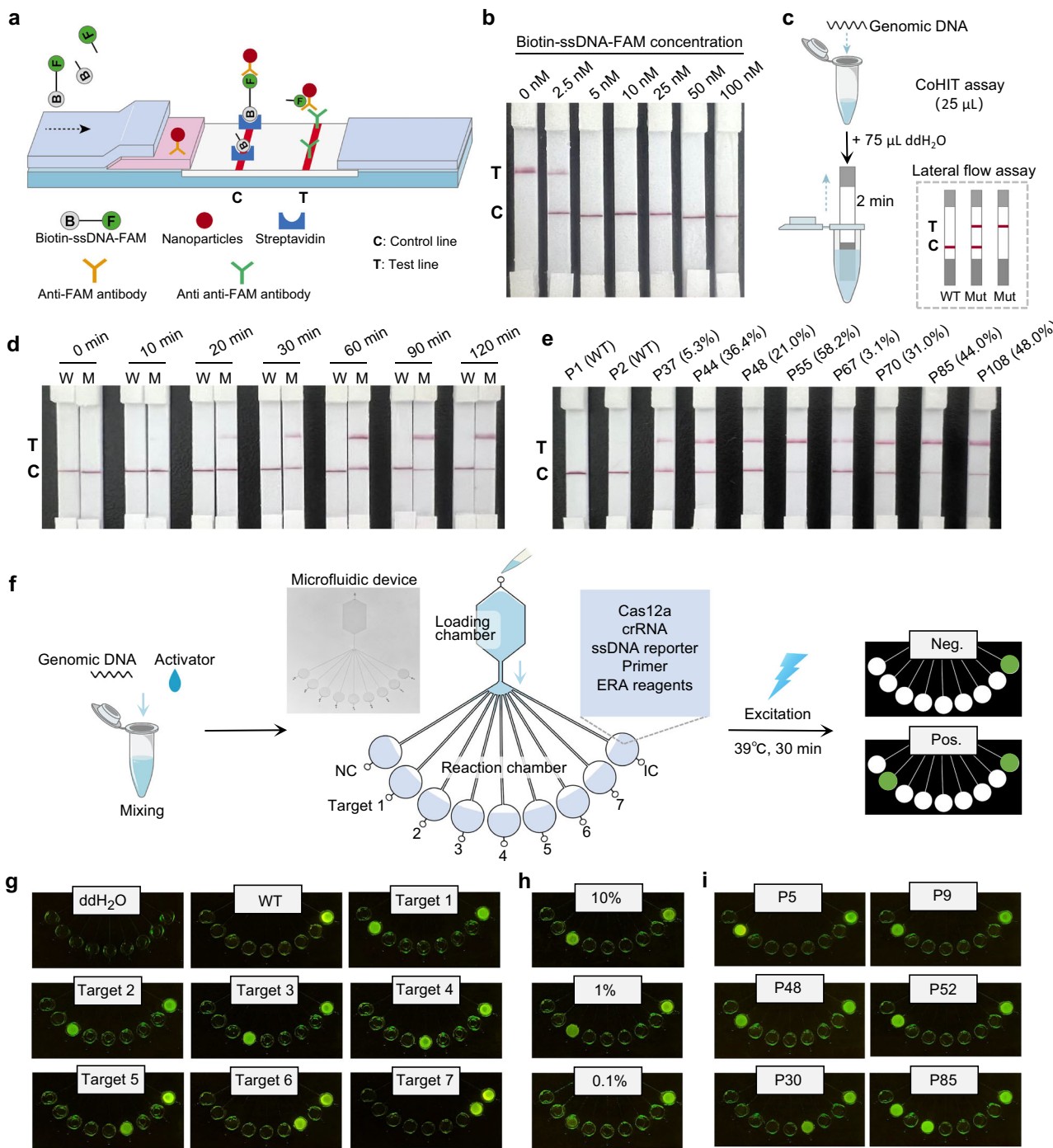

**Fig. 7 | CoHIT-LFA system and microfluidic chip-based CoHIT detection.**
**a** Detection principle of the LFA strip for the CoHIT system. **b** Biotin-ssDNA-FAM
concentration optimization for the LFA assay. **c** Flowchart of the CoHIT-LFA assay.
**d** Optimization of the CoHIT reaction time for the CoHIT-LFA assay. W, WT sample.
M, mutant sample. **e** Detection of the *NPM1* gene c.863_864 4-bp insertion in ten
AML patient samples. The percentages are mutation rates. **f** Flowchart of the

microfluidic chip-based multiplexing CoHIT detection. IC, inner control.
**g** Detection of 1E6 copies of mutated fragments-containing plasmid in 100 ng WT
genomic DNA. **h** Detection of Target 2 at frequencies of 10%, 1%, and 0.1%.
**i** Detection in clinical samples from six AML patients. Source data are provided as a
Source Data file.

reported functional amino acid substitutions, including E174R,
M537R, S542R, K548R, K607R, and F870L. Surprisingly, enAsU-R
Cas12a has not only improved mismatch tolerance but also a dis-
tinctive and wide PAM preference (NNCV and TTTV) (Fig. 3a), which
makes it easy to design and screen crRNAs for almost any mutation
site in the human genome. Notably, the M537R and F870L substitu-
tions did not significantly impact the PAM preference of enAsU
compared with that of enAs (Supplementary Fig. 2a) because the

M537R and F870L substitutions were screened mainly for improving
gene editing efficiency, not for expanding the PAM scope[32]. However,
the addition of K607R, a key substitution of the AsCas12a-RR variant
related to TYCV PAM preference[33], significantly changed the PAM
scope of enAsU-R compared with that of enAsU. Considering that a
loose PAM preference may increase the risk of off-target effects, we
tested 60 predicted off-target sites of the *NPM1*, *KIT*, *BRAF*, and *EGFR*
targets using an enAsU-R in vitro cleavage assay. The results verified

the low off-target effect of enAsU-R on the human genomic DNA sequence (Supplementary Fig. 26 and Supplementary Data 3 and 4). Moreover, the specific ERA amplification process further ensures the specificity of the CoHIT system. Interestingly, in the optimization process of the CoHIT system, we observed a negative impact of the NEBuffers, which are commonly used in CRISPR/Cas12a in vitro cleavage reactions[41,42]. Our study suggested that this negative impact is likely due to the effect of NEBuffers on the concentration of salt ions in the one-pot system. Previous research has also demonstrated that the concentration of salt ions can affect the efficiency and specificity of Cas12a-based detection assays[43]. Furthermore, in our investigation of genomic DNA input, we observed that the output fluorescence signal initially increased but then decreased beyond a certain point (Fig. 6a). This phenomenon could be attributed to low genomic DNA input falling below the detection limit, while high input leads to system crowding, which is unfavourable for the reaction.

In this study, we also investigated the MRD detection potential of the CoHIT system and combined it with LFA strips and microfluidic devices to demonstrate its flexibility and multiplexing potential for clinical application. It should be noted that although our LoD for *NPM1* mutation is 0.01%, MRD level below 0.01% may still be significant for several mutated oncogenic genes, such as *NPM1*, *FLT3*, and *BRAF*. The standard definition of MRD negativity is dependent on the type of MRD technology and target[44]. Using microfluidic chip-based multiplexing detection, we successfully detected seven mutation loci simultaneously, which is useful for patients carrying multiple mutations. However, its utility in upfront mutational profiling may be limited, making it more suitable for regular follow-up, efficacy evaluation, and MRD monitoring. To sum up, our CoHIT method is not only sensitive, specific, rapid and convenient but also cost-effective (Supplementary Table 1). It offers great advantages in point-of-care testing applications, including cancer classification, efficacy evaluation, resistance mutation warning, and MRD detection[45]. We believe that this system holds great potential for advancing precision cancer medicine.

## Methods

### Ethics and inclusion statement
All authors met the authorship inclusion criteria for Nature Portfolio journals. The role of each author was discussed and agreed upon prior to the completion of the project.

Blood samples in this study were collected from the Haematology Department of Zhongnan Hospital of Wuhan University under an approved Institutional Review Board protocol (Scientific Ethics Approval No. 2017064). For all human patient samples, informed written consent was obtained prior to donation. Each participant received nutritional products as compensation for their participation. All patient samples in this study were deidentified and patient demographics are not available.

### Patient samples
A total of 125 patient samples were collected from the Haematology Department of Zhongnan Hospital of Wuhan University. Blood genomic DNA from these patients was extracted using a genomic DNA extraction kit (Tiangen, Beijing, China). Briefly, 200 μL of blood was incubated with 200 μL of Buffer GB and 20 μL of proteinase K at 56 °C for 10 min. After cooling at room temperature for 3 min, 300 μL of Buffer BD was added and fully mixed by inversion. Next, the mixture was filtered through an adsorption column at 13,800 g for 30 s. Then, the adsorbed DNA in the column was sequentially washed with 500 μL of Buffer GDB and 600 μL of Buffer PWB at 13,800 g for 30 s. After the residue was dried at room temperature for 2 min, the DNA was eluted with 50 μL of ddH$_2$O and quantified with a Nanodrop2000 (Thermo Fisher, MA, USA).

### DNA fragments and plasmid preparation
The DNA fragments used in the mismatch tolerance assay and 2/1-bp small indel detection were directly synthetized by GenScript (Nanjing, China), and the sequences are listed in Supplementary Data 5. The plasmids carrying the *NPM1* gene WT and c.863_864insTCTG DNA were constructed by TA cloning using the PCR-amplified (Primers P1 and P2) DNA fragments of a patient sample. The plasmids carrying the *NPM1* gene c.863_864insCATG, CCTG, CCAG, CCGG, CTTG, TATG, TCGG, TAAG, CAGG, and CAGA DNA were constructed by cycling PCR using the c.863_864insTCTG plasmid as a template and primers P3 - 22. The PCR primer sequences are listed in Supplementary Data 6.

### Cas12a protein expression and purification
Plasmids expressing Cas12a proteins were transformed into BL21(DE3) competent *E. coli* cells. Colonies were picked and incubated in LB medium at 37 °C in a shaker for 14 - 16 h. The culture was then seeded in 1 L of LB for scale-up until the OD600 reached 0.6 - 0.8. Protein expression was induced with 1 mM IPTG at 16 °C for 18 h. Cells were harvested by centrifugation and then resuspended in lysis buffer (25 mM Tris-HCl, pH 8.0, 500 mM NaCl, 10% (v/v) glycerol, 0.5 mM PMSF), the suspended cell mixture was disrupted by sonication and centrifuged at 48,000 g for 30 min at 4 °C. The supernatant was filtered through a 0.22 μm filter (Millipore, MA, USA) and incubated with Ni-NTA resin at 4 °C for 2 h. After the resin was washed with washing buffer (25 mM Tris-HCl, pH 8.0, 500 mM NaCl, 20 mM imidazole, pH 8.0), the Cas12a proteins were eluted with elution buffer (25 mM Tris-HCl, pH 8.0, 500 mM NaCl, 250 mM imidazole, pH 8.0) and concentrated with a 100 kDa centrifugal filter unit (Millipore, MA, USA), after which the buffer was changed to storage Buffer (50 mM Tris-HCl, 200 mM NaCl, 1 mM DTT, 20% glycerol, pH 8.0). The purity of the proteins was analysed by SDS- PAGE and Coomassie blue staining, and the protein concentration was quantified using a BCA Protein Assay Kit (Thermo Fisher, MA, USA). The amino acid sequences of the Cas12a proteins in this study are listed in Supplementary Data 7.

### CRISPR/Cas12a in vitro cleavage assay
For direct detection of the DNA fragments shown in Fig. 2 and Fig. 3, the Cas12a-mediated in vitro cleavage reaction was performed in a 20 μL system containing 200 ng of Cas12a, 1 pmol of crRNA, 2 μL of 10× NEBuffer3.1, 0.1 μL of RNase inhibitor (Novoprotein, Shanghai, China), 25 pmol of the FAM-ssDNA-BHQ1 reporter (Genewiz, Suzhou, China), an appropriate amount of target DNA fragments, and nuclease-free water. After incubation at 37°C, the cleavage products were visualized by fluorescence under a 485 nm blue light (Sangon, Shanghai, China). Fluorescence kinetics were monitored using a portable fluorescence detector (GenDx Biotech, Suzhou, China) with excitation at 485 nm and emission at 520 nm. The DNA copy number was calculated with NEBioCalculator (version 1.15.5). The crRNAs were designed according to the target sequences and synthesized by GenScript (Nanjing, China). The nucleotide sequences of all the crRNAs are listed in Supplementary Data 8.

### In vitro PAM identification assay
A PAM library containing 256 kinds of linear dsDNA substrates, consisting of a 20 nt protospacer and 4 randomized nucleotides upstream of the protospacer, was synthesized, and all 256 substrates were mixed at the same concentration. For the in vitro PAM identification assay, each reaction with a total volume of 10 μL contained 25 ng of PAM library, 400 ng of Cas12a protein, 0.1 μL of RNase inhibitor, 10 pmol of crRNA, 1 μL of 10×NEBuffer3.1, and nuclease-free water. Cleavage reactions were incubated at 37 °C, and cleavage products were collected at 0, 5, 10,

30, 60 and 120 min. After amplification with barcoded primers and purification using an AxyPrep PCR Clean-up Kit, the samples were subjected to NGS to quantify the PAM regions. The unmodified library amplicon was used as a negative control to determine the initial PAM representation in the libraries.

## ERA isothermal amplification

For the ERA primer screen assay, ERA amplification was conducted using the GenDx ERA Kit (GenDx Biotech, Suzhou, China) according to the manufacturer's instructions. Briefly, the amplification was conducted in a 50 μL reaction system containing 20 μL of ERA reaction buffer, 10 μL of ERA basic buffer, 25 pmol of forward primer, 25 pmol of reverse primer, 2 μL of activator, DNA template and nuclease-free water. The reaction was performed at 37 °C for 20 min, after which 5 μL of the amplification product was transferred to the CRISPR/Cas12a in vitro cleavage system for target detection. ERA primer sequences are listed in Supplementary Data 9.

## ERA-Cas12a one-pot assay

The ERA-CRISPR one-pot assay integrates ERA isothermal amplification and CRISPR/Cas12a in vitro cleavage reaction. In Fig. 4, the one-pot system was initially set up as a 25 μL system containing 10 μL of ERA reaction buffer, 5 μL of ERA basic buffer, 12.5 pmol of forward primer, 12.5 pmol of reverse primer, 2.5 μL of 10× NEBuffer3.1, 0.1 μL of RNase inhibitor, 200 ng (8 ng/μL) of Cas12a protein, 1 pmol (40 fmol/μL) of crRNA, 25 pmol (1 pmol/μL) of the FAM-ssDNA-BHQ1 reporter, 1 μL of the ERA activator, DNA template, and nuclease-free water. The initial reaction condition for fluorescence signal detection was 37 °C for 30 min. After optimization, NEBuffer3.1 was omitted from the CoHIT system, the crRNA concentration was adjusted to 20 fmol/μL, and the reaction temperature was set to 39 °C.

## First-generation sequencing and next-generation sequencing

Approximately 300 ng of amplified DNA fragments per sample were subjected to FGS after purification using an AxyPrep PCR Clean-up Kit (Axygen, CA, USA) and quantified by a Nanodrop2000. For NGS, samples were PCR-amplified with different barcoded primers. Then, the PCR products were purified, quantified, and mixed with equal amounts for NGS by the Illumina NextSeq 500 (2 × 150) platform at the CAS-MPG Partner Institute for Computational Biology Omics Core, Shanghai, China. The primers used for NGS are listed in Supplementary Data 10.

## Lateral flow strip assay (LFA)

The special LFA strip for CRISPR was produced by GenDx Biotech (Suzhou, China). The CoHIT system for LFA used biotin-ssDNA-FAM instead of FAM-ssDNA-BHQ1 while keeping the other components unchanged, including 10 μL of ERA reaction buffer, 5 μL of ERA basic buffer, 12.5 pmol of forward primer, 12.5 pmol of reverse primer, 0.1 μL of RNase inhibitor, 200 ng of Cas12a protein, 0.5 pmol of crRNA, 1 μL of ERA activator, DNA template, and nuclease-free water for a 25 μL final volume. The optimized concentration of biotin-ssDNA-FAM in the CoHIT system was 20 nM. After the CoHIT reaction, 75 μL of ddH$_2$O was added to the 25 μL of the product, and then the LFA strip was inserted into the tube vertically. After approximately 2 min, the results were readable by the naked eye.

## Microfluidic chip assay

The microfluidic chip was fabricated by a standard soft-lithography technique (CChip Scientific Instrument, Suzhou, China)[46]. First, the reaction chambers were prefilled with all components required for the CoHIT reaction, except for the DNA template and activator. For each reaction chamber, the crRNA and primer pair used differed depending on the target. For the NC chamber, the crRNA and primers were replaced with ddH$_2$O.

For the IC chamber, crRNA and primers were designed for a conserved GAPDH gene site. Then, 9 μL of genomic DNA and 9 μL of activator were mixed and pushed into the loading chamber. After the genoimc DNA and activator were evenly pushed into the nine reaction chambers, the chip was incubated at 39 °C for 30 min and imaged by an LED transilluminator (Sangon, China) to obtain the fluorescence signals. The crRNAs and ERA primers used in the microfluidic chip assay are listed in Supplementary Data 8 and 9.

## Statistics and Reproducibility

All experiments were performed in triplicate. Statistical analyses were carried out with GraphPad Prism 8.0 (GraphPad Software, CA, USA) and SPSS 27 (IBM, Armonk, NY, USA). Students' t-test (two-tailed) were used to determine the statistical significance of differences in levels among experimental groups. Quantitative data are expressed as the mean value ± standard error. All the statistical details of the experiments can be found in the figure legends. No statistical method was used to predetermine the sample size. No data were excluded from any of the experiments.

## Reporting summary

Further information on research design is available in the Nature Portfolio Reporting Summary linked to this article.

## Data availability

The authors declare that all data of this study are available within the article and its supplementary files. NGS data has been deposited to the NCBI-SRA repository under BioProject accession number PRJNA1029775. The reference human genome assembly GRCh38/hg38 used for reads mapping is an openly accessible resource under accession number GCF_000001405.40. The crystal structure for AsCas12a, which was used in this work, is available from the Protein Data Bank under ID 5B43. Source data are provided with this paper.

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

## Acknowledgements

We thank members of Xingxu Huang's lab and Fuling Zhou's lab for helpful discussions. This study was supported by Guangdong Basic and Applied Basic Research Foundation (2022A1515110852 to Y.L.), National Natural Science Foundation of China (82300264 to Y.L., 81770179 to F.Z. and 82370176 to F.Z.), China Postdoctoral Science Foundation (2023M742696 to Y.L.), and Shenzhen Outstanding Talent Program (Grant No. 202102 to K.L.).

## Author contributions

Conceptualization, Y.L., Xinyi L., X.H., K.L., X.W. and F.Z.; Methodology, Y.L., Xinyi L. and X.W.; Investigation and validation, Y.L., Xinyi L., D.W., L.D., X.X., L.L., S.W., J.W., Xiaoyan L., W.S., W.T. and Y.W.; Resources, Y.L., X.H., K.L., X.W. and F.Z.; Data analysis, Y.L. and S.H.; Original draft preparation, Y.L. and Xinyi L.; Article revision, X.H., K.L., X.W. and F.Z.

## Competing interests

The authors declare no competing interests.
