## [Peer Review File · Nature Communications]

Reviewers' comments:

Reviewer #1 (Remarks to the Author):

Key Results

Liu et al describe in this work a CRISPR-based diagnostic detection platform capable of detecting multiple disease variants in a single sample using just a single Cas12a crRNA. This work is an extension of previously described Cas13a and Cas12a-based detection platforms that gained notoriety for the detection of COVID-19 and other respiratory pathogens. A key finding from this study is that the “CoHIT” technique is a rapid, sensitive, and cost-effective tool for cancer diagnosis and simultaneous detection of multiple variants at a single disease locus. The application of this tool is both novel and relevant to real challenges in cancer diagnostics and represents a novel application for Cas12a/Cas13a detection systems that have not been able to supplant qPCR and antigen tests for RNA pathogen detection. The potential application of this work to the field of minimal residual disease detection (MRD) is of high relevance and the authors could consider including this in the discussion to elevate the relevance and significance of their findings. I think the notion of providing point of care cancer diagnostics is quite provocative. Overall, this is generally high quality work and my only reservations are towards the presentation of the protein engineering aspects and their relative novelty/significance, which I discuss later in this review.

Validity

Generally the data are robust and valid – the authors do a nice job of presenting the qualitative, point of care fluorescence data while simultaneously showing real time fluorescence data to further bolster claims about the technique. The authors also nicely describe multiple cancer disease applications and thoroughly explore these in great detail. I have two main concerns regarding the general validity of the work. First, with NGS, there is less concern about false positives/negatives with respect to analyzing cancer disease variants. Presumably with CoHIT, since you are amplifying only the region of interest, this minimizes the potential for off-target cleavage to occur and give a false positive result. I’m nevertheless concerned about combining Cas12a mutants with massively increased intrinsic activity and reduced PAM specificity (thus higher off-target effects) and what that would mean for off-target cleavage, particularly with ERA amplicons that are highly repetitive. I think an exploration of the specificity of these combined nucleases (either in the context of CoHIT or otherwise) is warranted – particularly given the consequences of a false positive or negative result. Secondly (somewhat related), I’m concerned about the data supporting the need for a further engineered nuclease. There are two figures dedicated towards combining previously described enhanced activity and PAM specificity altering mutations and nevertheless the data in Figure 4 would suggest that perhaps these further engineered variants, which haven’t been characterized to the extent of enAsCas12a or AsCas12a Ultra, are in fact not needed. Additional data supporting and justifying the need for these further engineered and combined variants is needed.

Significance

I think the application of the Cas12a/Cas13a diagnostics detection technology to cancer disease typing is relevant and significant. I typically have only seen this technology being used to detect presence/absence of respiratory pathogens. However, my main concern about significance is due to the protein engineering aspect, the fraction of the manuscript devoted to it, and whether it truly represents a novel variant or whether this piece is best relegated to the supplemental. As mentioned, while the clinical application is noteworthy, it is not obvious to me that this work absent the protein engineering piece is of high significance.

Ultimately, the previously described AsCas12a variants enAsCas12a and AsCas12a Ultra were the result of an exhaustive set of directed evolution and rational mutagenesis investigations to uncover protein mutants of high intrinsic potency and reduced PAM specificity. In the work presented here, the mutants carried forward in CoHIT were simply various combinations of mutants already described in the literature and the resulting characterization thereof. The authors use words/phrases like “screening, protein engineering, and novel” to describe these variants and devote 2 figures to the topic which seems disingenuous and not truly novel. Of further concern, it is not convincing to me that the resulting mutants are critical for the CoHIT technique to be successful, and I don't believe the data yet support that conclusion either. I recommend moving the protein engineering figures to the supplemental and revising the paper to instead focus on the application work, which is novel and significant. I acknowledge that a protein engineering or directed evolution study to isolate mutants specifically adapted for the CoHIT technique is also both interesting and provocative.

Data and methodology

No additional concerns around the data as presented or methodology.

Analytical approach

No general concerns about the analytical approach though the authors will need to comply with the Nature Communications statistics policies in more finalized drafts (ie - N=3, etc).

Suggested improvements

I have indicated some general concerns about the protein engineering content and whether the combined mutants described/characterized in this work are novel and are in fact necessary for the CoHIT approach. I think it would be useful to further characterize these mutants for off-target cleavage effects and discuss what that might mean for false positive detection using the CoHIT approach. I would suggest doing this with any of the described live cell methods (ie – GuideSEQ) or cell free methods available in the literature. I also think that the authors should provide a convincing demonstration that the described mutants are truly necessary for this methodology to be successfully employed. Finally, I would suggest at minimum greatly reducing the protein engineering figures or relegating them to the supplemental entirely depending on what is discovered in follow-up experiments. I think with some revision this could be a really impactful paper with real significance in clinical diagnostics.

Reviewer #2 (Remarks to the Author):

In the submitted manuscript, the authors reported the development of a CRISPR-based diagnostic assay for the detection of cancer mutations. The idea is interesting and relies on the fact that AsCas12a variants can tolerate some mismatches at specific positions along their binding site to detect the presence of indels. Usually, indels are detected by loss of signal as it is difficult to design one set of primers or probes against all possible combinations of indels and thus it is easier to design against the wildtype sequence instead. But in the authors' assay, indels are detected by gain of signal, which is more unusual.

Nevertheless, while the work is interesting from an academic point of view, I am not entirely convinced of its real-world applicability. A major issue is that detection of cancer mutations is not extremely time critical (unlike infectious diseases like the COVID-19 pandemic). Generally, it will not matter if a cancer is diagnosed in 30 minutes or a few days. In the introduction, the authors tried to give hyperleukocytosis as an example. But the pathophysiology of hyperleukocytosis is not well understood, so it is impossible to create a CRISPR assay for it. Hence, I believe that clinicians would prefer to rely on methods like next generation sequencing (NGS) or real-time PCR. Moreover, CRISPR-based assays are challenging to multiplex and thus cannot serve as a general screening tool for multiple types of mutation. Consequently, the authors might want to consider alternative applications of their method.

Major comments:

1) Can the method work for small (1bp or 2bp) indels? Larger indels (like 4bp) are easier to detect and distinguish from wildtype.

2) For the NPM1 target, why don't the authors use or test a crRNA that hybridises to the TCTG insertion perfectly since it is most common (fig 4a)? I would imagine testing a crRNA that recognises the most common insertion (and then checking whether it would work against other insertions) to be a natural thing to do.

3) In the last results section, I find it odd that the authors titrated input template amounts using a single patient sample and then generalised. Fluorescence intensity would depend on gene expression level and mutation load. Can the authors test a few different patient samples with varying gene expression levels and mutation loads?

4) As mentioned above, cancer mutation detection might not be the best use case. Can the authors show another application that genuinely requires fast turnaround time? I am not convinced that identification of a 4bp insertion in NPM1 for AML requires a 30min turnaround time. There should be rigorous benchmarking against NGS and qPCR. Notably, targeted amplicon sequencing on Illumina can readily achieve 0.1% mutation detection rate and can further multiplex to cover many oncogenes and tumor suppressors. In addition, the turnover time is even faster for nanopore sequencing, which can also achieve 0.1% VAF (variant allele frequency) detection.

Minor comments:

5) Please indicate that “OS” stands for original sequence in the main text

6) CoHIT does not detect all variants (definitely not single nucleotide substitutions). Hence, the title and abstract are misleading, and the authors should make clear that their assay is for indels only.

7) Please indicate that “FGS” stands for first generation sequencing such as Sanger sequencing in the main text. Where does the claim of 10% rate for FGS come from? It is still possible to quantify 5-10% from Sanger sequencing chromatograms.

8) The persistent claim of one crRNA to identify numerous gene variants (indels) is misleading. The authors’ assay can effectively only check one mutation site at a time.

9) There is a typo on line 241: “two important variants: c. 1459_1473del and c.1459_1473del” (they are the same).

Reviewer #3 (Remarks to the Author):

In this manuscript the authors use Cas12a to detect mutations in genes where there are multiple possible different nucleotide sequences, as in the case of indels that define NPM1 mutations. The outlined approach could be broadly applicable across various diseases defined by multiple mutation possibilities at the same site, and they showed the ability to use their system for other

mutations as well. The approaches seem robust however the main concern is the novelty and likelihood this approach will truly find a place in clinical diagnostics. The authors approach ties together several approaches that were recently novel but in and of itself the approach is not.

There are multiple examples of using Cas12a approaches to detect genetic variants. This flourished in the COVID literature:

- Ding X, Yin K, Li Z, Lalla RV, Ballesteros E, Sfeir MM, Liu C. Ultrasensitive and visual detection of SARS-CoV-2 using all-in-one dual CRISPR-Cas12a assay. *Nat Commun.* 2020 Sep 18;11(1):4711. doi: 10.1038/s41467-020-18575-6. PMID: 32948757; PMCID: PMC7501862.
- Liang Y, Zou L, Lin H, Li B, Zhao J, Wang H, Sun J, Chen J, Mo Y, Yang X, Deng X, Tang S. Detection of Major SARS-CoV-2 Variants of Concern in Clinical Samples via CRISPR-Cas12a-Mediated Mutation-Specific Assay. *ACS Synth Biol.* 2022 May 20;11(5):1811-1823. doi: 10.1021/acssynbio.1c00643. Epub 2022 Apr 28. PMID: 35481381.
- Fasching CL, Servellita V, McKay B, Nagesh V, Broughton JP, Sotomayor-Gonzalez A, Wang B, Brazer N, Reyes K, Streithorst J, Deraney RN, Stanfield E, Hendriks CG, Fung B, Miller S, Ching J, Chen JS, Chiu CY. COVID-19 Variant Detection with a High-Fidelity CRISPR-Cas12 Enzyme. *J Clin Microbiol.* 2022 Jul 20;60(7):e0026122. doi: 10.1128/jcm.00261-22. Epub 2022 Jun 29. PMID: 35766492; PMCID: PMC9297821.

Cas12a-based methods for variant detection have further expanded into cancers as well, with this article describing a rapid, highly sensitive and specific, Cas12a-based method for detection of FLT3 D835Y/H/V/F mutations:

- Liu Y, Chen Y, Dang L, et al. EasyCatch, a convenient, sensitive and specific CRISPR detection system for cancer gene mutations. *Mol Cancer.* 2021;20(1):157. Published 2021 Dec 2. doi:10.1186/s12943-021-01456-x

Strengths:

- This work is potentially relevant to NPM1 mutation identification. They picked a good disease model for their test. This problem is that it is not so clear that the approach is drastically better than current mutation identification approaches.
- The fact that they engineered Cas12a mutants with more permissive PAM recognition to allow for better detection of mutants is a strength. They did a good job creating Cas12a mutations that will identify most of the major NPM1 mutations. That said, other groups have worked on developing Cas9 and Cas12a mutants that allow for more permissive PAMs for various functions. But the fact this group created several mutants and screened them to improve for tolerance of mismatches is well done.

Weaknesses:

- Style and writing: Significant editing is needed.
- Introduction/framing of their work: Their introduction to why detection of mutations is important, while brief, needs work. One of the sentences is: “Moreover, early detection of low-frequency mutations can help prevent cancer in patients.” I don’t think this is entirely true of current practice, though maybe in the future with cell free DNA. They do not offer an example or a citation for this either. The other background importance for mutation detection that they give is that it helps clinicians diagnose, select treatment, and monitor for recurrence. They cite a review article and the NCCN guidelines, which is pretty vague.
- They refer to sensitivity to mean two different things. They mean it both in terms of a novel statistical test to rule out a mutation and then also the ability to detect low levels of a mutation. They should more clearly differentiate between these.
- In some of the figures and figure legends, it is unclear how much replication they did, as they only have solid bars and the figure legends don’t indicate number of replicates. It would be helpful to get a sense of the variability inherent in their assays (Figure 4E, for example). Also, clarifying what statistical comparisons between groups they were doing (and why) would be helpful.

All said much more validation for potential use as a clinical diagnostic would be very helpful since the approach is not novel.

Reviewer #4 (Remarks to the Author):

Liu et al report a novel, CRISPR-CAS12a based assay for mutation detection with the advantage of a good mismatch tolerance. Through a series of optimization, they could show this assay is sensitive and specific, and surpass first generation sequencing test, and argue that this assay may be applied to clinical samples. The authors should be congratulated for having done such an excellent technical development work! The assay appears technically sound. However, evidence of clinical utility is insufficient.

An clinical assay can be used at diagnosis and/or disease monitoring. An ideal assay should be fast, sensitive/specific and/or cost-effective.

1) Although this Co-HIT assay is faster with a readout probably in hours, and it is more sensitive than first generation sequencing (which has been gradually phased out in clinical practice), I doubt it will surpass NGS as a diagnostic test. As they showed in this study, NGS reliably detected all

NPM1 mutations. Cancer patients often carry multiple mutations (3-8 mutations in AML, 10-30 mutations in solid tumors), and NGS, either whole exome or targeted panel, has the benefit of providing all the information on numerous mutations at once (although often with weeks delay but this will certainly be shortened as bioinformatic pipelines improve). The authors haven't shown a multiplex approach of CO-HIT for detecting multiple mutations simultaneously with reasonable specificity and sensitivity. Even they could, the requirement of knowing the mutation site for assay design would preclude the detection of unknown mutations, making it impossible for screening at diagnosis.

2). The authors showed that CO-HIT can detect as low as 0.01% mutation burden, suggesting its potential utility in minimal residual disease detection. However, the authors used spiked in plasmid DNA but not genomic DNA. Therefore, its detection limit in real clinical practice is unproven. Furthermore, currently ddPCR for NPM1 mutation can easily reach 0.001% in clinical samples, far better than CO-HIT. The authors should be very cautious when stating the sensitivity based on plasmid spike in experiments.

3) One potential application in clinical practice could be a rapid assay for critical clinical actionable mutation detection such as NPM1, FLT3, BRAF, EGFR, etc. For that, the authors need to show it will work as a multiplex assay. With rapid improvement in NGS reporting, some laboratories have developed a short NGS panel including these mutations and can report results in 3-5 days. Therefore, unless this CO-HIT assay clearly edges NGS, its clinical utility even in this narrow space will remain questionable.

Reviewers' comments:

Reviewer #1:

Remarks to the Author:

Key Results

Liu et al describe in this work a CRISPR-based diagnostic detection platform capable of detecting multiple disease variants in a single sample using just a single Cas12a crRNA. This work is an extension of previously described Cas13a and Cas12a-based detection platforms that gained notoriety for the detection of COVID-19 and other respiratory pathogens. A key finding from this study is that the “CoHIT” technique is a rapid, sensitive, and cost-effective tool for cancer diagnosis and simultaneous detection of multiple variants at a single disease locus. The application of this tool is both novel and relevant to real challenges in cancer diagnostics and represents a novel application for Cas12a/Cas13a detection systems that have not been able to supplant qPCR and antigen tests for RNA pathogen detection. The potential application of this work to the field of minimal residual disease detection (MRD) is of high relevance and the authors could consider including this in the discussion to elevate the relevance and significance of their findings. I think the notion of providing point of care cancer diagnostics is quite provocative. Overall, this is generally high-quality work and my only reservations are towards the presentation of the protein engineering aspects and their relative novelty/significance, which I discuss later in this review.

Validity

Generally the data are robust and valid – the authors do a nice job of presenting the qualitative, point of care fluorescence data while simultaneously showing real time fluorescence data to further bolster claims about the technique. The authors also nicely describe multiple cancer disease applications and thoroughly explore these in great detail. I have two main concerns regarding the general validity of the work. First, with NGS, there is less concern about false positives/negatives with respect to analyzing cancer disease variants. Presumably with CoHIT, since you are amplifying only the region of interest, this minimizes the potential for off-target cleavage to occur and give a false positive result. I’m nevertheless concerned about combining Cas12a mutants with massively increased intrinsic activity and reduced PAM specificity (thus higher off-target effects) and what that would mean for off-target cleavage, particularly with ERA amplicons that are highly repetitive. I think an exploration of the specificity of these combined nucleases (either in the context of CoHIT or otherwise) is warranted – particularly given the consequences of a false positive or negative result. Secondly (somewhat related), I’m concerned about the data supporting the need for a further engineered nuclease. There are two figures dedicated towards combining previously described enhanced activity and PAM specificity altering mutations and nevertheless the data in Figure 4 would suggest that perhaps these further engineered variants, which haven’t been characterized to the extent of enAsCas12a or AsCas12a Ultra, are in fact not needed. Additional data supporting and justifying the need for these further engineered and combined variants is needed.

Significance

I think the application of the Cas12a/Cas13a diagnostics detection technology to cancer disease typing is relevant and significant. I typically have only seen this technology being used to detect presence/absence of respiratory pathogens. However, my main concern about significance is due to the protein engineering aspect, the fraction of the manuscript devoted to it, and whether it truly represents a novel variant or whether this piece is best relegated to the supplemental. As mentioned, while the clinical application is noteworthy, it is not obvious to me that this work absent the protein engineering piece is of high significance.

Ultimately, the previously described AsCas12a variants enAsCas12a and AsCas12a Ultra were the result of an exhaustive set of directed evolution and rational mutagenesis investigations to uncover protein mutants of high intrinsic potency and reduced PAM specificity. In the work presented here, the mutants carried forward in CoHIT were simply various combinations of mutants already described in the literature and the resulting characterization thereof. The authors use words/phrases like “screening, protein engineering, and novel” to describe these variants and devote 2 figures to the topic which seems disingenuous and not truly novel. Of further concern, it is not convincing to me that the resulting mutants are critical for the CoHIT

technique to be successful, and I don't believe the data yet support that conclusion either. I recommend moving the protein engineering figures to the supplemental and revising the paper to instead focus on the application work, which is novel and significant. I acknowledge that a protein engineering or directed evolution study to isolate mutants specifically adapted for the CoHIT technique is also both interesting and provocative.

Data and methodology

No additional concerns around the data as presented or methodology.

Analytical approach

No general concerns about the analytical approach though the authors will need to comply with the Nature Communications statistics policies in more finalized drafts (ie - N=3, etc).

Suggested improvements

I have indicated some general concerns about the protein engineering content and whether the combined mutants described/characterized in this work are novel and are in fact necessary for the CoHIT approach. I think it would be useful to further characterize these mutants for off-target cleavage effects and discuss what that might mean for false positive detection using the CoHIT approach. I would suggest doing this with any of the described live cell methods (ie – GuideSEQ) or cell free methods available in the literature. I also think that the authors should provide a convincing demonstration that the described mutants are truly necessary for this methodology to be successfully employed. Finally, I would suggest at minimum greatly reducing the protein engineering figures or relegating them to the supplemental entirely depending on what is discovered in follow-up experiments. I think with some revision this could be a really impactful paper with real significance in clinical diagnostics.

Reviewer #2 :

Remarks to the Author:

In the submitted manuscript, the authors reported the development of a CRISPR-based diagnostic assay for the detection of cancer mutations. The idea is interesting and relies on the fact that AsCas12a variants can tolerate some mismatches at specific positions along their binding site to detect the presence of indels. Usually, indels are detected by loss of signal as it is difficult to design one set of primers or probes against all possible combinations of indels and thus it is easier to design against the wildtype sequence instead. But in the authors' assay, indels are detected by gain of signal, which is more unusual.

Nevertheless, while the work is interesting from an academic point of view, I am not entirely convinced of its real-world applicability. A major issue is that detection of cancer mutations is not extremely time critical (unlike infectious diseases like the COVID-19 pandemic). Generally, it will not matter if a cancer is diagnosed in 30 minutes or a few days. In the introduction, the authors tried to give hyperleukocytosis as an example. But the pathophysiology of hyperleukocytosis is not well understood, so it is impossible to create a CRISPR assay for it. Hence, I believe that clinicians would prefer to rely on methods like next generation sequencing (NGS) or real-time PCR. Moreover, CRISPR-based assays are challenging to multiplex and thus cannot serve as a general screening tool for multiple types of mutation. Consequently, the authors might want to consider alternative applications of their method.

Major comments:

1) Can the method work for small (1bp or 2bp) indels? Larger indels (like 4bp) are easier to detect and distinguish from wildtype.

2) For the NPM1 target, why don't the authors use or test a crRNA that hybridises to the TCTG insertion perfectly since it is most common (fig 4a)? I would imagine testing a crRNA that recognises the most common insertion (and then checking whether it would work against other insertions) to be a natural thing to do.

3) In the last results section, I find it odd that the authors titrated input template amounts using a single patient sample and then generalised. Fluorescence intensity would depend on gene expression level and mutation load. Can the authors test a few different patient samples with varying gene expression levels and mutation loads?

4) As mentioned above, cancer mutation detection might not be the best use case. Can the authors show another application that genuinely requires fast turnaround time? I am not convinced that identification of a 4bp insertion in NPM1 for AML requires a 30min turnaround time. There should be rigorous benchmarking against NGS and qPCR. Notably, targeted amplicon sequencing on Illumina can readily achieve 0.1% mutation detection rate and can further multiplex to cover many oncogenes and tumor suppressors. In addition, the turnover time is even faster for nanopore sequencing, which can also achieve 0.1% VAF (variant allele frequency) detection.

Minor comments:

5) Please indicate that "OS" stands for original sequence in the main text

6) CoHIT does not detect all variants (definitely not single nucleotide substitutions). Hence, the title and abstract are misleading, and the authors should make clear that their assay is for indels only.

7) Please indicate that "FGS" stands for first generation sequencing such as Sanger sequencing in the main text. Where does the claim of 10% rate for FGS come from? It is still possible to quantify 5-10% from Sanger sequencing chromatograms.

8) The persistent claim of one crRNA to identify numerous gene variants (indels) is misleading. The authors' assay can effectively only check one mutation site at a time.

9) There is a typo on line 241: "two important variants: c. 1459_1473del and c.1459_1473del" (they are the same).

Reviewer #3 :

Remarks to the Author:

In this manuscript the authors use Cas12a to detect mutations in genes where there are multiple possible different nucleotide sequences, as in the case of indels that define NPM1 mutations. The outlined approach could be broadly applicable across various diseases defined by multiple mutation possibilities at the same site, and they showed the ability to use their system for other mutations as well. The approaches seem robust however the main concern is the novelty and likelihood this approach will truly find a place in clinical diagnostics. The authors approach ties together several approaches that were recently novel but in and of itself the approach is not.

There are multiple examples of using Cas12a approaches to detect genetic variants. This flourished in the COVID literature:

- Ding X, Yin K, Li Z, Lalla RV, Ballesteros E, Sfeir MM, Liu C. Ultrasensitive and visual detection of SARS-CoV-2 using all-in-one dual CRISPR-Cas12a assay. *Nat Commun.* 2020 Sep 18;11(1):4711. doi: 10.1038/s41467-020-18575-6. PMID: 32948757; PMCID: PMC7501862.

- Liang Y, Zou L, Lin H, Li B, Zhao J, Wang H, Sun J, Chen J, Mo Y, Yang X, Deng X, Tang S. Detection of Major SARS-CoV-2 Variants of Concern in Clinical Samples via CRISPR-Cas12a-Mediated Mutation-Specific Assay. *ACS Synth Biol.* 2022 May 20;11(5):1811-1823. doi: 10.1021/acssynbio.1c00643. Epub 2022 Apr 28. PMID: 35481381.

- Fasching CL, Servellita V, McKay B, Nagesh V, Broughton JP, Sotomayor-Gonzalez A, Wang B, Brazer N, Reyes K, Streithorst J, Deraney RN, Stanfield E, Hendriks CG, Fung B, Miller S, Ching J, Chen JS, Chiu CY. COVID-19 Variant Detection with a High-Fidelity CRISPR-Cas12 Enzyme. *J Clin Microbiol.* 2022 Jul 20;60(7):e0026122. doi: 10.1128/jcm.00261-22. Epub 2022 Jun 29. PMID: 35766492; PMCID: PMC9297821.

Cas12a-based methods for variant detection have further expanded into cancers as well, with this article describing a rapid, highly sensitive and specific, Cas12a-based method for detection of FLT3 D835Y/H/V/F mutations:

- Liu Y, Chen Y, Dang L, et al. EasyCatch, a convenient, sensitive and specific CRISPR detection system for cancer gene mutations. *Mol Cancer*. 2021;20(1):157. Published 2021 Dec 2. doi:10.1186/s12943-021-01456-x

Strengths:

- This work is potentially relevant to NPM1 mutation identification. They picked a good disease model for their test. This problem is that it is not so clear that the approach is drastically better than current mutation identification approaches.
- The fact that they engineered Cas12a mutants with more permissive PAM recognition to allow for better detection of mutants is a strength. They did a good job creating Cas12a mutations that will identify most of the major NPM1 mutations. That said, other groups have worked on developing Cas9 and Cas12a mutants that allow for more permissive PAMs for various functions. But the fact this group created several mutants and screened them to improve for tolerance of mismatches is well done.

Weaknesses:

- Style and writing: Significant editing is needed.
- Introduction/framing of their work: Their introduction to why detection of mutations is important, while brief, needs work. One of the sentences is: "Moreover, early detection of low-frequency mutations can help prevent cancer in patients." I don't think this is entirely true of current practice, though maybe in the future with cell free DNA. They do not offer an example or a citation for this either. The other background importance for mutation detection that they give is that it helps clinicians diagnose, select treatment, and monitor for recurrence. They cite a review article and the NCCN guidelines, which is pretty vague.
- They refer to sensitivity to mean two different things. They mean it both in terms of a novel statistical test to rule out a mutation and then also the ability to detect low levels of a mutation. They should more clearly differentiate between these.
- In some of the figures and figure legends, it is unclear how much replication they did, as they only have solid bars and the figure legends don't indicate number of replicates. It would be helpful to get a sense of the variability inherent in their assays (Figure 4E, for example). Also, clarifying what statistical comparisons between groups they were doing (and why) would be helpful.

All said much more validation for potential use as a clinical diagnostic would be very helpful since the approach is not novel.

Reviewer #4 :

Remarks to the Author:

Liu et al report a novel, CRISPR-CAS12a based assay for mutation detection with the advantage of a good mismatch tolerance. Through a series of optimization, they could show this assay is sensitive and specific, and surpass first generation sequencing test, and argue that this assay may be applied to clinical samples. The authors should be congratulated for having done such an excellent technical development work! The assay appears technically sound. However, evidence of clinical utility is insufficient.

An clinical assay can be used at diagnosis and/or disease monitoring. An ideal assay should be fast, sensitive/specific and/or cost-effective.

1) Although this Co-HIT assay is faster with a readout probably in hours, and it is more sensitive than first generation sequencing (which has been gradually phased out in clinical practice), I doubt it will surpass NGS as a diagnostic test. As they showed in this study, NGS reliably detected all NPM1 mutations. Cancer patients often carry multiple mutations (3-8 mutations in AML, 10-30 mutations in solid tumors), and NGS, either whole exome or targeted panel, has the benefit of providing all the information on numerous mutations at once (although often with weeks delay

but this will certainly be shortened as bioinformatic pipelines improve). The authors haven't shown a multiplex approach of CO-HIT for detecting multiple mutations simultaneously with reasonable specificity and sensitivity. Even they could, the requirement of knowing the mutation site for assay design would preclude the detection of unknown mutations, making it impossible for screening at diagnosis.

2). The authors showed that CO-HIT can detect as low as 0.01% mutation burden, suggesting its potential utility in minimal residual disease detection. However, the authors used spiked in plasmid DNA but not genomic DNA. Therefore, its detection limit in real clinical practice is unproven. Furthermore, currently ddPCR for NPM1 mutation can easily reach 0.001% in clinical samples, far better than CO-HIT. The authors should be very cautious when stating the sensitivity based on plasmid spike in experiments.

3) One potential application in clinical practice could be a rapid assay for critical clinical actionable mutation detection such as NPM1, FLT3, BRAF, EGFR, etc. For that, the authors need to show it will work as a multiplex assay. With rapid improvement in NGS reporting, some laboratories have developed a short NGS panel including these mutations and can report results in 3-5 days. Therefore, unless this CO-HIT assay clearly edges NGS, its clinical utility even in this narrow space will remain questionable.

Response to Reviewer #1

Reviewer #1 (Remarks to the Author):

Key Results

Liu et al describe in this work a CRISPR-based diagnostic detection platform capable of detecting multiple disease variants in a single sample using just a single Cas12a crRNA. This work is an extension of previously described Cas13a and Cas12a-based detection platforms that gained notoriety for the detection of COVID-19 and other respiratory pathogens. A key finding from this study is that the “CoHIT” technique is a rapid, sensitive, and cost-effective tool for cancer diagnosis and simultaneous detection of multiple variants at a single disease locus. The application of this tool is both novel and relevant to real challenges in cancer diagnostics and represents a novel application for Cas12a/Cas13a detection systems that have not been able to supplant qPCR and antigen tests for RNA pathogen detection.

Authors: We sincerely appreciate the reviewer for the on-point summary and the positive feedback of our manuscript, as well as the extensive guidance on how to improve our work. We have incorporated every suggestion, in some cases performing follow-on experiments to strengthen our claims.

The potential application of this work to the field of minimal residual disease detection (MRD) is of high relevance and the authors could consider including this in the discussion to elevate the relevance and significance of their findings. I think the notion of providing point of care cancer diagnostics is quite provocative.

Authors: We are grateful for the professional comment and suggestion. We fully agree that MRD detection is a promising application of our work. In the revised manuscript, we have added both relevant words and experiments to demonstrate this point:

“For instance, the measurement of minimal residual disease (MRD) levels, which indicate the remaining quantity of malignant cells in the body, has important prognostic and therapeutic implications (Blood. 2022;139(6):835-844) (J Clin Oncol. 2022;40(6):567-575).” (*The Introduction section, Line 46 to 48*)

“Considering that NPM1 mutation is a clinically recommended MRD testing item (Haematologica. 2022;107(12):2810-2822), we tested the CoHIT system to detect residual leukaemia in NPM1-mutated AML patients. We utilized cryopreserved bone marrow DNA samples collected from three patients (Patient-MRD-1~3) at multiple time points, ranging from initial diagnosis through remission to recurrence during follow-up visits.” (*The Results section, Line 273 to 276*)

We have added MRD related experiments through testing clinical samples. As shown in the following figures (Fig. 6d-g, Supplementary Fig. 22-24), we collected a succession of cryopreserved DNA of bone marrow samples belonging to three AML patients (Patients-MRD-1~3), ranging from initial diagnosis through remission to recurrence during follow-up visits. These samples were detected using FGS, NGS, and our CoHIT method at the same time. The results showed that the detection outcomes of CoHIT is consistent with NGS. Moreover, the CoHIT method can detect low abundance mutations earlier than FGS, as shown by the samples of Patient-MRD-1 (dated 06/2022 and 10/2022) and Patient-MRD-2 (dated 03/2021). Therefore, CoHIT detection can provide earlier warning of leukaemia recurrence. Based on its benefits in terms of speed, sensitivity, convenience, and cost-effectiveness, we believe that the CoHIT method has the potential to become a new MRD monitoring method, particularly offering enhanced convenience to patients in underdeveloped areas.

Fig. 6 | Detection of clinical samples and MRD using the CoHIT system. **d** Flowchart of the MRD experiment. **e** CoHIT result for Patient-MRD-1. The sampling dates and naked-eye results are shown for each sample. The asterisks indicate P values compared with healthy WT control (HC). **f** NGS and FGS results of Patient-MRD-1 samples dated 10/2021, 06/2022, 10/2022, and 01/2023. **g** CoHIT and NGS results for Patient-MRD-2 and 3. The asterisks indicate P values obtained by comparison with HC in CoHIT detection. The percentages are the insTCTG mutation rates determined by NGS.

Fig. S22 | FGS chromatograms of the eight bone marrow samples of Patient-MRD-1, collected between 10/2021 (First visit) and 01/2023. Mutant peaks are underlined in red. The percentages near the dates are NGS results of NPM1 gene c.863_864insTCTG mutation.

Fig. S23 | FGS, NGS, and CoHIT detection results of Patient-MRD-2. **a** Six bone marrow samples were collected between 04/2020 (First visit) and 07/2021 and tested by CoHIT assay. Final fluorescence values and naked-eye photos of the six samples are shown in chronological order. The asterisks indicate P values compared with healthy WT control (HC). **b** Time course of CoHIT detection of the six samples and HC. **c** FGS chromatograms of the six samples. Mutant peaks are underlined in red. The percentages near the dates are NGS results of the NPM1 gene c.863_864insTCTG mutation.

Fig. S24 | FGS, NGS, and CoHIT detection results of Patient-MRD-3. **a** Three bone marrow samples were collected between 10/2021 (First visit) and 06/2022 and tested by CoHIT assay. Final fluorescence values and naked-eye photos of the three samples were shown in chronological order. The asterisks indicate P values compared with healthy WT control (HC). **b** Time-course of CoHIT detection of the three samples and HC. **c** FGS chromatograms of the three samples. Mutant peaks are underlined in red. The percentages near the dates are NGS results of the NPM1 gene c.863_864insTCTG mutation.

Overall, this is generally high quality work and my only reservations are towards the presentation of the protein engineering aspects and their relative novelty/significance, which I discuss later in this review.

Validity

Generally the data are robust and valid – the authors do a nice job of presenting the qualitative, point of care fluorescence data while simultaneously showing real time fluorescence data to further bolster claims about the technique. The authors also nicely describe multiple cancer disease applications and thoroughly explore these in great detail.

Authors: We sincerely appreciate the reviewer for the positive comment.

I have two main concerns regarding the general validity of the work. First, with NGS, there is less concern about false positives/negatives with respect to analyzing cancer disease variants. Presumably with CoHIT, since you are amplifying only the region of interest, this minimizes the potential for off-target cleavage to occur and give a false positive result. I'm nevertheless concerned about combining Cas12a mutants with massively increased intrinsic activity and reduced PAM specificity (thus higher off-target effects) and what that would mean for off-target cleavage, particularly with ERA amplicons that are highly repetitive. I think an exploration of the specificity of these combined nucleases (either in the context of CoHIT or otherwise) is warranted – particularly given the consequences of a false positive or negative result.

Authors: Thanks for the reviewer's thoughtful feedback and concerns. We agree that NGS performs well in analyzing cancer disease variants. However, NGS has its limitations in cost and turnaround time. We understand that off-target effects are a critical consideration in CRISPR-based methodologies, and have taken several steps to address and assess this issue. We acknowledge the importance of specificity in mutation detection, and we agree that investigating off-target cleavage effects is crucial for assessing the reliability of the CoHIT approach. To evaluate the potential off-target effects of our approach, we utilized Cas-OFFinder (<http://www.rgenome.net/cas-offinder/>) to predict the top potential off-target sites on the human genome for KIT, EGFR, BRAF, and NPM1 crRNAs, which were used in our manuscript. As illustrated in Fig. S26 and Supplementary Table S3 and S4, we searched potential off-target sites both with and without TTTN PAM, and notably, there are a minimum of three nucleic acid mismatches between the target gene and these predicted off-target sites. Subsequently, we cloned 60 potential off-target gene fragments for further testing. Encouragingly, our experimental results, depicted in Fig. S26, did not reveal any instances of cross-reaction, providing confidence in the specificity of our system.

Furthermore, as the reviewer mentioned, the CoHIT system includes an ERA target gene amplification. This amplification reaction is highly specific, as there are no false-positive signals in our tests with both plasmid samples and clinical samples (Fig. 4k-o in the manuscript). These results affirm the specificity of the CoHIT system in our experimental settings. We appreciate the reviewer's concerns and have taken rigorous steps to address and validate the specificity of our methodology, emphasizing its suitability for accurate and reliable detection of disease variants.

Fig. S26 | Off-target evaluation of Cas12a proteins on four sites. a Detection of top five predicted off-targets in human genome with TTTN PAM using six different Cas12a proteins. PAM and mismatch bases in the sequences are marked in blue and red, respectively. The off-targets were predicted using the online web tool Cas-OFFinder (<http://www.rgenome.net/cas-offinder/>), then PCR-amplified and purified for the Cas12a in vitro cleavage assay, with 5E10 copies of DNA as substrates, reacting for 30 min. Naked-eye photos show the results of enAsU-R Cas12a. **b** Detection of top ten predicted off-targets in human genome without PAM limitation using enAsU-R Cas12a. The off-target sequences are listed in Supplementary Table S3 and S4.

Table S3. Top 5 potential off-targets predicted by Cas-OFFinder with canonical Cas12a TTTN PAM

Targets	Off-targets	Sequence	Chromosome	Position	Direction	Mismatches
NPM1	Off-1	TTTTTCCACTGCCAGGcTgAtgTCTTt	chr5	109936851	+	4
	Off-2	TTTGcCCACTGcTAgGgAGAGATCTgG	chr5	160602494	+	4
	Off-3	TTTGTCCTACTGCCAaaCAaAGATCTTa	chr4	179715531	-	4
	Off-4	TTTCTCCACTaCCAGGCAGgGAaCTcG	chr9	93945527	+	4
	Off-5	TTTTTCCACTctCAGGCAGAcATCTTa	chr9	98450169	-	4
KIT	Off-1	TTTAAAGTAaAGGTTcTTGAGGAGgTg	chr8	118313976	+	4
	Off-2	TTTCAAcTACcatTTGTTGAGGAcATA	chr8	36434365	-	5
	Off-3	TTTCATGTACAGTTtTTGtGcAGAcA	chr5	32542820	-	5
	Off-4	TTTCAAGTAaAaGTTtTTGtGtAGATA	chr5	73789040	-	5
	Off-5	TTTGAAGTACTGcTTGaTGAtGAGATT	chr5	108786853	+	5
BRAF	Off-1	TTTTCTtCTGAGCcaTCAACATTTTCA	chr4	28107059	+	3
	Off-2	TTTCCTtCTtAGCATgCAACATTaTCA	chr1	218027763	-	4
	Off-3	TTTTCTcCTtAGCATTtAACATTTTCTt	chr1	227140302	-	4
	Off-4	TTTTgTGCTatGCATTtAACATTTTCA	chr22	22113958	-	4
	Off-5	TTTTCaGCTGccCATTCAACATTTcCA	chr2	31033818	+	4
EGFR	Off-1	TTTCaGAGATGTTTTaATAGCagCtGG	chr5	67637256	+	5
	Off-2	TTTGGGAGATGTTTTGtTAGaGAgGaa	chr1	20836628	-	5
	Off-3	TTTTGGAttTtTTTTaATAGaGACGGG	chr7	16690369	+	5
	Off-4	TTTTGaAGAgTTTTcATAGCGAgGaG	chr7	141387589	+	5
	Off-5	TTTTGtgGATtTTTTGgTAGaGACGGG	chr2	28848868	-	5

* The lowercase letters in the sequences represent bases that are different from the on-target sequence.

Table S4. Top 10 genome-wide potential off-targets predicted without canonical Cas12a TTTN PAM

Targets	Off-targets	Sequences	Chromosome	Position	Direction	Mismatches
NPM1	Off-1	TCCACTGCctGGCAGAAATCTTG	chr16	12215583	-	2
	Off-2	TCCACTGCaAGGCAGAGATagTG	chr11	6598961	+	3
	Off-3	TCCcaTGCCAGGCAGAGAgCTTG	chr17	16815425	-	3
	Off-4	TCCcaTGCCAGGCAGAGAgCTTG	chr17	41576298	-	3
	Off-5	TCCAtctCCAGGCAGAGATCTTG	chr18	72718056	+	3
	Off-6	TaCAgTGCCAGGCACAGATCTTG	chr20	13927353	+	3
	Off-7	TCCAtTtCCAGGCAGAGAACTTG	chr2	52579792	-	3
	Off-8	TCCaCaGCCAGGCAGtGAgCTTG	chr6	1066859	+	3
	Off-9	TaCACTGCCAGGCAGAgTCTTt	chr6	112317595	+	3
	Off-10	TtCACTGCCAGGCAtAGATCTTt	chr7	98266151	+	3
KIT	Off-1	GGAGATGTTTTGATaGACGtG	chr6	145403139	+	3
	Off-2	GGAGgTGTTTTGATAGtGAgGGG	chr6	46570824	-	3
	Off-3	tGAGATGTTTTGATAGaGACatG	chr10	123049076	+	4
	Off-4	GGAGATGTTgTGATgGCGAaGaG	chr10	98525375	-	4
	Off-5	GGAGATGTTTTGATAcAGACatG	chr11	101519274	-	4
	Off-6	tGAGATGTTTTGATAGgGACatG	chr11	6082661	-	4
	Off-7	GGAGccGTTTctATAGCAGCGGG	chr12	4562146	+	4
	Off-8	GGAGATtTTTTGAaAGCaAgGGG	chr13	31051787	-	4
	Off-9	tGAGATGTTTTGATAcAGACaGG	chr14	49751391	-	4
	Off-10	tGAGATGTTTTGATAGaGACatG	chr1	45071029	+	4
BRAF	Off-1	CTGCTGAtCATgCcACATTTTCA	chr1	163410627	+	3
	Off-2	CTGCaGAGCATTCAAtAaTTTCA	chr15	97020980	-	3
	Off-3	CTGCTGAGCAGtGAAcAcTTTCA	chr17	14804138	-	3
	Off-4	CTGgTGAGCATTCAACATaTaCA	chr18	48704342	+	3
	Off-5	CTaCaGAGCATTCAaAATTTTCA	chr2	143167099	+	3
	Off-6	tTGCTGAGCATTCAACATgTgCA	chr2	81372939	+	3
	Off-7	CTGCTtAGCgTTCAACATTTTtA	chr3	14118800	-	3
	Off-8	CgGCcGAGCATTCAAgATTTTCA	chr3	58583094	+	3
	Off-9	CTtCTGAGCcaTCAACATTTTCA	chr4	28107064	+	3
	Off-10	CTGtTtAGCATTCAACATaTTCA	chr5	163612491	+	3
EGFR	Off-1	AAGTACAGtTTcTTgGGAGATA	chr1	107861369	-	3
	Off-2	AAGaACAGGTTGcTGAGGAGAgA	chr1	238012654	+	3
	Off-3	AAGTACAGGgTGTtGtGGAAaATA	chr16	49336558	-	3
	Off-4	AAGTACAGGTTcTaGAGaAGATA	chr17	52551267	-	3
	Off-5	AtGTAAaAGGTTGTTGAGGAGAAaA	chr5	136732235	+	3
	Off-6	AAGTACAGGaTGTTGAGaAGtTA	chr5	140684538	+	3
	Off-7	AAGTACAAaGaTGTTGAGGAGATg	chr6	123860139	+	3
	Off-8	AAGgACTgGTTGTTgGGAGATg	chr10	109240837	+	4
	Off-9	AAGaACAGGTTGTTGtGaAaATA	chr10	118334916	-	4
	Off-10	AAacACAGGTTGtAGAGGAGAgA	chr10	11897980	-	4

* The lowercase letters in the sequences represent bases that are different from the on-target sequence.

Secondly (somewhat related), I'm concerned about the data supporting the need for a further engineered nuclease. There are two figures dedicated towards combining previously described enhanced activity and PAM specificity altering mutations and nevertheless the data in Figure 4 would suggest that perhaps these further engineered variants, which haven't been characterized to the extent of enAsCas12a or AsCas12a Ultra, are in fact not needed. Additional data supporting and justifying the need for these further engineered and combined variants is needed.

Authors: We apologize for the confusion and the insufficient presentation on the necessity of our work on Cas12a engineering. In fact, this work is of great significance for establishing the CoHIT system. The main advantages of our engineered enAsU-R Cas12a protein lies in the improvement of mismatch tolerance and the expansion of PAM scope. On the one hand, its improved mismatch tolerance allows it to simultaneously detect multiple similar variants using a single crRNA, for example, hundreds of c.863_864insNNNN variants of NPM1 gene. To further confirm the effectiveness of the protein engineering, we added two experiments in the revised manuscript:

- 1) Compare enAsU-R protein with other five Cas12a proteins in the detection of 32 random TA clones carrying the NPM1 gene c.863_864insNNNN mutated fragments. As shown in Fig. 3g, more clones with higher fluorescence signals were detected by enAsU-R than by the other five Cas12a variants (Fig. 3g and Supplementary Fig. S4a). Subsequent Sanger sequencing showed that most of the 32 clones had three or four mismatches to the crRNA, further confirming the heightened mismatch tolerance of the enAsU-R protein (Supplementary Table S2)

Fig. 3 | Detection of multiple NPM1 mutations via an enAsU-R Cas12a-induced in vitro cleavage assay. g The six chessboard diagrams display the normalized fluorescence intensity of six Cas12a/crRNA1-induced in vitro detection of the PCR purified products of 32 random TA clones, WT, and NC.

Fig. S4 | Comparison of six AsCas12a variants on mismatch tolerance. a Detection of 32 random TA clones carrying the NPM1 c.863_864insNNNN mutated fragments by different Cas12a-induced in vitro cleavage assays. The histogram shows the number of monoclonal cases with NF > 0.1 for each Cas12a protein variant. NF, normalized fluorescence.

Table S2. Sequences of the 32 random TA clones with NPM1 mutations

Clone ID	Sequence	Clone ID	Sequence
X1	gatctctgTGCgagcagtgaggaggaa	X17	gatctctgCGGCgagcagtgaggaggaa
X2	gatctctgCTCTgagcagtgaggaggaa	X18	gatctctgGTTTgagcagtgaggaggaa
X3	gatctctgCCTCgagcagtgaggaggaa	X19	gatctctgCAAGgagcagtgaggaggaa
X4	gatctctgAATCgagcagtgaggaggaa	X20	gatctctgGGAAgagcagtgaggaggaa
X5	gatctctgCGTCgagcagtgaggaggaa	X21	gatctctgTTCGgagcagtgaggaggaa
X6	gatctctgTCCGgagcagtgaggaggaa	X22	gatctctgCGCGgagcagtgaggaggaa
X7	gatctctgTCGAgagcagtgaggaggaa	X23	gatctctgTCGTgagcagtgaggaggaa
X8	gatctctgGAGGgagcagtgaggaggaa	X24	gatctctgCTCGgagcagtgaggaggaa
X9	gatctctgCGAAgagcagtgaggaggaa	X25	gatctctgATTGgagcagtgaggaggaa
X10	gatctctgGTAGgagcagtgaggaggaa	X26	gatctctgGACTgagcagtgaggaggaa
X11	gatctctgTTAGgagcagtgaggaggaa	X27	gatctctgTTATgagcagtgaggaggaa
X12	gatctctgAAGGgagcagtgaggaggaa	X28	gatctctgGACAgagcagtgaggaggaa
X13	gatctctgCCCGgagcagtgaggaggaa	X29	gatctctgTCTTgagcagtgaggaggaa
X14	gatctctgGTACgagcagtgaggaggaa	X30	gatctctgTTTGgagcagtgaggaggaa
X15	gatctctgAGTGgagcagtgaggaggaa	X31	gatctctgCATAgagcagtgaggaggaa
X16	gatctctgTTAgagcagtgaggaggaa	X32	gatctctgAGGAgagcagtgaggaggaa

2) Compare enAsU-R protein with other five Cas12a proteins in the detection of top eleven c.863_864insNNNN mutations of the NPM1 gene. By employing enAsU-R/crRNA1, we detected the top eleven NPM1 gene c.863_864 4-bp insertions, including instCTG, CATG, CCTG, CCAG, CCGG, CTTG, TATG, TCGG, TAAG, CAGG, and CAGA, covering more than 97% of the NPM1-mutated AML cases in clinical practice. enAsU-R consistently produced robust fluorescence signals for all the tested insertions, outperforming the other five Cas12a variants, and no significant WT cross-reactivity was observed (Fig. 3h and Supplementary Fig. S4b-c)

Fig. 3 | Detection of multiple NPM1 mutations via an enAsU-R Cas12a-induced in vitro cleavage assay. h The normalized fluorescence intensity of six Cas12a/crRNA1-induced in vitro detection of the top 11 NPM1 gene c.863_864 4-bp insertions using 5E10 copies of DNA as substrates, and reacting for 20 min at 37°C. The fluorescence image shows the naked-eye result of enAsU-R under blue light.

Fig. S4 | Comparison of six AsCas12a variants on mismatch tolerance. b Detection of the top eleven c.863_864insNNNN mutations by different Cas12a-induced in vitro cleavage assays. The histogram shows the number of 4-bp insertion cases with NF > 0.5 for each Cas12a protein variant. **c** Final fluorescence values of the top eleven c.863_864insNNNN mutations by enAsU-R/crRNA1 detection. The fluorescence image shows the naked-eye result under blue light.

On the other hand, the broad and distinctive NNCV and TTTV PAM of enAsU-R Cas12a significantly simplifies crRNA design for most mutation sites within the human genome, as shown in Fig. 3a. This PAM feature greatly expand the application scope of the CoHIT system.

Fig. 3 | Detection of multiple NPM1 mutations via an enAsU-R Cas12a-induced in vitro cleavage assay. a PAM identification assay of enAsU-R Cas12a. Normalized cleavage rates of all possible 4-base PAMs are shown at different blue depths.

Significance

I think the application of the Cas12a/Cas13a diagnostics detection technology to cancer disease typing is relevant and significant. I typically have only seen this technology being used to detect presence/absence of respiratory pathogens. However, my main concern about significance is due to the protein engineering aspect, the fraction of the manuscript devoted to it, and whether it truly represents a novel variant or whether this piece is best relegated to the supplemental. As mentioned, while the clinical application is noteworthy, it is not obvious to me that this work absent the protein engineering piece is of high significance.

Ultimately, the previously described AsCas12a variants enAsCas12a and AsCas12a Ultra were the result of an exhaustive set of directed evolution and rational mutagenesis investigations to uncover protein mutants of high intrinsic potency and reduced PAM specificity. In the work presented here, the mutants carried forward in CoHIT were simply various combinations of mutants already described in the literature and the resulting characterization thereof. The authors use words/phrases like “screening, protein engineering, and novel” to describe these variants and devote 2 figures to the topic which seems disingenuous and not truly novel. Of further concern, it is not convincing to me that the resulting mutants are critical for the CoHIT technique to be successful, and I don’t believe the data yet support that conclusion either. I recommend moving the protein engineering figures to the supplemental and revising the paper to instead focus on the application work, which is novel and significant. I acknowledge that a protein engineering or directed evolution study to isolate mutants specifically adapted for the CoHIT technique is also both interesting and provocative.

Authors: We thank the reviewer for the positive view of our manuscript. We understand the reviewer’s concern and thanks for the thoughtful suggestion. As shown above, we have added some experiments to confirm the effectiveness of the protein engineering work. We believe that the enAsU-R Cas12a represents a novel variant and have its own advantages compared to existing Cas12a variants. It is characterized by an improved mismatch tolerance and a broad and distinctive NNCV and TTTV PAM. enAsU-R protein is critical for the CoHIT technique because: 1) Its improved mismatch tolerance allows the CoHIT system to achieve full-coverage detection of multiple variants. 2) Its broad PAM makes it easy to design and screen the best-tolerant crRNAs for most mutation sites in the human genome. In the revised manuscript, we have replaced the original Fig. 4g-h with the results of multi-protein comparison experiments to more clearly demonstrate the significance of the protein engineering work (Fig. 3g-h and Supplementary Fig. S4). And according to the reviewer’s advice, we moved part of the work to the supplementary materials and integrated the original Fig. 3 and Fig. 4 into one figure (Fig. 3) in the revised version.

Fig. 3 | Detection of multiple NPM1 mutations via an enAsU-R Cas12a-induced in vitro cleavage assay. **g** The six chessboard diagrams display the normalized fluorescence intensity of six Cas12a/crRNA1-induced in vitro detection of the PCR purified products of 32 random TA clones, WT, and NC. **h** The normalized fluorescence intensity of six Cas12a/crRNA1-induced in vitro detection of the top 11 NPM1 gene c.863_864 4-bp insertions using 5E10 copies of DNA as substrates, and reacting for 20 min at 37°C. The fluorescence image shows the naked-eye result of enAsU-R under blue light.

Fig. S4 | Comparison of six AsCas12a variants on mismatch tolerance. **a** Detection of 32 random TA clones carrying the NPM1 c.863_864insNNNN mutated fragments by different Cas12a-induced in vitro cleavage assays. The histogram shows the number of monoclonal cases with NF > 0.1 for each Cas12a protein variant. NF, normalized fluorescence. **b** Detection of the top eleven c.863_864insNNNN mutations by different Cas12a-induced in vitro cleavage assays. The histogram shows the number of 4-bp insertion cases with NF > 0.5 for each Cas12a protein variant. **c** Final fluorescence values of the top eleven c.863_864insNNNN mutations by enAsU-R/crRNA1 detection. The fluorescence image shows the naked-eye result under blue light.

We fully agree that supplementing the application work would benefit the novelty of our article. In the revised manuscript, we have expanded the clinical detection scale from 70 patients to 108 patients, and added three parts of experiments:

1) MRD detection, as mentioned above. As shown in Fig. 6d-g and Supplementary Fig. 22-24, we tested cryopreserved DNA of Patients-MRD-1~3 ranging from initial diagnosis through remission to recurrence during follow-up visits. The results showed that the detection outcomes of CoHIT is consistent with NGS. Moreover, the CoHIT method can detect low abundance mutations earlier than FGS. Therefore, CoHIT detection can provide earlier warning of leukaemia recurrence.

Fig. 6 | Detection of clinical samples and MRD using the CoHIT system. **d** Flowchart of the MRD experiment. **e** CoHIT result for Patient-MRD-1. The sampling dates and naked-eye results are shown for each sample. The asterisks indicate P values compared with healthy WT control (HC). **f** NGS and FGS results of Patient-MRD-1 samples dated 10/2021, 06/2022, 10/2022, and 01/2023. **g** CoHIT and NGS results for Patient-MRD-2 and Patient-MRD-3. The asterisks indicate P values obtained by comparison with HC in CoHIT detection. The percentages are the insTCTG mutation rates determined by NGS.

Fig. S22 | FGS chromatograms of the eight bone marrow samples of Patient-MRD-1, collected between 10/2021 (First visit) and 01/2023. Mutant peaks are underlined in red. The percentages near the dates are NGS results of NPM1 gene c.863_864insTCTG mutation.

Fig. S23 | FGS, NGS, and CoHIT detection results of Patient-MRD-2. a Six bone marrow samples were collected between 04/2020 (First visit) and 07/2021 and tested by CoHIT assay. Final fluorescence values and naked-eye photos of the six samples are shown in chronological order. The asterisks indicate P values compared with healthy WT control (HC). **b** Time course of CoHIT detection of the six samples and HC. **c** FGS chromatograms of the six samples. Mutant peaks are underlined in red. The percentages near the dates are NGS results of the NPM1 gene c.863_864insTCTG mutation.

Fig. S24 | FGS, NGS, and CoHIT detection results of Patient-MRD-3. **a** Three bone marrow samples were collected between 10/2021 (First visit) and 06/2022 and tested by CoHIT assay. Final fluorescence values and naked-eye photos of the three samples were shown in chronological order. The asterisks indicate P values compared with healthy WT control (HC). **b** Time-course of CoHIT detection of the three samples and HC. **c** FGS chromatograms of the three samples. Mutant peaks are underlined in red. The percentages near the dates are NGS results of NPM1 gene c.863_864insTCTG mutation.

2) Combine the CoHIT system with lateral flow strip assay (LFA), which is beneficial for application in underdeveloped areas. As shown in Fig. 7a-e, by replacing FAM-ssDNA-BHQ1 reporters with biotin-ssDNA-FAM reporters in the CoHIT system, the cleaved ssDNA would be captured on the test line, suggesting a positive result. After optimization of the biotin-ssDNA-FAM concentration and CoHIT reaction time, we used the CoHIT-LFA system to detect ten patient samples. As shown in Fig. 7e, the CoHIT-LFA method successfully distinguished between positive and negative patient samples.

Fig. 7 | CoHIT-LFA system and microfluidic chip-based CoHIT detection. **a** Detection principle of the LFA strip for the CoHIT system. **b** Biotin-ssDNA-FAM concentration optimization for the LFA assay. **c** Flow chart of the CoHIT-LFA assay. **d** Optimization of the CoHIT reaction time for the CoHIT-LFA assay. W, WT sample. M, mutant sample. **e** Detection of the NPM1 gene c.863_864 4-bp insertion in ten AML patient samples. The percentages are mutation rates.

3) Combine the CoHIT assay with a microfluidic chip to achieve multiplexing detection. As shown in Fig. 7f-i and Supplementary Fig. S25, we made a microfluidic device with one loading chamber and nine reaction chambers. Mutation status of seven sites can be obtained simultaneously through a single loading step. To verify this point, we tested the NPM1 mutation site (Target 1) and six FLT3-ITD (internal tandem duplication) mutation sites (Targets 2~7), which are all hotspot mutations in AML. The results showed that each target could be effectively detected and the sensitivity could reach 0.1% under naked eye. Further experiments on clinical samples proved that it can simultaneously detect mutations at different sites, P85 in Fig. 7i for example.

Fig. 7 | CoHIT-LFA system and microfluidic chip-based CoHIT detection. **f** Flow chart of the microfluidic chip-based multiplexing CoHIT detection. IC, inner control. **g** Detection of 1E6 copies of mutated fragments-containing plasmid in 100 ng WT genomic DNA. **h** Detection of Target 2 at frequencies of 10%, 1%, and 0.1%. **i** Detection in clinical samples from six AML patients.

Target	Gene mutation
1	NPM1 c.863_864 4-bp insertion
2	FLT3 c.1770_1771 24-bp insertion
3	FLT3 c.1793_1794 21-bp insertion
4	FLT3 c.1790_1791 18-bp insertion
5	FLT3 c.1776_1777 18-bp insertion
6	FLT3 c.1784_1785 21-bp insertion
7	FLT3 c.1796_1797 27-bp insertion
IC	GAPDH WT

Fig. S25 | Gene mutation sites and types of targets 1~7 in the microfluidic chip-based multiplexing CoHIT assay.

In addition, we appreciate the reviewer's suggestion on protein evolution, which will be a part of our future work.

Data and methodology

No additional concerns around the data as presented or methodology.

Analytical approach

No general concerns about the analytical approach though the authors will need to comply with the Nature Communications statistics policies in more finalized drafts (ie - N=3, etc).

Authors: We thank the reviewer for the positive feedback. In the revised manuscript, we have ensured that we comply with the Nature Communications statistics policies, including indicating the repetition of experiments.

Suggested improvements

I have indicated some general concerns about the protein engineering content and whether the combined mutants described/characterized in this work are novel and are in fact necessary for the CoHIT approach. I think it would be useful to further characterize these mutants for off-target cleavage effects and discuss what that might mean for false positive detection using the CoHIT approach. I would suggest doing this with any of the described live cell methods (ie – GuideSEQ) or cell free methods available in the literature. I also think that the authors should provide a convincing demonstration that the described mutants are truly necessary for this methodology to be successfully employed. Finally, I would suggest at minimum greatly reducing the protein engineering figures or relegating them to the supplemental entirely depending on what is discovered in follow-up experiments. I think with some revision this could be a really impactful paper with real significance in clinical diagnostics.

Authors: We appreciate the reviewer's insightful suggestions and have revised the manuscript to further address concerns regarding the potential off-target effects of the engineered Cas12a mutants and justify their necessity for the CoHIT methodology.

As mentioned above, to assess potential off-target effects, we used Cas-OFFinder to predict off-target sites for four target genes in the human genome. Subsequently, we experimentally validated 60 predicted off-target sites, finding no cross-reactions induced by the engineered Cas12a protein (Fig. S26 and Supplementary Table S3 and S4). Additionally, the specific gene amplification process in the CoHIT system minimizes the likelihood of off-target reactions. This was confirmed using human DNA samples and clinical samples (Fig. 4k-o). These findings collectively demonstrate the high specificity of the engineered enAsU-R Cas12a protein in the CoHIT assay.

Our goal is to detect multiple indel variants with a single crRNA with high sensitivity. Therefore, a Cas protein with high mismatch tolerance and cleavage efficiency is crucial. While the wild-type AsCas12a exhibited higher tolerance than LbCas12a, its cleavage efficiency decreases significantly at two and three mismatches (Supplementary Fig. S1). To enhance sensitivity, further engineering of AsCas12a was necessary. Notably, the enAsU-R variant enables highly sensitive and mismatch-tolerant CoHIT detection of multiple similar DNA variants with a single crRNA. Its mismatch tolerance is much better than other AsCas12a variants (Fig. 3f-h). Moreover, enAsU-R Cas12a has a wide and distinctive NNCV and TTTV PAM, which is not reported in other Cas12a variants (Fig. 3a). This wide PAM provides great convenience for designing and screening the best crRNA.

Following the reviewer's suggestions, we moved part of the protein engineering work to the supplementary materials and integrated the original Fig. 3 and Fig. 4 into one figure (Fig. 3) in the revised version. In addition, as mentioned above, we supplemented the application work by:

- 1) Adding MRD related experiments (Fig. 6d-g, Supplementary Fig. 22-24). We detected a succession of cryopreserved DNA of bone marrow samples belonging to three AML patients (Patients-MRD-1~3), ranging from initial diagnosis through remission to recurrence during follow-up visits. The results showed that CoHIT detection can provide early warning of leukaemia recurrence.
- 2) Combining the CoHIT system with lateral flow strip assay (LFA), which is beneficial for application in underdeveloped areas (Fig. 7a-e).
- 3) Combining the CoHIT system with a microfluidic chip to achieve multiplexing detection (Fig. 7f-i).

4) Expanding the clinical detection scale from 70 patients to 108 patients.

We believe these revisions will significantly improve the manuscript's focus and emphasize the clinical significance of our work.

Thank the reviewer again for the thorough review and valuable feedback on our manuscript. The reviewer's comments have provided insightful perspectives that will undoubtedly enhance the quality and impact of our research.

References

Cavo M, San-Miguel J, Usmani SZ, et al. Prognostic value of minimal residual disease negativity in myeloma: combined analysis of POLLUX, CASTOR, ALCYONE, and MAIA. *Blood*. 2022;139(6):835-844. doi:10.1182/blood.2021011101

Pellini B, Chaudhuri AA. Circulating Tumor DNA Minimal Residual Disease Detection of Non-Small-Cell Lung Cancer Treated With Curative Intent. *J Clin Oncol*. 2022;40(6):567-575. doi:10.1200/JCO.21.01929

Heuser M, Freeman SD, Ossenkoppele GJ, et al. 2021 Update on MRD in acute myeloid leukemia: a consensus document from the European LeukemiaNet MRD Working Party. *Blood*. 2021;138(26):2753-2767. doi:10.1182/blood.2021013626

Eisenstein M. Seven technologies to watch in 2022. *Nature*. 2022;601(7894):658-661. doi:10.1038/d41586-022-00163-x

Response to Reviewer #2

Reviewer #2 (Remarks to the Author):

In the submitted manuscript, the authors reported the development of a CRISPR-based diagnostic assay for the detection of cancer mutations. The idea is interesting and relies on the fact that AsCas12a variants can tolerate some mismatches at specific positions along their binding site to detect the presence of indels. Usually, indels are detected by loss of signal as it is difficult to design one set of primers or probes against all possible combinations of indels and thus it is easier to design against the wildtype sequence instead. But in the authors' assay, indels are detected by gain of signal, which is more unusual.

Authors: Thanks for the reviewer's professional review work on our manuscript. We are grateful for the reviewer's positive feedback.

Nevertheless, while the work is interesting from an academic point of view, I am not entirely convinced of its real-world applicability. A major issue is that detection of cancer mutations is not extremely time critical (unlike infectious diseases like the COVID-19 pandemic). Generally, it will not matter if a cancer is diagnosed in 30 minutes or a few days. In the introduction, the authors tried to give hyperleukocytosis as an example. But the pathophysiology of hyperleukocytosis is not well understood, so it is impossible to create a CRISPR assay for it. Hence, I believe that clinicians would prefer to rely on methods like next generation sequencing (NGS) or real-time PCR. Moreover, CRISPR-based assays are challenging to multiplex and thus cannot serve as a general screening tool for multiple types of mutation. Consequently, the authors might want to consider alternative applications of their method.

Authors: Thanks for the reviewer's thoughtful comments. We agree with the reviewer that the example of hyperleukocytosis is inappropriate, thus we have deleted it in our revised manuscript. In fact, our Cas12a-based CoHIT method is not only rapid but also sensitive, specific, convenient, cost-effective and easy to popularize (Nature. 2022;601(7894):658-661).

References	Original text
Nature. 2022;601(7894):658-661.	Seven technologies to watch in 2022 Other Cas enzymes could flesh out the diagnostic toolbox, Doudna notes, including the Cas12 proteins, which exhibit similar properties to Cas13 but target DNA rather than RNA. Collectively, these could detect a broader range of pathogens, or even enable efficient diagnosis of other non-infectious diseases. "That could be very useful if you could do that relatively quickly, especially as different cancer subtypes become defined by particular types of mutations," Doudna says.

For cancer patients, combating cancer is generally a long-term process and usually causes economic difficulties. On the one hand, a convenient and cost-effective method is more suitable for regular follow-up and disease monitoring. On the other hand, for patients in underdeveloped areas, an easy-to-popularize method will benefit more patients and reduce their travel cost. Moreover, a fast diagnostic method will shorten hospitalization time and help timely efficacy evaluation. We agree with the reviewer that current clinicians usually rely on methods like NGS or real-time PCR, however, NGS has significant limitations in terms of expense, turnaround time, instrument dependence, and requirement for highly trained operators or analysts. Real-time PCR also need expensive instruments and complex operations, and can not use one probe to target multiple variants.

We agree that supplementing the application work would benefit our article. In the revised manuscript, we have expanded the clinical detection scale from 70 patients to 108 patients, and added three parts of experiments:

- 1) Added minimal residual disease (MRD)-related experiments through testing clinical samples. The measurement of MRD levels, which indicate the remaining quantity of malignant cells in the body, has important prognostic and therapeutic implications (Blood. 2022;139(6):835-844) (J Clin Oncol. 2022;40(6):567-575). Considering that NPM1 gene mutation is a clinically recommended MRD testing item (Haematologica. 2022;107(12):2810-2822), we tested the CoHIT system to detect residual

leukaemia in NPM1-mutated AML patients. As shown in the following figures (Fig. 6d-g, Supplementary Fig. 22-24), we collected a succession of cryopreserved DNA of bone marrow samples belonging to three AML patients (Patients-MRD-1~3), ranging from initial diagnosis through remission to recurrence during follow-up visits. These samples were detected using FGS, NGS, and our CoHIT method at the same time. The results showed that the detection outcomes of CoHIT is consistent with NGS. Moreover, the CoHIT method can detect low abundance mutations earlier than FGS, as shown by the samples of Patient-MRD-1 (dated 06/2022 and 10/2022) and Patient-MRD-2 (dated 03/2021). Therefore, CoHIT detection can provide earlier warning of leukaemia recurrence. Based on its benefits in terms of speed, sensitivity, convenience, and cost-effectiveness, we believe that the CoHIT method has the potential to become a new MRD monitoring method, particularly offering enhanced convenience to patients in underdeveloped areas.

Fig. 6 | Detection of clinical samples and MRD using the CoHIT system. d Flowchart of the MRD experiment. **e** CoHIT result for Patient-MRD-1. The sampling dates and naked-eye results are shown for each sample. The asterisks indicate P values compared with healthy WT control (HC). **f** NGS and FGS results of Patient-MRD-1 samples dated 10/2021, 06/2022, 10/2022, and 01/2023. **g** CoHIT and NGS results for Patient-MRD-2 and 3. The asterisks indicate P values obtained by comparison with HC in CoHIT detection. The percentages are the insTCTG mutation rates determined by NGS.

Fig. S22 | FGS chromatograms of the eight bone marrow samples of Patient-MRD-1, collected between 10/2021 (First visit) and 01/2023. Mutant peaks are underlined in red. The percentages near the dates are NGS results of NPM1 gene c.863_864insTCTG mutation.

Fig. S23 | FGS, NGS, and CoHIT detection results of Patient-MRD-2. **a** Six bone marrow samples were collected between 04/2020 (First visit) and 07/2021 and tested by CoHIT assay. Final fluorescence values and naked-eye photos of the six samples are shown in chronological order. The asterisks indicate P values compared with healthy WT control (HC). **b** Time course of CoHIT detection of the six samples and HC. **c** FGS chromatograms of the six samples. Mutant peaks are underlined in red. The percentages near the dates are NGS results of the NPM1 gene c.863_864insTCTG mutation.

Fig. S24 | FGS, NGS, and CoHIT detection results of Patient-MRD-3. **a** Three bone marrow samples were collected between 10/2021 (First visit) and 06/2022 and tested by CoHIT assay. Final fluorescence values and naked-eye photos of the three samples were shown in chronological order. The asterisks indicate P values compared with healthy WT control (HC). **b** Time-course of CoHIT detection of the three samples and HC. **c** FGS chromatograms of the three samples. Mutant peaks are underlined in red. The percentages near the dates are NGS results of the NPM1 gene c.863_864insTCTG mutation.

2) Combine the CoHIT system with lateral flow strip assay (LFA), which is beneficial for application in underdeveloped areas. As shown in Fig. 7a-e, by replacing FAM-ssDNA-BHQ1 reporters with biotin-ssDNA-FAM reporters in the CoHIT system, the cleaved ssDNA would be captured on the test line, suggesting a positive result. After optimization of the biotin-ssDNA-FAM concentration and CoHIT reaction time, we used the CoHIT-LFA system to detect ten patient samples. As shown in Fig. 7e, the CoHIT-LFA method successfully distinguished between positive and negative patient samples.

Fig. 7 | CoHIT-LFA system and microfluidic chip-based CoHIT detection. **a** Detection principle of the LFA strip for the CoHIT system. **b** Biotin-ssDNA-FAM concentration optimization for the LFA assay. **c** Flow chart of the CoHIT-LFA assay. **d** Optimization of the CoHIT reaction time for the CoHIT-LFA assay. W, WT sample. M, mutant sample. **e** Detection of the NPM1 gene c.863_864 4-bp insertion in ten AML patient samples. The percentages are mutation rates.

3) Combine the CoHIT assay with a microfluidic chip to achieve multiplexing detection. As shown in Fig. 7f-i and Supplementary Fig. S25, we made a microfluidic device with one loading chamber and nine reaction chambers. Mutation status of seven sites can be obtained simultaneously through a single loading step. To verify this point, we tested the NPM1 mutation site (Target 1) and six FLT3-ITD (internal tandem duplication) mutation sites (Targets 2~7), which are all hotspot mutations in AML. The results showed that each target could be effectively detected and the sensitivity could reach 0.1% under naked eye. Further experiments on clinical samples proved that it can simultaneously detect mutations at different sites, P85 in Fig. 7i for example.

Fig. 7 | CoHIT-LFA system and microfluidic chip-based CoHIT detection. **f** Flow chart of the microfluidic chip-based multiplexing CoHIT detection. IC, inner control. **g** Detection of 1E6 copies of mutated fragments-containing plasmid in 100 ng WT genomic DNA. **h** Detection of Target 2 at frequencies of 10%, 1%, and 0.1%. **i** Detection in clinical samples from six AML patients.

Target	Gene mutation
1	NPM1 c.863_864 4-bp insertion
2	FLT3 c.1770_1771 24-bp insertion
3	FLT3 c.1793_1794 21-bp insertion
4	FLT3 c.1790_1791 18-bp insertion
5	FLT3 c.1776_1777 18-bp insertion
6	FLT3 c.1784_1785 21-bp insertion
7	FLT3 c.1796_1797 27-bp insertion
IC	GAPDH WT

Fig. S25 | Gene mutation sites and types of targets 1~7 in the microfluidic chip-based multiplexing CoHIT assay.

We agree that CRISPR-based assays may not serve as a large-scale screening tool. However, genetic testing is needed not only in the first visit but also in regular follow-up, efficacy evaluation, resistance mutation detection, and MRD monitoring. For cancer patients with known genetic information, it will be more cost-effective to detect their personal mutation markers instead of repetitive NGS panel test. Considering that the CoHIT method is an isothermal one-pot assay, it will be lower-cost, easier-to-use, and more advantageous for the promotion. We believe that each genotyping method has its advantages and disadvantages, and using different methods in different situations will maximize the benefits for cancer patients in clinical practice.

Major comments:

1) Can the method work for small (1bp or 2bp) indels? Larger indels (like 4bp) are easier to detect and distinguish from wildtype.

Authors: Thanks for the comment. The reviewer raises a good question that we can more thoroughly describe. To prove that CoHIT method can work for small (2-bp or 1-bp) indels, we synthesized and tested three kind of targets, including one 2-bp-deletion target (Target 1), one 1-bp-insertion target (Target 2), and one 1-bp-deletion target (Target 3). Each target has three or four indel variants (Supplementary Fig. S13). After screening ERA amplification primers for each target, we used CoHIT assay to detect these indels. As shown in the following figure, for all the three targets, multiple indel variants could be sensitively detected, without obvious WT-induced cross signal. In fact, although CoHIT system could tolerate several mismatch bases, it has a low tolerance for indels. This is why CoHIT system could avoid WT-induced cross signal and detect multiple indel variants at the same time. We have added these experiments in our revised manuscript.

Fig. S13 | Detect 2-bp and 1-bp indels using the CoHIT system. Sequences of the crRNAs and the DNA templates, naked-eye results, and time course of fluorescence changes for Target 1 (a), Target 2 (b), and Target 3 (c). Indel bases and PAMs in DNA sequences are marked in red and green, respectively.

2) For the NPM1 target, why don't the authors use or test a crRNA that hybridises to the TCTG insertion perfectly since it is most common (fig 4a)? I would imagine testing a crRNA that recognises the most common insertion (and then checking whether it would work against other insertions) to be a natural thing to do.

Authors: We thank the reviewer for this comment and apologize for the confusion. We used the CCTG-crRNA because it has less than one mismatch base with top six NPM1 c.863_864 4-bp insertions (insTCTG, CATG, CCTG, CCAG, CCGG, and CTTG), while TCTG-crRNA has two mismatch bases with four of them (Supplementary Fig. S3a). Therefore, we believe that CCTG-crRNA can detect the top six insertions with stronger fluorescence signals. To prove this, we synthesized TCTG-crRNA1 and compared it with CCTG-crRNA1. As shown in the following figure (Supplementary Fig. S3b), CCTG-crRNA1 and TCTG-crRNA1 have induced comparable fluorescence intensity on the most common TCTG insertion. Moreover, CCTG-crRNA1 has induced higher fluorescence intensity than TCTG-crRNA1 on CATG, CCAG, CCGG, and CTTG insertions. We agree with the reviewer that testing a crRNA that recognises the most common TCTG insertion is a natural thing to do, and it is necessary to prove to readers the rationality of our adoption of CCTG-crRNA1. We have added this experiment and relevant explanation in our revised manuscript:

“Considering the minimal nucleotide divergence of insCCTG from the six most frequent insertion types, we designed crRNAs (NPM1-crRNA1~6) complementary to the insCCTG sequence rather than the most common insTCTG sequence, to obtain stronger fluorescence signals for multiple frequent insertions (Fig. 3d, Supplementary Fig. S3).” (*The Results section, Line 166 to 169*)

Fig. S3 | Comparison between CCTG-crRNA1 and TCTG-crRNA1 to detect top six NPM1 c.863_864 4-bp insertions. a Numbers of mismatch bases between the two crRNAs and the top six insertions. **b** Statistical chart of fluorescence values of enAsU-R-induced in vitro cleavage assay with the two crRNAs, using 2E10 copies of DNA fragments of the top six insertions as substrates, reacting for 15 min at 37°C.

3) In the last results section, I find it odd that the authors titrated input template amounts using a single patient sample and then generalised. Fluorescence intensity would depend on gene expression level and mutation load. Can the authors test a few different patient samples with varying gene expression levels and mutation loads?

Authors: We apologize for the confusion and thank the reviewer for providing the helpful suggestion. In the revised version, we have added experiments using genomic DNA from three NPM1-mutated AML patients with different insertion types and mutant frequencies, including P9 with c.863_864 insTCTG (12.3%), P33 with c.863_864 insCATG (39.9%), and P111 with c.863_864 insTCTG (44.0%). The DNA input assay showed that 50~250 ng genomic DNA input are suitable for CoHIT detection (Fig. 6a, Supplementary Fig. S14).

Fig. 6 | Detection of clinical samples and MRD using the CoHIT system. **a** Template input assay of the CoHIT system using blood genomic DNA of three NPM1-mutated AML patients, including P9 with 12.3% c.863_864insTCTG, P33 with 39.9% insCATG, and P111 with 44.0% insTCTG.

Fig. S14 | Template input assay of the CoHIT system. **a** Time course of CoHIT detection using 0~1000 ng genomic DNA of P9, P33, and P111. **b** Statistical chart of the final fluorescence intensities after 30 min of reaction at 39°C.

4) As mentioned above, cancer mutation detection might not be the best use case. Can the authors show another application that genuinely requires fast turnaround time? I am not convinced that identification of a 4bp insertion in NPM1 for AML requires a 30min turnaround time. There should be rigorous benchmarking against NGS and qPCR. Notably, targeted amplicon sequencing on Illumina can readily achieve 0.1% mutation detection rate and can further multiplex to cover many oncogenes and tumor suppressors. In addition, the turnover time is even faster for nanopore sequencing, which can also achieve 0.1% VAF (variant allele frequency) detection.

Authors: We appreciate the reviewer's suggestion to explore additional applications. In order to prove that CoHIT system can also be applied in pathogen detection, we used it to detect simulated DNA samples of five reported SARS-CoV-2 variants (V1~V5), all of which have a p.D215AAGY indel mutation in the spike gene and have changed their immune epitopes (Blood. 2020 Dec 17;136(25):2905-2917, <https://ngdc.cncb.ac.cn/ncov/knowledge/mutation>). As shown in the following figure, five indel variants were sensitively detected by the CoHIT assay using a single crRNA, which has a sequence complementary to V3 (UAAUUUCUACUAAGUGUAGAUGUGCGUGCGGCAGGCUAUCUC). At the same time, WT DNA of SARS-CoV-2 showed no observable cross signal.

Figure | Detection of simulated DNA samples of SARS-CoV-2 variants using the CoHIT assay. a Amino acid mutation and genomic DNA changes of the five SARS-CoV-2 indel variants. **b** Time course of CoHIT detection of the five variants and WT, using 1E5 copies of plasmids carrying the SARS-CoV-2 DNA fragments as templates, the ERA primer sequences were: forward primer-TTATTTTAAAATATATTCTAAGCACACGCC, reverse primer-CCAATGGTTCTAAGCCGAAAAACCTGAG, reacting for 30 min at 39°C.

We agree that a turnaround time of 30 min or a few days may not have a significant impact on the cancer progression. But as we mentioned above, combating cancer is generally a long-term process, and genetic testing is needed not only in the first visit but also in regular follow-up, efficacy evaluation, resistance mutation detection, and MRD monitoring. NGS has significant limitations in terms of expense, turnaround time, instrument dependence, and requirement for highly trained operators or analysts. qPCR also need expensive instruments and complex operations, and can not use one probe to target multiple variants. As for nanopore sequencing, it is fast but error-prone for small indels, need large data analyzer and highly trained analysts, and requires a high DNA input amount (microgram level) if without PCR amplification step (Nat Methods. 2023 Oct;20(10):1483-1492., Front Genet. 2023 Sep 14;14:1169868). Compared with traditional methods, CRISPR-based diagnostics have many advantages for cancer patients (Supplementary Fig. S27) (References are shown in the table below). We believe that using different methods in different situations will maximize the benefits for cancer patients in clinical practice.

References	Original text
Nature. 2022;601(7894):658-661.	Seven technologies to watch in 2022 Other Cas enzymes could flesh out the diagnostic toolbox, Doudna notes, including the Cas12 proteins, which exhibit similar properties to Cas13 but target DNA rather than RNA. Collectively, these could detect a broader range of pathogens, or even enable efficient diagnosis of other non-infectious diseases. "That could be very useful if you could do that relatively quickly, especially as different cancer subtypes become defined by particular types of mutations," Doudna says.
Trends Biotechnol. 2022 Nov;40(11):1326-1345.	Compared with traditional detection methods, such as PCR and ELISA, CRISPR detection technology does not require expensive instruments and experienced operators. For example, CRISPR technology can detect circulating tumor DNA (ctDNA) and miRNA in human circulation. Cas12a detection systems for EGFR790 and EGFR858 can be used to assess cancer progression.
Nat Rev Cancer. 2022 May;22(5):259-279.	In addition, Cas12 and Cas13-mediated detection of nucleic acids via specific high-sensitivity enzymatic reporter unlocking (SHERLOCK) and DNA endonuclease-targeted CRISPR trans reporter (DETECTR) has been used to identify cancer-associated mutations in tumour biopsy samples from patients. ... Thus, it is foreseeable that CRISPR technologies could serve as a personalized, sensitive detection and monitoring system for patients with cancer.
Nat Biomed Eng. 2021 Jul;5(7):643-656.	CRISPR-based diagnostics could facilitate the monitoring of genetic markers indicative of treatment response, such as mutations in the BRAF gene, which are commonly used to inform the treatment of melanoma skin cancer.

Comparison between different genotyping methods

	FGS	NGS	qPCR	ddPCR	CoHIT
Mutation LoD (%)	10 [1]	0.01 [2]	0.01	0.001	0.01
Turnaround time	~ 1 d	> 5 d	~ 2 h	4 h [3]	< 30 min
Instrument cost	+++	+++	++	+++	+
Operational complexity	+++	+++	++	+++	+
Result analysis difficulty	+++	+++	++	++	+
Cost per sample	+	+++	++	++	+

[1] J Med Virol. 2020 Dec;92(12):3604-3608.

[2] Lancet. 2023 Jun 17;401(10393):2073-2086.

[3] Nat Biomed Eng. 2017;1:714-723.

* +, ++, and +++ means low, moderate, and high.

Fig. S27 | Comparison between different genotyping methods.

Minor comments:

5) Please indicate that "OS" stands for original sequence in the main text

Authors: Thanks for the reviewer's kind reminder. We have added an explanation of "original sequence (OS)" in our revised manuscript, in *the Result section, Line 124*.

6) CoHIT does not detect all variants (definitely not single nucleotide substitutions). Hence, the title and abstract are misleading, and the authors should make clear that their assay is for indels only.

Authors: Thanks for the reviewer's kind reminder. We apologize for this confusion and have changed our title to "CoHIT: A one-pot ultrasensitive ERA-CRISPR system for detecting multiple same-site indels in cancer", as well as some words in the text that may mislead readers.

7) Please indicate that “FGS” stands for first generation sequencing such as Sanger sequencing in the main text. Where does the claim of 10% rate for FGS come from? It is still possible to quantify 5-10% from Sanger sequencing chromatograms.

Authors: Thanks for the reviewer’s kind reminder. In the revised main text, we have added the indication of “FGS, also known as Sanger sequencing” in *the Introduction section, Line 59*. For the limit of detection (LoD) of Sanger sequencing, we found several different opinions (References are shown in the table below), and we cited a reference in our revised manuscript (J Med Virol. 2020 Dec;92(12):3604-3608).

LoD	References	Original text
15%	Clin Chim Acta. 2022 Nov 1;536:98-103.	Sanger sequencing can identify a full range of EGFR mutations but is limited in sensitivity to approximately 15 %.
10%-20%	J Med Virol. 2020 Dec;92(12):3604-3608.	Sanger sequencing has traditionally been the gold standard in testing for somatic mutations. One of the limitations of Sanger sequencing is its limit of detection (~10%–20% mutant-type [MT] allele in a background of wild-type [WT])
20%	J Clin Med. 2020 Jan 19;9(1):271.	Sanger sequencing is the most used method with a limitation due to its poor limit of detection (~20%)
15-20%	J Hematol Oncol. 2019 Dec 5;12(1):131	However, Sanger sequencing has limited sensitivity (it cannot robustly identify mutations present in less than 15–20% of transcripts)
10%	Clin Chim Acta. 2016 Jun 1;457:75-80.	The sensitivities of HRM and conventional Sanger sequencing to detect mutations were 5% and 10%, respectively.

8) The persistent claim of one crRNA to identify numerous gene variants (indels) is misleading. The authors’ assay can effectively only check one mutation site at a time.

Authors: Thank the reviewer for pointing it out. The reviewer is correct, we have changed the misleading words and repeatedly mentioned “same-site”, “at same genetic locus”, and “at the same gene site” in our revised manuscript.

9) There is a typo on line 241: “two important variants: c. 1459_1473del and c.1459_1473del” (they are the same).

Authors: Thank the reviewer for pointing it out. We have changed it to “two important variants, c. 1459_1473del and c.1460_1474del” in our revised version.

Thank the reviewer again for taking time and effort to thoroughly read our manuscript, and for providing these insightful comments and valuable suggestions, which significantly help us to further improve the quality of our manuscript.

References

Eisenstein M. Seven technologies to watch in 2022. *Nature*. 2022;601(7894):658-661. doi:10.1038/d41586-022-00163-x

Cavo M, San-Miguel J, Usmani SZ, et al. Prognostic value of minimal residual disease negativity in myeloma: combined analysis of POLLUX, CASTOR, ALCYONE, and MAIA. *Blood*. 2022;139(6):835-844. doi:10.1182/blood.2021011101

Pellini B, Chaudhuri AA. Circulating Tumor DNA Minimal Residual Disease Detection of Non-Small-Cell Lung Cancer Treated With Curative Intent. *J Clin Oncol*. 2022;40(6):567-575. doi:10.1200/JCO.21.01929

Blachly JS, Walter RB, Hourigan CS. The present and future of measurable residual disease testing in acute myeloid leukemia. *Haematologica*. 2022;107(12):2810-2822. Published 2022 Dec 1. doi:10.3324/haematol.2022.282034

- Li L, Shen G, Wu M, Jiang J, Xia Q, Lin P. CRISPR-Cas-mediated diagnostics. *Trends Biotechnol.* 2022;40(11):1326-1345. doi:10.1016/j.tibtech.2022.04.006
- Katti A, Diaz BJ, Caragine CM, Sanjana NE, Dow LE. CRISPR in cancer biology and therapy. *Nat Rev Cancer.* 2022;22(5):259-279. doi:10.1038/s41568-022-00441-w
- Kaminski MM, Abudayyeh OO, Gootenberg JS, Zhang F, Collins JJ. CRISPR-based diagnostics. *Nat Biomed Eng.* 2021;5(7):643-656. doi:10.1038/s41551-021-00760-7
- Kolmogorov M, Billingsley KJ, Mastoras M, et al. Scalable Nanopore sequencing of human genomes provides a comprehensive view of haplotype-resolved variation and methylation. *Nat Methods.* 2023;20(10):1483-1492. doi:10.1038/s41592-023-01993-x
- Wen X, Du J, Li Z, et al. Establishment of linkage phase, using Oxford Nanopore Technologies, for preimplantation genetic testing of Coffin-Lowry syndrome with a de novo RPS6KA3 mutation. *Front Genet.* 2023;14:1169868. Published 2023 Sep 14. doi:10.3389/fgene.2023.1169868
- Jiang H, Chen X, Huang F, et al. Validation of a highly sensitive Sanger sequencing in detecting EGFR mutations from circulating tumor DNA in patients with lung cancers. *Clin Chim Acta.* 2022 Nov 1;536:98-103. doi: 10.1016/j.cca.2022.08.030.
- Akuta N, Suzuki F, Kobayashi M, et al. Detection of TERT promoter mutation in serum cell-free DNA using wild-type blocking PCR combined with Sanger sequencing in hepatocellular carcinoma. *J Med Virol.* 2020 Dec;92(12):3604-3608. doi: 10.1002/jmv.25724.
- Petiti J, Rosso V, Croce E, et al. Highly Sensitive Detection of IDH2 Mutations in Acute Myeloid Leukemia. *J Clin Med.* 2020 Jan 19;9(1):271. doi: 10.3390/jcm9010271.
- Soverini S, Abruzzese E, Bocchia M, et al. Next-generation sequencing for BCR-ABL1 kinase domain mutation testing in patients with chronic myeloid leukemia: a position paper. *J Hematol Oncol.* 2019 Dec 5;12(1):131. doi: 10.1186/s13045-019-0815-5.
- Ishige T, Itoga S, Matsushita K, et al. Locked nucleic acid probe enhances Sanger sequencing sensitivity and improves diagnostic accuracy of high-resolution melting-based KRAS mutational analysis. *Clin Chim Acta.* 2016 Jun 1;457:75-80. doi: 10.1016/j.cca.2016.04.005.

Response to Reviewer #3

Reviewer #3 (Remarks to the Author):

In this manuscript the authors use Cas12a to detect mutations in genes where there are multiple possible different nucleotide sequences, as in the case of indels that define NPM1 mutations. The outlined approach could be broadly applicable across various diseases defined by multiple mutation possibilities at the same site, and they showed the ability to use their system for other mutations as well.

Authors: Thank the reviewer for taking time and effort to thoroughly read our manuscript and for providing the thoughtful comments.

The approaches seem robust however the main concern is the novelty and likelihood this approach will truly find a place in clinical diagnostics. The authors approach ties together several approaches that were recently novel but in and of itself the approach is not.

There are multiple examples of using Cas12a approaches to detect genetic variants. This flourished in the COVID literature:

- Ding X, Yin K, Li Z, Lalla RV, Ballesteros E, Sfeir MM, Liu C. Ultrasensitive and visual detection of SARS-CoV-2 using all-in-one dual CRISPR-Cas12a assay. *Nat Commun.* 2020 Sep 18;11(1):4711. doi: 10.1038/s41467-020-18575-6. PMID: 32948757; PMCID: PMC7501862.

- Liang Y, Zou L, Lin H, Li B, Zhao J, Wang H, Sun J, Chen J, Mo Y, Yang X, Deng X, Tang S. Detection of Major SARS-CoV-2 Variants of Concern in Clinical Samples via CRISPR-Cas12a-Mediated Mutation-Specific Assay. *ACS Synth Biol.* 2022 May 20;11(5):1811-1823. doi: 10.1021/acssynbio.1c00643. Epub 2022 Apr 28. PMID: 35481381.

- Fasching CL, Servellita V, McKay B, Nagesh V, Broughton JP, Sotomayor-Gonzalez A, Wang B, Brazer N, Reyes K, Streithorst J, Deraney RN, Stanfield E, Hendriks CG, Fung B, Miller S, Ching J, Chen JS, Chiu CY. COVID-19 Variant Detection with a High-Fidelity CRISPR-Cas12 Enzyme. *J Clin Microbiol.* 2022 Jul 20;60(7):e0026122. doi: 10.1128/jcm.00261-22. Epub 2022 Jun 29. PMID: 35766492; PMCID: PMC9297821.

Cas12a-based methods for variant detection have further expanded into cancers as well, with this article describing a rapid, highly sensitive and specific, Cas12a-based method for detection of FLT3 D835Y/H/V/F mutations:

- Liu Y, Chen Y, Dang L, et al. EasyCatch, a convenient, sensitive and specific CRISPR detection system for cancer gene mutations. *Mol Cancer.* 2021;20(1):157. Published 2021 Dec 2. doi:10.1186/s12943-021-01456-x

Strengths:

- This work is potentially relevant to NPM1 mutation identification. They picked a good disease model for their test. This problem is that it is not so clear that the approach is drastically better than current mutation identification approaches.

Authors: We appreciate the reviewer's acknowledgment of the CoHIT system's contribution to NPM1 mutation identification and his/her recognition of the robustness of our approach.

Our gratitude extends to the reviewer's professional evaluation of relevant CRISPR detection studies. Notable works, such as Ding X, et al (*Nat Commun.* 2020), integrates isothermal amplification with CRISPR reactions for user-friendly application. Studies by Liang Y, et al (*ACS Synth Biol.* 2022) and Fasching CL, et al (*J Clin Microbiol*), as well as our prior work (Liu., et al., *Mol Cancer.* 2021), utilize the high specificity of Cas12a for specific detection down to a single nucleic acid variation. However, none of these studies attempt to address the challenge of detecting multiple mutations at a single gene site. To the best of our knowledge, "CoHIT" represents the first attempt to address the challenge of detecting hundreds of variants in a rapid, sensitive, and cost-effective way.

To establish the CoHIT system, we engineered and screened a novel Cas12a protein, enAsU-R, which is critical for the CoHIT technique because: 1) Its enhanced activity and mismatch tolerance allows the CoHIT system to achieve the full-coverage detection. 2) Its distinctive and broad NNCV and TTTV PAM makes it easy to design and screen crRNAs for most mutation sites in the human genome, thus greatly expands the application scope of the CoHIT system. As shown in our original manuscript, CoHIT achieved a remarkable limit of detection (LoD) of as low as 0.01% for different NPM1 gene variants with a single crRNA within 30 minutes. The novelty of our work was duly acknowledged by both reviewer 1, stating, "The application of this tool is both novel

and relevant to real challenges in cancer diagnostics and represents a novel application for Cas12a/Cas13a detection systems that have not been able to supplant qPCR and antigen tests for RNA pathogen detection," and reviewer 4, commending, "The authors should be congratulated for having done such excellent technical development work! The assay appears technically sound."

We value the reviewer's professional advice and have incorporated comprehensive comparisons with existing approaches to highlight the specific contributions of CoHIT to the field (Supplementary Fig. S27). These modifications aim to provide a clearer understanding of the significance of our work. Thanks for the reviewer's guidance, which has been instrumental in refining our manuscript.

Comparison between different genotyping methods

	FGS	NGS	qPCR	ddPCR	CoHIT
Mutation LoD (%)	10 [1]	0.01 [2]	0.01	0.001	0.01
Turnaround time	~ 1 d	> 5 d	~ 2 h	4 h [3]	< 30 min
Instrument cost	+++	+++	++	+++	+
Operational complexity	+++	+++	++	+++	+
Result analysis difficulty	+++	+++	++	++	+
Cost per sample	+	+++	++	++	+

[1] J Med Virol. 2020 Dec;92(12):3604-3608.

[2] Lancet. 2023 Jun 17;401(10393):2073-2086.

[3] Nat Biomed Eng. 2017;1:714-723.

* +, ++, and +++ means low, moderate, and high.

Fig. S27 | Comparison between different genotyping methods.

- The fact that they engineered Cas12a mutants with more permissive PAM recognition to allow for better detection of mutants is a strength. They did a good job creating Cas12a mutations that will identify most of the major NPM1 mutations. That said, other groups have worked on developing Cas9 and Cas12a mutants that allow for more permissive PAMs for various functions. But the fact this group created several mutants and screened them to improve for tolerance of mismatches is well done.

Weaknesses:

- Style and writing: Significant editing is needed.

Authors: We are grateful for the positive feedback of our work. According to the reviewer's suggestion, our revised manuscript has been reviewed and edited by the language services of Springer Nature. The certificate can be verified on the SNAS website using the verification code 836E-E936-6F66-4B2F-D42C.

- Introduction/framing of their work: Their introduction to why detection of mutations is important, while brief, needs work. One of the sentences is: “Moreover, early detection of low-frequency mutations can help prevent cancer in patients.” I don’t think this is entirely true of current practice, though maybe in the future with cell free DNA. They do not offer an example or a citation for this either. The other background importance for mutation detection that they give is that it helps clinicians diagnose, select treatment, and monitor for recurrence. They cite a review article and the NCCN guidelines, which is pretty vague.

Authors: Thank the reviewer for the insightful comments. We agree with the reviewer that early detecting gene mutations for cancer prevention has not been widely promoted in current clinical practice, our statement may seem a bit radical. In fact, we originally intended to mean that many cancers evolve through the accumulation of genetic mutations, thus early detection of these mutations may help early intervention, which means secondary prevention (References are shown in the table below). In the revised manuscript, we have changed this sentence to “Moreover, the early detection of cancer-related mutations may provide opportunities for early intervention.” (*The Introduction section, Line 48-49*), and cited a reference (JAMA. 2017;318(9):825-835).

References	Original text
Ann Oncol. 2022 Dec;33(12):1239-1249.	The accumulation of cancer driver mutations long before the appearance of cancer represents both opportunities and challenges for cancer prevention and early detection.
Nat Med. 2018 Jul;24(7):1015-1023.	The presence of detectable mutations years before diagnosis suggests that there is a period of latency that precedes AML during which early detection, monitoring and interventional studies should be considered.
Science. 2017 Mar 24;355(6331):1330-1334.	Secondary prevention, i.e., early detection and intervention, can also be lifesaving. For cancers in which all mutations are the result of R, secondary prevention is the only option.
Nature. 2018 Jul;559(7714):400-404.	Therefore, genetic testing for ARCH may also prove useful in the management of common age-related diseases. Moreover, this study has broader implications for cancer screening and early intervention beyond AML.
JAMA. 2017 Sep 5;318(9):825-835	Knowledge of these additional mutations can help guide therapeutic and preventive interventions.

We apologize for the inappropriate citation of the review article or NCCN guideline. In the revised version, we have cited more specific documents to illustrate the importance of mutation detection in cancer diagnosis, treatment, and monitoring (References are shown in the table below).

References	Original text
Blood. 2023 Feb 2;141(5):534-549.	Our findings establish that MDS with DDX41-mutation defines a unique subtype of MNs that is distinct from other MNs.
Nat Med. 2019 May;25(5):744-750.	Our findings suggest that the current clinical trial paradigm for precision oncology, which pairs one driver mutation with one drug, may be optimized by treating molecularly complex and heterogeneous cancers with combinations of customized agents.
Ann Oncol. 2023 May;34(5):468-476.	The serial monitoring of ctDNA T790M status in advanced EGFR-mutant non-small-cell lung cancer during treatment with first-generation EGFR inhibitors was feasible.

- They refer to sensitivity to mean two different things. They mean it both in terms of a novel statistical test to rule out a mutation and then also the ability to detect low levels of a mutation. They should more clearly differentiated between these.

Authors: Thank the reviewer for pointing this out. We apologize for the confusion. In our revised manuscript, we use “Limit of detection (LoD)” to describe the ability to detect low levels of a mutation.

- In some of the figures and figure legends, it is unclear how much replication they did, as they only have solid bars and the figure legends don’t indicate number of replicates. It would be helpful to get a sense of the variability inherent in their assays (Figure 4E, for example). Also, clarifying what statistical comparisons between groups they were doing (and why) would be helpful.

Authors: Thank the reviewer for the suggestion. In the revised manuscript, we have shown replicate dots in each bar chart and clarified the data analysis methods in the Methods section.

All said much more validation for potential use as a clinical diagnostic would be very helpful since the approach is not novel.

Authors: Thank the reviewer for the thoughtful comment. We appreciate the reviewer’s recognition of the importance of validation for potential clinical diagnostic use. Firstly, the use of engineered Cas proteins for the detection of multiple variations is a novel approach. As mentioned above, to the best of our knowledge, "CoHIT" represents the first attempt to address the challenge of using a single crRNA to detect multiple same-site variants in a rapid, sensitive, and cost-effective way. To achieve this goal, we engineered and screened a novel Cas12a protein variant, enAsU-R, which has enhanced mismatch tolerance and a distinctive and broad NNCV and TTTV PAM. Secondly, in the revised version, we have added three part of experiments to demonstrate the potential clinical use of the CoHIT system:

- 1) Added minimal residual disease (MRD)-related experiments through testing clinical samples. The measurement of MRD levels, which indicate the remaining quantity of malignant cells in the body, has important prognostic and therapeutic implications (Blood. 2022;139(6):835-844) (J Clin Oncol. 2022;40(6):567-575). Considering that NPM1 gene mutation is a clinically recommended MRD testing item (Haematologica. 2022;107(12):2810-2822), we tested the CoHIT system to detect residual leukaemia in NPM1-mutated AML patients. As shown in the following figures (Fig. 6d-g, Supplementary Fig. 22-24), we collected a succession of cryopreserved DNA of bone marrow samples belonging to three AML patients (Patients-MRD-1~3), ranging from initial diagnosis through remission to recurrence during follow-up visits. These samples were detected using FGS, NGS, and our CoHIT method at the same time. The results showed that the detection outcomes of CoHIT is consistent with NGS. Moreover, the CoHIT method can detect low abundance mutations earlier than FGS, as shown by the samples of Patient-MRD-1 (dated 06/2022 and 10/2022) and Patient-MRD-2 (dated 03/2021). Therefore, CoHIT detection can provide earlier warning of leukaemia recurrence. Based on its benefits in terms of speed, sensitivity, convenience, and cost-effectiveness, we believe that the CoHIT method has the potential to become a new MRD monitoring method, particularly offering enhanced convenience to patients in underdeveloped areas.

Fig. 6 | Detection of clinical samples and MRD using the CoHIT system. d Flowchart of the MRD experiment. **e** CoHIT result for Patient-MRD-1. The sampling dates and naked-eye results are shown for each sample. The asterisks indicate P values compared with healthy WT control (HC). **f** NGS and FGS results of Patient-MRD-1 samples dated 10/2021, 06/2022, 10/2022, and 01/2023. **g** CoHIT and NGS results for Patient-MRD-2 and 3. The asterisks indicate P values obtained by comparison with HC in CoHIT detection. The percentages are the insTCTG mutation rates determined by NGS.

Fig. S22 | FGS chromatograms of the eight bone marrow samples of Patient-MRD-1, collected between 10/2021 (First visit) and 01/2023. Mutant peaks are underlined in red. The percentages near the dates are NGS results of NPM1 gene c.863_864insTCTG mutation.

Fig. S23 | FGS, NGS, and CoHIT detection results of Patient-MRD-2. **a** Six bone marrow samples were collected between 04/2020 (First visit) and 07/2021 and tested by CoHIT assay. Final fluorescence values and naked-eye photos of the six samples are shown in chronological order. The asterisks indicate P values compared with healthy WT control (HC). **b** Time course of CoHIT detection of the six samples and HC. **c** FGS chromatograms of the six samples. Mutant peaks are underlined in red. The percentages near the dates are NGS results of the NPM1 gene c.863_864insTCTG mutation.

Fig. S24 | FGS, NGS, and CoHIT detection results of Patient-MRD-3. **a** Three bone marrow samples were collected between 10/2021 (First visit) and 06/2022 and tested by CoHIT assay. Final fluorescence values and naked-eye photos of the three samples were shown in chronological order. The asterisks indicate P values compared with healthy WT control (HC). **b** Time-course of CoHIT detection of the three samples and HC. **c** FGS chromatograms of the three samples. Mutant peaks are underlined in red. The percentages near the dates are NGS results of NPM1 gene c.863_864insTCTG mutation.

2) Combine the CoHIT system with lateral flow strip assay (LFA), which is beneficial for application in underdeveloped areas. As shown in Fig. 7a-e, by replacing FAM-ssDNA-BHQ1 reporters with biotin-ssDNA-FAM reporters in the CoHIT system, the cleaved ssDNA would be captured on the test line, suggesting a positive result. After optimization of the biotin-ssDNA-FAM concentration and CoHIT reaction time, we used the CoHIT-LFA system to detect ten patient samples. As shown in Fig. 7e, the CoHIT-LFA method successfully distinguished between positive and negative patient samples.

Fig. 7 | CoHIT-LFA system and microfluidic chip-based CoHIT detection. **a** Detection principle of the LFA strip for the CoHIT system. **b** Biotin-ssDNA-FAM concentration optimization for the LFA assay. **c** Flow chart of the CoHIT-LFA assay. **d** Optimization of the CoHIT reaction time for the CoHIT-LFA assay. W, WT sample. M, mutant sample. **e** Detection of the NPM1 gene c.863_864 4-bp insertion in ten AML patient samples. The percentages are mutation rates.

3) Combine the CoHIT assay with a microfluidic chip to achieve multiplexing detection. As shown in Fig. 7f-i and Supplementary Fig. S25, we made a microfluidic device with one loading chamber and nine reaction chambers. Mutation status of seven sites can be obtained simultaneously through a single loading step. To verify this point, we tested the NPM1 mutation site (Target 1) and six FLT3-ITD (internal tandem duplication) mutation sites (Targets 2~7), which are all hotspot mutations in AML. The results showed that each target could be effectively detected and the sensitivity could reach 0.1% under naked eye. Further experiments on clinical samples proved that it can simultaneously detect mutations at different sites, P85 in Fig. 7i for example.

Fig. 7 | CoHIT-LFA system and microfluidic chip-based CoHIT detection. **f** Flow chart of the microfluidic chip-based multiplexing CoHIT detection. IC, inner control. **g** Detection of 1E6 copies of mutated fragments-containing plasmid in 100 ng WT genomic DNA. **h** Detection of Target 2 at frequencies of 10%, 1%, and 0.1%. **i** Detection in clinical samples from six AML patients.

Target	Gene mutation
1	NPM1 c.863_864 4-bp insertion
2	FLT3 c.1770_1771 24-bp insertion
3	FLT3 c.1793_1794 21-bp insertion
4	FLT3 c.1790_1791 18-bp insertion
5	FLT3 c.1776_1777 18-bp insertion
6	FLT3 c.1784_1785 21-bp insertion
7	FLT3 c.1796_1797 27-bp insertion
IC	GAPDH WT

Fig. S25 | Gene mutation sites and types of targets 1~7 in the microfluidic chip-based multiplexing CoHIT assay.

These results provide a solid foundation for considering the potential application of our approach in diagnostic settings.

Thank the reviewer again for taking time and effort to carefully review our manuscript. We believe our revised manuscript is clearer and stronger after addressing these concerns.

References

Acha-Sagredo A, Ganguli P, Ciccarelli FD. Somatic variation in normal tissues: friend or foe of cancer early detection? *Ann Oncol.* 2022 Dec;33(12):1239-1249. doi:10.1016/j.annonc.2022.09.156.

Desai P, Mencia-Trinchant N, Savenkov O, et al. Somatic mutations precede acute myeloid leukemia years before diagnosis. *Nat Med.* 2018;24(7):1015-1023. doi:10.1038/s41591-018-0081-z

Tomasetti C, Li L, Vogelstein B. Stem cell divisions, somatic mutations, cancer etiology, and cancer prevention. *Science.* 2017 Mar 24;355(6331):1330-1334. doi: 10.1126/science.aaf9011.

Abelson S, Collord G, Ng SWK, et al. Prediction of acute myeloid leukaemia risk in healthy individuals. *Nature.* 2018;559(7714):400-404. doi:10.1038/s41586-018-0317-6

Mandelker D, Zhang L, Kemel Y, et al. Mutation Detection in Patients With Advanced Cancer by Universal Sequencing of Cancer-Related Genes in Tumor and Normal DNA vs Guideline-Based Germline Testing [published correction appears in *JAMA.* 2018 Dec 11;320(22):2381]. *JAMA.* 2017;318(9):825-835. doi:10.1001/jama.2017.

Makishima H, Saiki R, Nannya Y, et al. Germ line DDX41 mutations define a unique subtype of myeloid neoplasms. *Blood.* 2023;141(5):534-549. doi:10.1182/blood.2022018221

Sicklick JK, Kato S, Okamura R, et al. Molecular profiling of cancer patients enables personalized combination therapy: the I-PREDICT study. *Nat Med.* 2019;25(5):744-750. doi:10.1038/s41591-019-0407-5

Remon J, Besse B, Aix SP, et al. Osimertinib treatment based on plasma T790M monitoring in patients with EGFR-mutant non-small-cell lung cancer (NSCLC): EORTC Lung Cancer Group 1613 APPLE phase II randomized clinical trial. *Ann Oncol.* 2023;34(5):468-476. doi:10.1016/j.annonc.2023.02.012

Cavo M, San-Miguel J, Usmani SZ, et al. Prognostic value of minimal residual disease negativity in myeloma: combined analysis of POLLUX, CASTOR, ALCYONE, and MAIA. *Blood.* 2022;139(6):835-844. doi:10.1182/blood.2021011101

Pellini B, Chaudhuri AA. Circulating Tumor DNA Minimal Residual Disease Detection of Non-Small-Cell Lung Cancer Treated With Curative Intent. *J Clin Oncol.* 2022;40(6):567-575. doi:10.1200/JCO.21.01929

Blachly JS, Walter RB, Hourigan CS. The present and future of measurable residual disease testing in acute myeloid leukemia. *Haematologica.* 2022;107(12):2810-2822. Published 2022 Dec 1. doi:10.3324/haematol.2022.282034

Response to Reviewer #4

Reviewer #4 (Remarks to the Author):

Liu et al report a novel, CRISPR-CAS12a based assay for mutation detection with the advantage of a good mismatch tolerance. Through a series of optimization, they could show this assay is sensitive and specific, and surpass first generation sequencing test, and argue that this assay may be applied to clinical samples. The authors should be congratulated for having done such an excellent technical development work! The assay appears technically sound. However, evidence of clinical utility is insufficient.

Authors: Many thanks to the reviewer for the kind comments and the on-point summary of our manuscript. We have been very much encouraged by the reviewer's commendations. We fully agree that supplementing the evidence of clinical utility would benefit our article. In the revised manuscript, we have expanded the clinical detection scale from 70 patients to 108 patients, and added three parts of experiments:

- 1) Added minimal residual disease (MRD)-related experiments through testing clinical samples. The measurement of MRD levels, which indicate the remaining quantity of malignant cells in the body, has important prognostic and therapeutic implications (Blood. 2022;139(6):835-844) (J Clin Oncol. 2022;40(6):567-575). Considering that NPM1 gene mutation is a clinically recommended MRD testing item (Haematologica. 2022;107(12):2810-2822), we tested the CoHIT system to detect residual leukaemia in NPM1-mutated AML patients. As shown in the following figures (Fig. 6d-g, Supplementary Fig. 22-24), we collected a succession of cryopreserved DNA of bone marrow samples belonging to three AML patients (Patients-MRD-1~3), ranging from initial diagnosis through remission to recurrence during follow-up visits. These samples were detected using FGS, NGS, and our CoHIT method at the same time. The results showed that the detection outcomes of CoHIT is consistent with NGS. Moreover, the CoHIT method can detect low abundance mutations earlier than FGS, as shown by the samples of Patient-MRD-1 (dated 06/2022 and 10/2022) and Patient-MRD-2 (dated 03/2021). Therefore, CoHIT detection can provide earlier warning of leukaemia recurrence. Based on its benefits in terms of speed, sensitivity, convenience, and cost-effectiveness, we believe that the CoHIT method has the potential to become a new MRD monitoring method, particularly offering enhanced convenience to patients in underdeveloped areas.

Fig. 6 | Detection of clinical samples and MRD using the CoHIT system. **d** Flowchart of the MRD experiment. **e** CoHIT result for Patient-MRD-1. The sampling dates and naked-eye results are shown for each sample. The asterisks indicate P values compared with healthy WT control (HC). **f** NGS and FGS results of Patient-MRD-1 samples dated 10/2021, 06/2022, 10/2022, and 01/2023. **g** CoHIT and NGS results for Patient-MRD-2 and 3. The asterisks indicate P values obtained by comparison with HC in CoHIT detection. The percentages are the insTCTG mutation rates determined by NGS.

Fig. S22 | FGS chromatograms of the eight bone marrow samples of Patient-MRD-1, collected between 10/2021 (First visit) and 01/2023. Mutant peaks are underlined in red. The percentages near the dates are NGS results of NPM1 gene c.863_864insTCTG mutation.

Fig. S23 | FGS, NGS, and CoHIT detection results of Patient-MRD-2. **a** Six bone marrow samples were collected between 04/2020 (First visit) and 07/2021 and tested by CoHIT assay. Final fluorescence values and naked-eye photos of the six samples are shown in chronological order. The asterisks indicate P values compared with healthy WT control (HC). **b** Time course of CoHIT detection of the six samples and HC. **c** FGS chromatograms of the six samples. Mutant peaks are underlined in red. The percentages near the dates are NGS results of the NPM1 gene c.863_864insTCTG mutation.

Fig. S24 | FGS, NGS, and CoHIT detection results of Patient-MRD-3. **a** Three bone marrow samples were collected between 10/2021 (First visit) and 06/2022 and tested by CoHIT assay. Final fluorescence values and naked-eye photos of the three samples were shown in chronological order. The asterisks indicate P values compared with healthy WT control (HC). **b** Time-course of CoHIT detection of the three samples and HC. **c** FGS chromatograms of the three samples. Mutant peaks are underlined in red. The percentages near the dates are NGS results of NPM1 gene c.863_864insTCTG mutation.

2) Combine the CoHIT system with lateral flow strip assay (LFA), which is beneficial for application in underdeveloped areas. As shown in Fig. 7a-e, by replacing FAM-ssDNA-BHQ1 reporters with biotin-ssDNA-FAM reporters in the CoHIT system, the cleaved ssDNA would be captured on the test line, suggesting a positive result. After optimization of the biotin-ssDNA-FAM concentration and CoHIT reaction time, we used the CoHIT-LFA system to detect ten patient samples. As shown in Fig. 7e, the CoHIT-LFA method successfully distinguished between positive and negative patient samples.

Fig. 7 | CoHIT-LFA system and microfluidic chip-based CoHIT detection. **a** Detection principle of the LFA strip for the CoHIT system. **b** Biotin-ssDNA-FAM concentration optimization for the LFA assay. **c** Flow chart of the CoHIT-LFA assay. **d** Optimization of the CoHIT reaction time for the CoHIT-LFA assay. W, WT sample. M, mutant sample. **e** Detection of the NPM1 gene c.863_864 4-bp insertion in ten AML patient samples. The percentages are mutation rates.

3) Combine the CoHIT assay with a microfluidic chip to achieve multiplexing detection. As shown in Fig. 7f-i and Supplementary Fig. S25, we made a microfluidic device with one loading chamber and nine reaction chambers. Mutation status of seven sites can be obtained simultaneously through a single loading step. To verify this point, we tested the NPM1 mutation site (Target 1) and six FLT3-ITD (internal tandem duplication) mutation sites (Targets 2~7), which are all hotspot mutations in AML. The results showed that each target could be effectively detected and the sensitivity could reach 0.1% under naked eye. Further experiments on clinical samples proved that it can simultaneously detect mutations at different sites, P85 in Fig. 7i for example.

Fig. 7 | CoHIT-LFA system and microfluidic chip-based CoHIT detection. **f** Flow chart of the microfluidic chip-based multiplexing CoHIT detection. IC, inner control. **g** Detection of 1E6 copies of mutated fragments-containing plasmid in 100 ng WT genomic DNA. **h** Detection of Target 2 at frequencies of 10%, 1%, and 0.1%. **i** Detection in clinical samples from six AML patients.

Target	Gene mutation
1	NPM1 c.863_864 4-bp insertion
2	FLT3 c.1770_1771 24-bp insertion
3	FLT3 c.1793_1794 21-bp insertion
4	FLT3 c.1790_1791 18-bp insertion
5	FLT3 c.1776_1777 18-bp insertion
6	FLT3 c.1784_1785 21-bp insertion
7	FLT3 c.1796_1797 27-bp insertion
IC	GAPDH WT

Fig. S25 | Gene mutation sites and types of targets 1~7 in the microfluidic chip-based multiplexing CoHIT assay.

An clinical assay can be used at diagnosis and/or disease monitoring. An ideal assay should be fast, sensitive/specific and/or cost-effective.

1) Although this Co-HIT assay is faster with a readout probably in hours, and it is more sensitive than first generation sequencing (which has been gradually phased out in clinical practice), I doubt it will surpass NGS as a diagnostic test. As they showed in this study, NGS reliably detected all NPM1 mutations. Cancer patients often carry multiple mutations (3-8 mutations in AML, 10-30 mutations in solid tumors), and NGS, either whole exome or targeted panel, has the benefit of providing all the information on numerous mutations at once (although often with weeks delay but this will certainly be shortened as bioinformatic pipelines improve). The authors haven't shown a multiplex approach of CO-HIT for detecting multiple mutations simultaneously with reasonable specificity and sensitivity. Even they could, the requirement of knowing the mutation site for assay design would preclude the detection of unknown mutations, making it impossible for screening at diagnosis.

Authors: Thank the reviewer for the insightful comment. As the reviewer mentioned, an ideal assay should be fast, sensitive/specific and/or cost-effective. CRISPR-based diagnostic tools, including our CoHIT system (Supplementary Fig. S27), have these advantages and the potential to become a helpful monitoring method for cancer patients (References are shown in the table below).

Comparison between different genotyping methods

	FGS	NGS	qPCR	ddPCR	CoHIT
Mutation LoD (%)	10 [1]	0.01 [2]	0.01	0.001	0.01
Turnaround time	~ 1 d	> 5 d	~ 2 h	4 h [3]	< 30 min
Instrument cost	+++	+++	++	+++	+
Operational complexity	+++	+++	++	+++	+
Result analysis difficulty	+++	+++	++	++	+
Cost per sample	+	+++	++	++	+

[1] J Med Virol. 2020 Dec;92(12):3604-3608.

[2] Lancet. 2023 Jun 17;401(10393):2073-2086.

[3] Nat Biomed Eng. 2017;1:714-723.

* +, ++, and +++ means low, moderate, and high.

Fig. S27 | Comparison between different genotyping methods.

References	Original text
Nature. 2022;601(7894):658-661.	Seven technologies to watch in 2022 Other Cas enzymes could flesh out the diagnostic toolbox, Doudna notes, including the Cas12 proteins, which exhibit similar properties to Cas13 but target DNA rather than RNA. Collectively, these could detect a broader range of pathogens, or even enable efficient diagnosis of other non-infectious diseases. "That could be very useful if you could do that relatively quickly, especially as different cancer subtypes become defined by particular types of mutations," Doudna says.
Trends Biotechnol. 2022 Nov;40(11):1326-1345.	Compared with traditional detection methods, such as PCR and ELISA, CRISPR detection technology does not require expensive instruments and experienced operators. For example, CRISPR technology can detect circulating tumor DNA (ctDNA) and miRNA in human circulation. Cas12a detection systems for EGFR790 and EGFR858 can be used to assess cancer progression.
Nat Rev Cancer. 2022 May;22(5):259-279.	In addition, Cas12 and Cas13-mediated detection of nucleic acids via specific high-sensitivity enzymatic reporter unlocking (SHERLOCK) and DNA endonuclease-targeted CRISPR trans reporter (DETECTR) has been used to identify cancer-associated mutations in tumour biopsy samples from patients. ... Thus, it is foreseeable that CRISPR technologies could serve as a personalized, sensitive detection and monitoring system for patients with cancer.
Nat Biomed Eng. 2021 Jul;5(7):643-656.	CRISPR-based diagnostics could facilitate the monitoring of genetic markers indicative of treatment response, such as mutations in the BRAF gene, which are commonly used to inform the treatment of melanoma skin cancer.

We agree with the reviewer that NGS performs well in multiplexing and screening. In fact, many hospitals including our department have carried out NGS panel test for years and frequently recommend it to patients, it dose provide great help in precise diagnosis of diseases. But we also find that only a small part of patients can afford the NGS test. For cancer patients, combating cancer is generally a long-term process and usually causes economic difficulties. NGS has its limitations in clinical use:

- 1) The cost of NGS is very high due to its dependence on expensive instruments, complex operations, and highly trained operators and analysts.
- 2) The NGS platform is difficult to establish in less-developed regions. Sending samples to a special institution for NGS test will cause additional time and costs for sample preservation and transportation.
- 3) Although the NGS turnaround time can be shortened to a few days, it is still not convenient for timely efficacy evaluation and clinical decision-making and regular follow-up.

Moreover, for cancer patients with known genetic information, it will be more cost-effective to detect their personal mutation markers in regular follow-up, disease monitoring, and efficacy evaluation, instead of repetitive NGS panel test. Considering that the CoHIT method is an isothermal one-pot assay, it will be lower-cost, easier-to-use, and more advantageous for the promotion. We believe that each genotyping method has its advantages and disadvantages, and using different methods in different situations can maximize the benefits for patients in clinical practice.

To further demonstrate the multiplexing potential of the CoHIT method, we have combined the CoHIT assay with a microfluidic chip to achieve multiplexing detection. As shown in Fig. 7f-i and Supplementary Fig. S25, we made a microfluidic device with one loading chamber and nine reaction chambers. Mutation status of seven sites can be obtained simultaneously through a single loading step. To verify this point, we tested the NPM1 mutation site (Target 1) and six FLT3-ITD (internal tandem duplication) mutation sites (Targets 2~7), which are all hotspot mutations in AML. The results showed that each target could be effectively detected and the sensitivity could reach 0.1% under naked eye. Further experiments on clinical samples proved that it can simultaneously detect mutations at different sites, P85 in Fig. 7i for example.

Fig. 7 | CoHIT-LFA system and microfluidic chip-based CoHIT detection. **f** Flow chart of the microfluidic chip-based multiplexing CoHIT detection. IC, inner control. **g** Detection of 1E6 copies of mutated fragments-containing plasmid in 100 ng WT genomic DNA. **h** Detection of Target 2 at frequencies of 10%, 1%, and 0.1%. **i** Detection in clinical samples from six AML patients.

Target	Gene mutation
1	NPM1 c.863_864 4-bp insertion
2	FLT3 c.1770_1771 24-bp insertion
3	FLT3 c.1793_1794 21-bp insertion
4	FLT3 c.1790_1791 18-bp insertion
5	FLT3 c.1776_1777 18-bp insertion
6	FLT3 c.1784_1785 21-bp insertion
7	FLT3 c.1796_1797 27-bp insertion
IC	GAPDH WT

Fig. S25 | Gene mutation sites and types of targets 1~7 in the microfluidic chip-based multiplexing CoHIT assay.

2) The authors showed that CO-HIT can detect as low as 0.01% mutation burden, suggesting its potential utility in minimal residual disease detection. However, the authors used spiked in plasmid DNA but not genomic DNA. Therefore, its detection limit in real clinical practice is unproven. Furthermore, currently ddPCR for NPM1 mutation can easily reach 0.001% in clinical samples, far better than CO-HIT. The authors should be very cautious when stating the sensitivity based on plasmid spike in experiments.

Authors: Thanks for the reviewer's kind reminder. In the revised manuscript, we have added genomic DNA-based experiments to identify the limit of detection (LoD) of the CoHIT system. As shown in the following Fig. 4m-o and Supplementary Fig. S9, genomic DNA from three NPM1-mutated AML patients were tested. These patients have different c.863_864 4-bp insertion types (P33, insCATG, 39.9%; P65, insCCGG, 46.7%; P111, insTCTG, 44.0%). We diluted their DNA to concentration gradient by mixing them with WT genomic DNA from healthy people, then obtained samples with mutation rates of 25%, 10%, 5%, 1%, 0.1%, and 0.01%. The real-time fluorescence curve and terminal fluorescence statistics showed that as low as 0.01% mutation burden could be detected by CoHIT assay for all three mutation types.

Fig. 4 | Establishment and optimization of the CoHIT system. m, n Genomic DNA-based LoD assay of the CoHIT system for insTCTG and insCATG mutations using 100 ng genomic DNA as a template, reacting for 30 min at 39 °C. **o** Time-course results of the genomic DNA-based LoD assay for insCATG detection.

Fig. S9 | LoD assay of the CoHIT system using genomic DNA templates from AML patients. a The time-course of CoHIT detection of the NPM1 c.863_864insTCTG mutation, with gradient mutation ratios by mixing patient genomic DNA with healthy WT control. **b** The time-course of CoHIT detection of the insCATG mutation. **c** The time-course of CoHIT detection of the insCCTG mutation. **d** Final fluorescence statistics of figure c.

We agree with the author that ddPCR has high sensitivity and specificity. However, ddPCR has its limitations. On the one hand, the cost of building a ddPCR platform is very high, because it is composed of a droplet generation system, a thermal cycling system, and a droplet reading system, as shown in the following figure. On the other hand, the complex operations of ddPCR lead to both higher requirements for operators and greater challenges in overcoming cross contamination, compared to our one-step CoHIT assay.

Figure | Comparison of equipment needed for ddPCR and CoHIT.

3) One potential application in clinical practice could be a rapid assay for critical clinical actionable mutation detection such as NPM1, FLT3, BRAF, EGFR, etc. For that, the authors need to show it will work as a multiplex assay. With rapid improvement in NGS reporting, some laboratories have developed a short NGS panel including these mutations and can report results in 3-5 days. Therefore, unless this CO-HIT assay clearly edges NGS, its clinical utility even in this narrow space will remain questionable.

Authors: We appreciate the reviewer's suggestion. As mentioned above, in the revised manuscript, we have combined CoHIT with a microfluidic chip to achieve multiplexing detection. As shown in Fig. 7f-i, we made a microfluidic device with one loading chamber and nine reaction chambers. Mutation status of seven sites can be obtained simultaneously through a single loading step. To verify this point, we tested the NPM1 mutation site (Target 1) and six FLT3-ITD (internal tandem duplication) mutation sites (Targets 2~7), which are all hotspot mutations in AML. The results showed that each target could be effectively detected and the sensitivity could reach 0.1% under naked eye. Further experiments on clinical samples proved that it can simultaneously detect mutations at different sites, P85 in Fig. 7i for example.

Fig. 7 | CoHIT-LFA system and microfluidic chip-based CoHIT detection. **f** Flow chart of the microfluidic chip-based multiplexing CoHIT detection. IC, inner control. **g** Detection of 1E6 copies of mutated fragments-containing plasmid in 100 ng WT genomic DNA. **h** Detection of Target 2 at frequencies of 10%, 1%, and 0.1%. **i** Detection in clinical samples from six AML patients.

As for the comparison between CoHIT and NGS, as we mentioned above, NGS has significant limitations in terms of cost, turnaround time, instrument dependence, and requirement for highly trained operators or analysts (Supplementary Fig. S27). Notably, combating cancer is generally a long-term process for cancer patients, and genetic testing is needed not only in the first diagnosis but also in regular follow-up, efficacy evaluation, resistance mutation detection, and MRD monitoring. In these scenarios, for cancer patients with known genetic information, it will be more cost-effective and convenient to detect their personal mutation markers instead of repetitive NGS panel test. Considering that the CoHIT method is a rapid isothermal one-pot assay, it will be more advantageous than NGS. Taking the MRD detection as an example, NPM1 gene mutation is a clinically recommended MRD marker (Haematologica. 2022;107(12):2810-2822). As shown in Fig. 6d-g and Supplementary Fig. 22-24, the MRD detection outcomes of CoHIT assay were completely consistent with the NPM1-mutated patients' disease progression and NGS results. Based on the benefits of CoHIT in speed, sensitivity, convenience, and cost-effectiveness, we believe that the CoHIT method has the potential to become a new MRD monitoring method, particularly offering enhanced convenience to patients in underdeveloped areas.

Comparison between different genotyping methods

	FGS	NGS	qPCR	ddPCR	CoHIT
Mutation LoD (%)	10 [1]	0.01 [2]	0.01	0.001	0.01
Turnaround time	~ 1 d	> 5 d	~ 2 h	4 h [3]	< 30 min
Instrument cost	+++	+++	++	+++	+
Operational complexity	+++	+++	++	+++	+
Result analysis difficulty	+++	+++	++	++	+
Cost per sample	+	+++	++	++	+

[1] J Med Virol. 2020 Dec;92(12):3604-3608.

[2] Lancet. 2023 Jun 17;401(10393):2073-2086.

[3] Nat Biomed Eng. 2017;1:714-723.

* +, ++, and +++ means low, moderate, and high.

Fig. S27 | Comparison between different genotyping methods.

Fig. 6 | Detection of clinical samples and MRD using the CoHIT system. **d** Flowchart of the MRD experiment. **e** CoHIT result for Patient-MRD-1. The sampling dates and naked-eye results are shown for each sample. The asterisks indicate P values compared with healthy WT control (HC). **f** NGS and FGS results of Patient-MRD-1 samples dated 10/2021, 06/2022, 10/2022, and 01/2023. **g** CoHIT and NGS results for Patient-MRD-2 and 3. The asterisks indicate P values obtained by comparison with HC in CoHIT detection. The percentages are the insTCTG mutation rates determined by NGS.

Fig. S22 | FGS chromatograms of the eight bone marrow samples of Patient-MRD-1, collected between 10/2021 (First visit) and 01/2023. Mutant peaks are underlined in red. The percentages near the dates are NGS results of NPM1 gene c.863_864insTCTG mutation.

Fig. S23 | FGS, NGS, and CoHIT detection results of Patient-MRD-2. **a** Six bone marrow samples were collected between 04/2020 (First visit) and 07/2021 and tested by CoHIT assay. Final fluorescence values and naked-eye photos of the six samples are shown in chronological order. The asterisks indicate P values compared with healthy WT control (HC). **b** Time course of CoHIT detection of the six samples and HC. **c** FGS chromatograms of the six samples. Mutant peaks are underlined in red. The percentages near the dates are NGS results of the NPM1 gene c.863_864insTCTG mutation.

Fig. S24 | FGS, NGS, and CoHIT detection results of Patient-MRD-3. **a** Three bone marrow samples were collected between 10/2021 (First visit) and 06/2022 and tested by CoHIT assay. Final fluorescence values and naked-eye photos of the three samples were shown in chronological order. The asterisks indicate P values compared with healthy WT control (HC). **b** Time-course of CoHIT detection of the three samples and HC. **c** FGS chromatograms of the three samples. Mutant peaks are underlined in red. The percentages near the dates are NGS results of NPM1 gene c.863_864insTCTG mutation.

Thank the reviewer again for the thorough review and valuable feedback on our manuscript. The reviewer's comments have provided insightful perspectives that will undoubtedly enhance the quality and impact of our research.

References

- Cavo M, San-Miguel J, Usmani SZ, et al. Prognostic value of minimal residual disease negativity in myeloma: combined analysis of POLLUX, CASTOR, ALCYONE, and MAIA. *Blood*. 2022;139(6):835-844. doi:10.1182/blood.2021011101
- Pellini B, Chaudhuri AA. Circulating Tumor DNA Minimal Residual Disease Detection of Non-Small-Cell Lung Cancer Treated With Curative Intent. *J Clin Oncol*. 2022;40(6):567-575. doi:10.1200/JCO.21.01929
- Blachly JS, Walter RB, Hourigan CS. The present and future of measurable residual disease testing in acute myeloid leukemia. *Haematologica*. 2022;107(12):2810-2822. Published 2022 Dec 1. doi:10.3324/haematol.2022.282034
- Eisenstein M. Seven technologies to watch in 2022. *Nature*. 2022;601(7894):658-661. doi:10.1038/d41586-022-00163-x
- Li L, Shen G, Wu M, Jiang J, et al. CRISPR-Cas-mediated diagnostics. *Trends Biotechnol*. 2022;40(11):1326-1345. doi:10.1016/j.tibtech.2022.04.006
- Katti A, Diaz BJ, Caragine CM, et al. CRISPR in cancer biology and therapy. *Nat Rev Cancer*. 2022;22(5):259-279. doi:10.1038/s41568-022-00441-w
- Kaminski MM, Abudayyeh OO, Gootenberg JS, et al. CRISPR-based diagnostics. *Nat Biomed Eng*. 2021;5(7):643-656. doi:10.1038/s41551-021-00760-7

REVIEWERS' COMMENTS

Reviewer #1 (Remarks to the Author):

In my previous review, I indicated that the authors novel, CRISPR-based "CoHIT" approach and resulting manuscript were both high quality and thorough. My main concerns were that a) the authors could elevate the potential significance of this work by further exploring the application of minimal residual disease particularly in the discussion, and that b) the protein engineering claims were a focus of the paper and that this portion of the work could be minimized or relegated to the supplemental since their mutant combinations were derived from exhaustive directed evolution studies that were previously published.

The authors did a very thorough job of revising the manuscript to emphasize the application of minimum residual disease, and I believe this has elevated the significance of the work substantially. I have no further comments on this aspect.

The authors did additional experimentation to demonstrate the necessity for the Cas12a variants that were generated. I think this is convincing and elevates the novelty of their mutant combinations. I would suggest that the authors consider some minor writing changes to better frame and acknowledge the previous work that their protein engineering work is built off of (enAsCas12a and AsCas12a Ultra, etc.).

References

"AsCas12a ultra nuclease facilitates the rapid generation of therapeutic cell medicines" Zhang et al. Nature Communications 2021.

"Engineered CRISPR-Cas12a variants with increased activities and improved targeting ranges for gene, epigenetic and base editing" Kleinstiver et al. Nature Biotechnology 2019.

Reviewer #2 (Remarks to the Author):

The authors submitted a revised manuscript on their CoHIT assay. Overall, I appreciate all the efforts that they have put in to address my and other reviewers' comments.

Nevertheless, I am not fully convinced of the applicability of their assay for cancer in the real-world. They have chosen to argue that point-of-care tests like CoHIT is useful for cancer screening in underdeveloped areas or rural communities. However, the use case is too narrow. For example, in the manuscript, the authors developed an assay to detect 4bp insertion in NPM1. But this is only for AML and more specifically, one-third of adult AML cases. I can't imagine any rural community suffering only from AML and not other types of cancer. Hence, the authors' NPM1 assay will at most benefit only a small percentage of the population in any rural setting. This is unlike the COVID-19 pandemic, where the virus can infect almost everyone.

On a more positive note, besides the doubt on real-world applicability for cancer, I do think that the work is still interesting from an academic viewpoint. Notwithstanding the first reviewer's criticism that combining different known mutations is not truly novel, the authors have indeed found a variant that is more mismatch tolerant and the enzyme might be useful for other types of applications. A minor suggestion is that regardless of where the manuscript is published, the authors should be more careful of their wordings. In the revised manuscript, they have added in phrases like "detecting variable insertions or deletions (indels) at a single gene site is challenging" and "superior performance of CoHIT in detecting multiple prevalent NPM1 gene indels" etc. Such phrases give an initial impression that CoHIT can detect all possible indels. But this is not true. Any indel must be of the same length (e.g., 4bp insertion) as the enzyme, despite being more mismatch tolerant, is apparently very sensitive to a difference in length between its spacer and the target site. For example, if there are 1bp or 2bp insertions in the NPM1 gene, I doubt that the reported CoHIT assay can detect such insertions as the spacer length has been designed for a 4bp insertion.

Reviewer #4 (Remarks to the Author):

The authors performed additional experiments to demonstrate the potential utility of CoHIT in clinical scenario: rapid diagnosis on hotspot mutations, monitoring (MRD) and multiplex potential. The authors need to be congratulated for such extensive work to address these concerns. I have no major concerns. Several minor issues that may be worth discussing:

1. for several oncogenic mutations, such as NPM1, FLT3, BRAF, etc, MRD level below 0.01% may still be significant. therefore, they should acknowledge this potential shortcoming of Co-HIT assay as its LOD is likely not more sensitive than 0/01%.

2. Its utility in upfront mutational profiling may still be limited. this need to be pointed out.

Reviewers' comments:

Reviewer #1:

Remarks to the Author:

In my previous review, I indicated that the authors novel, CRISPR-based "CoHIT" approach and resulting manuscript were both high quality and thorough. My main concerns were that a) the authors could elevate the potential significance of this work by further exploring the application of minimal residual disease particularly in the discussion, and that b) the protein engineering claims were a focus of the paper and that this portion of the work could be minimized or relegated to the supplemental since their mutant combinations were derived from exhaustive directed evolution studies that were previously published.

The authors did a very thorough job of revising the manuscript to emphasize the application of minimum residual disease, and I believe this has elevated the significance of the work substantially. I have no further comments on this aspect.

The authors did additional experimentation to demonstrate the necessity for the Cas12a variants that were generated. I think this is convincing and elevates the novelty of their mutant combinations. I would suggest that the authors consider some minor writing changes to better frame and acknowledge the previous work that their protein engineering work is built off of (enAsCas12a and AsCas12a Ultra, etc.).

References

"AsCas12a ultra nuclease facilitates the rapid generation of therapeutic cell medicines" Zhang et al. Nature Communications 2021.

"Engineered CRISPR-Cas12a variants with increased activities and improved targeting ranges for gene, epigenetic and base editing" Kleinstiver et al. Nature Biotechnology 2019.

Authors: We appreciate the reviewer for the positive feedback and the constructive comment. As suggested, we have added the following text in our revised manuscript to clarify that our protein engineering efforts are based on prior studies, and we have cited the relevant works:

"Among these substitutions, E174R, S542R, and K548R are from the enAsCas12a (enAs) variant which has expanded target range (Nat Biotechnol. 2019;37(3):276-282.), M537R and F870L are from the AsCas12a-Ultra (AsU) variant which has improved gene editing efficiency (Nat Commun. 2021;12(1):3908.), and K548V, N552R, and K607R are from the AsCas12a-RVR variant and the AsCas12a-RR variant which recognize TATV and TYCV PAMs, respectively (Nat Biotechnol. 2017;35(8):789-792.)." (The Result section, Line 128 to 132)

Reviewer #2:**Remarks to the Author:**

The authors submitted a revised manuscript on their CoHIT assay. Overall, I appreciate all the efforts that they have put in to address my and other reviewers' comments.

Nevertheless, I am not fully convinced of the applicability of their assay for cancer in the real-world. They have chosen to argue that point-of-care tests like CoHIT is useful for cancer screening in underdeveloped areas or rural communities. However, the use case is too narrow. For example, in the manuscript, the authors developed an assay to detect 4bp insertion in *NPM1*. But this is only for AML and more specifically, one-third of adult AML cases. I can't imagine any rural community suffering only from AML and not other types of cancer. Hence, the authors' *NPM1* assay will at most benefit only a small percentage of the population in any rural setting. This is unlike the COVID-19 pandemic, where the virus can infect almost everyone.

On a more positive note, besides the doubt on real-world applicability for cancer, I do think that the work is still interesting from an academic viewpoint. Notwithstanding the first reviewer's criticism that combining different known mutations is not truly novel, the authors have indeed found a variant that is more mismatch tolerant and the enzyme might be useful for other types of applications. A minor suggestion is that regardless of where the manuscript is published, the authors should be more careful of their wordings. In the revised manuscript, they have added in phrases like "detecting variable insertions or deletions (indels) at a single gene site is challenging" and "superior performance of CoHIT in detecting multiple prevalent *NPM1* gene indels" etc. Such phrases give an initial impression that CoHIT can detect all possible indels. But this is not true. Any indel must be of the same length (e.g., 4bp insertion) as the enzyme, despite being more mismatch tolerant, is apparently very sensitive to a difference in length between its spacer and the target site. For example, if there are 1bp or 2bp insertions in the *NPM1* gene, I doubt that the reported CoHIT assay can detect such insertions as the spacer length has been designed for a 4bp insertion.

Authors: Thank the reviewer for the insightful comments and constructive suggestions. We acknowledge the challenges associated with mutational profiling and cancer screening using targeted genotyping methods such as qPCR, ddPCR, Sanger sequencing, and CRISPR technologies. Although these methods cannot detect all mutation types, they are widely utilized in clinical practice due to their shorter turnaround, lower equipment requirements, and lower cost compared to NGS. Our study leverages the ease of designing and synthesizing crRNA for CRISPR-based detection for diverse cancer targets, thereby potentially benefiting a larger patient base. While the *NPM1* 4bp insertion assay primarily aids a subset of AML patients, our methodology serves as a template for adapting point-of-care testing to various cancers through the customization of crRNAs and primers. This adaptability could indeed broaden the application of our assay, mitigating the limitations the reviewer noted.

We appreciate the reviewer's advice on precise wording. In response, we have carefully revised our manuscript to eliminate possible misinterpretations. The terminology has been updated to more precisely define CoHIT's capabilities, using phrases such as "the ability of CoHIT in detecting multiple prevalent *NPM1* gene c.863_864 4-bp indels" to better reflect the focused scope of our assay.

Moreover, we have taken into consideration the concern regarding the detection of indels of varying lengths. This problem can be effectively solved by mixing crRNAs that target different-length indels. For example, if there are several kinds of *NPM1* gene c.863_864 4-bp insertions, several kinds of c.863_864 1-bp insertions, and several kinds of c.863_864 2-bp insertions, a one-pot CoHIT system with three crRNAs has the potential to detect these different insertions. The CoHIT method aims to use less crRNAs to target more indels.

Reviewer #4:

Remarks to the Author:

The authors performed additional experiments to demonstrate the potential utility of CoHIT in clinical scenario: rapid diagnosis on hotspot mutations, monitoring (MRD) and multiplex potential. The authors need to be congratulated for such extensive work to address these concerns. I have no major concerns. Several minor issues that may be worth discussing:

1. for several oncogenic mutations, such as NPM1, FLT3, BRAF, etc, MRD level below 0.01% may still be significant. therefore, they should acknowledge this potential shortcoming of Co-HIT assay as its LOD is likely not more sensitive than 0/01%.
2. Its utility in upfront mutational profiling may still be limited. this need to be pointed out.

Authors: We are grateful for the reviewer's positive feedback and professional suggestions. We have included a discussion on the LoD and limitations of mutational profiling in the revised manuscript:

"It should be noted that although our LoD for NPM1 mutation is 0.01%, MRD level below 0.01% may still be significant for several mutated oncogenic genes, such as *NPM1*, *FLT3*, and *BRAF*. The standard definition of MRD negativity is dependent on the type of MRD technology and the target (Lancet. 2023;401(10393):2073-2086.). Using microfluidic chip-based multiplexing detection, we successfully detected seven mutation loci simultaneously, which is useful for patients carrying multiple mutations. However, its utility in upfront mutational profiling may be limited, making it more suitable for regular follow-up, efficacy evaluation, and MRD monitoring."(The Discussion section, Line 376 to 382)